nature
ecology & evolution
# A single-cell transcriptomic atlas tracking the neural basis of division of labour in an ant superorganism

Qiye Li [1,2,12], Mingyue Wang[1,12], Pei Zhang[1,12], Yang Liu[1,12], Qunfei Guo[1,2], Yuanzhen Zhu[1], Tinggang Wen[1], Xueqin Dai[3,4], Xiafang Zhang[3,4], Manuel Nagel [5], Bjarke Hamberg Dethlefsen[5], Nianxia Xie [1,2], Jie Zhao[3], Wei Jiang[1], Lei Han[1], Liang Wu[1,2], Wenjiang Zhong[3,4], Zhifeng Wang[1], Xiaoyu Wei[1], Wei Dai[1], Longqi Liu [1,2,6], Xun Xu [1,2,7], Haorong Lu [8], Huanming Yang[1,9], Jian Wang[1,9], Jacobus J. Boomsma [5], Chuanyu Liu [1 ✉], Guojie Zhang [1,3,5,10,11 ✉] and Weiwei Liu [3 ✉]

Ant colonies with permanent division of labour between castes and highly distinct roles of the sexes have been conceptualized to be superorganisms, but the cellular and molecular mechanisms that mediate caste/sex-specific behavioural specialization have remained obscure. Here we characterized the brain cell repertoire of queens, gynes (virgin queens), workers and males of *Monomorium pharaonis* by obtaining 206,367 single-nucleus transcriptomes. In contrast to *Drosophila*, the mushroom body Kenyon cells are abundant in ants and display a high diversity with most subtypes being enriched in worker brains, the evolutionarily derived caste. Male brains are as specialized as worker brains but with opposite trends in cell composition with higher abundances of all optic lobe neuronal subtypes, while the composition of gyne and queen brains remained generalized, reminiscent of solitary ancestors. Role differentiation from virgin gynes to inseminated queens induces abundance changes in roughly 35% of cell types, indicating active neurogenesis and/or programmed cell death during this transition. We also identified insemination-induced cell changes probably associated with the longevity and fecundity of the reproductive caste, including increases of ensheathing glia and a population of dopamine-regulated *Dh31*-expressing neurons. We conclude that permanent caste differentiation and extreme sex-differentiation induced major changes in the neural circuitry of ants.

Socially advanced ants appear to have brain cell numbers comparable to solitary fruit flies[1,2] and their brains are smaller than in many weakly social or solitary wasps and bees[1], indicating that social complexity is not obviously correlated with larger brains. Instead, remodelling of neural circuits and functional cellular innovations are probably more important predictors of social complexity[3], particularly in social systems where brain development is caste-specific and developmentally hardwired. William Morton Wheeler was the first to identify that the highly divergent and complementary specialization of caste phenotypes resembles the ontogenetic differentiation of cell lineages in metazoans. This led him to coin the term superorganism for ant colonies to highlight the fundamental difference with animal societies where most individuals remain behaviourally and reproductively totipotent[4,5]. Permanent reproductive division of labour has indicated that the roles of the sexes have also become highly specialized and stereotyped[6,7]. It thus seems reasonable to propose that the superorganismal answer to social life of higher organizational complexity has been brain specialization rather than brain enlargement[8].

Complex social behaviours are governed by neural circuits whose structure and function are determined by underlying gene regulatory networks, but the operational details remain poorly understood. Some recent studies have combined single-cell transcriptomics with neuroanatomy to better understand the organization of primate brains[9–11], but such approaches have barely been developed for ants. Comparative transcriptomics have identified many differentially expressed genes (DEGs) across ant castes using whole-bodies[12–14] or pooled brain tissues[15–17], but have lacked the resolution to map the heterogeneity of brain cells and gene expression differences across cell populations. So far, only a single ant species, *Harpegnathos saltator*, has been interrogated at the single-cell level and only for the midbrains of workers and gamergates (inseminated and reproductively active workers)[18]. However, comprehensive profiling of whole brain single-cell transcriptomes across the full panel of distinct adult phenotypes of different sexes, castes and reproductive roles is necessary to understand how brain functions combine phenotypic specialization with integration in a superorganismal colony.

Inspired by Wheeler's superorganism concept, we combined the power of massively parallel single-nucleus RNA-sequencing (snRNA-seq) with the unique biology of the pharaoh ant *M. pharaonis* to interrogate the neural correlates underlying obligate division

[1]BGI-Shenzhen, Shenzhen, China. [2]College of Life Sciences, University of Chinese Academy of Sciences, Beijing, China. [3]State Key Laboratory of Genetic Resources and Evolution, Kunming Institute of Zoology, Chinese Academy of Sciences, Kunming, China. [4]Kunming College of Life Science, University of Chinese Academy of Sciences, Kunming, China. [5]Section for Ecology and Evolution, Department of Biology, University of Copenhagen, Copenhagen, Denmark. [6]Shenzhen Bay Laboratory, Shenzhen, China. [7]Guangdong Provincial Key Laboratory of Genome Read and Write, Shenzhen, China. [8]China National GeneBank, BGI-Shenzhen, Shenzhen, China. [9]James D. Watson Institute of Genome Science, Hangzhou, China. [10]Center for Excellence in Animal Evolution and Genetics, Chinese Academy of Sciences, Kunming, China. [11]Present address: Evolutionary and Organismal Biology Research Center, School of Medicine, Zhejiang University, Hangzhou, China. [12]These authors contributed equally: Qiye Li, Mingyue Wang, Pei Zhang, Yang Liu. ✉e-mail: liuchuanyu@genomics.cn; guojiezhang@zju.edu.cn; liuweiwei@mail.kiz.ac.cn

of labour and reproductive specialization. Pharaoh ant queens are inseminated within the nest and establish new colonies through budding, rather than alone after mating flights[19]. Colonies are always highly polygynous: that is, many egg-laying queens coexist peacefully in a nest[20]. A typical pharaoh ant colony has three other phenotypes besides queens: gynes, workers and males. Gynes are virgin reproductives that will become queens after insemination, but will assume worker tasks and express reduced lifespans when they fail to become inseminated within a narrow time window after hatching from the pupal stage[21,22]. Workers are permanently sterile lacking both ovaries and sperm storage organs, and are responsible for all colony maintenance tasks[20]. Males are very short-lived and only meant to inseminate gynes[23]. The special social biology of *M. pharaonis* allowed mass rearing in the laboratory and collection of abundant brain tissues from all four adult phenotypes for comparative snRNA-seq analysis in a well-controlled sampling scheme. This allowed us to map important aspects of multi-brain complementarity and functional coordination in a superorganismal ant colony.

## Results

**Cell-type classification in *M. pharaonis* brains.** To create a comprehensive cell atlas, whole brain snRNA-seq was performed for four to five biological replicates of each adult phenotype: gynes (*n* = 4), queens (*n* = 4), males (*n* = 4) and workers (*n* = 5) (Fig. 1a and Supplementary Data 1). After stringent quality control and filtering, we obtained an average of roughly 50,000 high-quality nuclei from each of these four phenotypes, adding up to 206,367 nuclei (Extended Data Fig. 1a–d). This is 1.3 to 4 times the estimated cell number of 50,000–150,000 in a single individual ant brain[1], and an order of magnitude higher than the recent study that obtained 18,583 cells for the midbrains (that is, optic lobes (OLs) removed) of the ant *H. saltator*[18] (Supplementary Data 2). Correlations between gene expression quantifications via snRNA-seq and conventional bulk RNA-seq were high for each phenotype (Pearson's *r*, 0.88–0.91), confirming that our snRNA-seq data were representative for the functionality of entire brains (Extended Data Fig. 1e).

Overall, the 206,367 nuclei separated into 43 cell clusters with distinct gene expression patterns (Fig. 1b; see Supplementary Data 3 for a 3D view). All clusters showed high reproducibility across the biological replicates within phenotypes (Supplementary Data 1), suggesting that none of them are artefacts resulting from batch effects. By examining the expression of known marker genes from *Drosophila* and hymenopteran species, we could clearly distinguish the neurons from the glia (Extended Data Fig. 2a) and annotate many clusters to known cell types in insect brains, including Kenyon cells (KCs), olfactory projection neurons (OPNs), monoaminergic neurons, astrocytes, ensheathing glia, cortex glia, surface glia and photoreceptors (which may come from ocelli that were not completely removed) (Fig. 1b and Extended Data Fig. 2b,c).

To annotate the OL cell types for which well-established marker genes are lacking, we mapped the *Monomorium* cell clusters to those identified in adult *Drosophila melanogaster* whole brains[24] and *H. saltator* midbrains[18] on the basis of transcriptional similarity of orthologous genes. The mean area under the receiver operator characteristic curve (AUROC) score acquired with MetaNeighbor[25] was adopted to quantify the pairwise similarity of cell clusters between species. This revealed a total of 35 *Monomorium* clusters (81.4%) with high similarity (AUROC > 0.9) to at least one *Drosophila* cell cluster. Moreover, the cell-cluster dendrogram based on AUROC scores remained structured according to cell categories rather than species (Extended Data Fig. 2d), indicating that the main brain cell types are highly conserved across the two insects. A slightly lower proportion of the *Monomorium* cell clusters (31/43, 72.1%) could be mapped to *Harpegnathos*, probably due to all OL cell types being absent in the *Harpegnathos* midbrain dataset[18]. Nevertheless, these cross-species mapping analyses allowed us to

identify six *Monomorium* clusters as putative OL neurons for ants, because they clearly grouped with the *Drosophila* OL clusters in the *Monomorium*-versus-*Drosophila* tree and formed a single clade in the *Monomorium*-versus-*Harpegnathos* tree (Extended Data Fig. 2d,e). Taken together, our combined efforts led to the annotation of 70% (30/43) of the cell clusters identified across the brains of the four adult *Monomorium* phenotypes.

**Cell compositional differences between ant and fly brains.** By comparing the relative abundances of cell types in adult brains of *M. pharaonis*, *H. saltator* and *D. melanogaster*, we found that the most striking difference between the ant and fly brains concerns the KCs: the intrinsic neurons of the mushroom bodies, the centre of associative learning and memory in insects[26]. The KCs alone represent roughly 24% of cells in the whole brains of *Monomorium* and roughly 36% of the *Harpegnathos* midbrain cells[18], in sharp contrast to the mere 5 and 10% of cells in the *Drosophila* whole brains[24] and midbrains[27], respectively (Fig. 2a,b, Extended Data Fig. 3 and Supplementary Data 4). Another notable difference between the ant and fly brains was observed for the OPNs, a group of neurons that transfer olfactory information from the antennal lobes to the higher olfactory centres[28,29]. On average, the relative abundances of OPNs in entire *Monomorium* brains (roughly 3.0%) and *Harpegnathos* midbrains (roughly 3.2%) were three times and twice higher than in *Drosophila*, respectively (Fig. 2a,b). These higher abundances of KCs and OPNs observed in ant brains are consistent with the typical adaptations of ants to social life on the surface and underground where olfactory communication is key[30], in contrast to the often airborne solitary flies.

**Diversification and evolution of mushroom body KCs.** The *Monomorium* KCs are characterized by a high overall expression of *Pka-C1*, *trio* and *PLCε* (Extended Data Fig. 2b). RNA in situ hybridization (ISH) of *Pka-C1* labelled the cell bodies of KCs around the calyces of the mushroom bodies (Extended Data Fig. 4a), consistent with previous observations in other hymenopteran insects[18,31]. The *Monomorium* KCs are highly diverse and could be divided into 12 transcriptionally differentiated cell clusters in the uniform manifold approximation and projection (UMAP) space (Fig. 1b). These 12 *Monomorium* KC clusters could be clearly separated into two distinct classes (class-A and class-B) according to gene expression-based clustering analysis (Fig. 3a). Class-A comprised eight of the 12 clusters and preferentially expressed *CaMKII* and *Mblk-1*, the marker genes of large-type (non-compact) KCs in adult honeybee brains[32,33], whereas the class-B KCs were characterized by preferential expression of *dati*, *Rbp6* and *Cow* (Fig. 3b). Moreover, the *Monomorium* class-A KCs could be divided further into three subclasses (KCA-1, KCA-2 and KCA-3; Fig. 3a). The class-A KCs were almost twice as abundant as the class-B KCs in all three female phenotypes, whereas they were slightly less abundant than class-B KCs in male brains (Fig. 3c), indicating a differential rate of neurogenesis of these two KC classes between the sexes during development.

We next assessed the importance of each KC subtype for different social roles by comparing their relative abundances against total brain cells across the four adult *Monomorium* phenotypes. This showed that almost all class-A KC subtypes had the highest abundances in worker brains, whereas half of the class-B KC subtypes revealed higher abundancies in male or gyne brains (Fig. 3d). The class-A KCs might thus be particularly important for regulating worker behaviours, while some of the class-B KCs have probably been co-opted for mating-related behaviours in newly emerged reproductives. Consistent with this conjecture, functional enrichment analysis for the DEGs between KC (sub)classes showed that up-regulated DEGs in class-A KCs were enriched in cGMP- and cAMP-mediated signalling involved in memory formation[34],

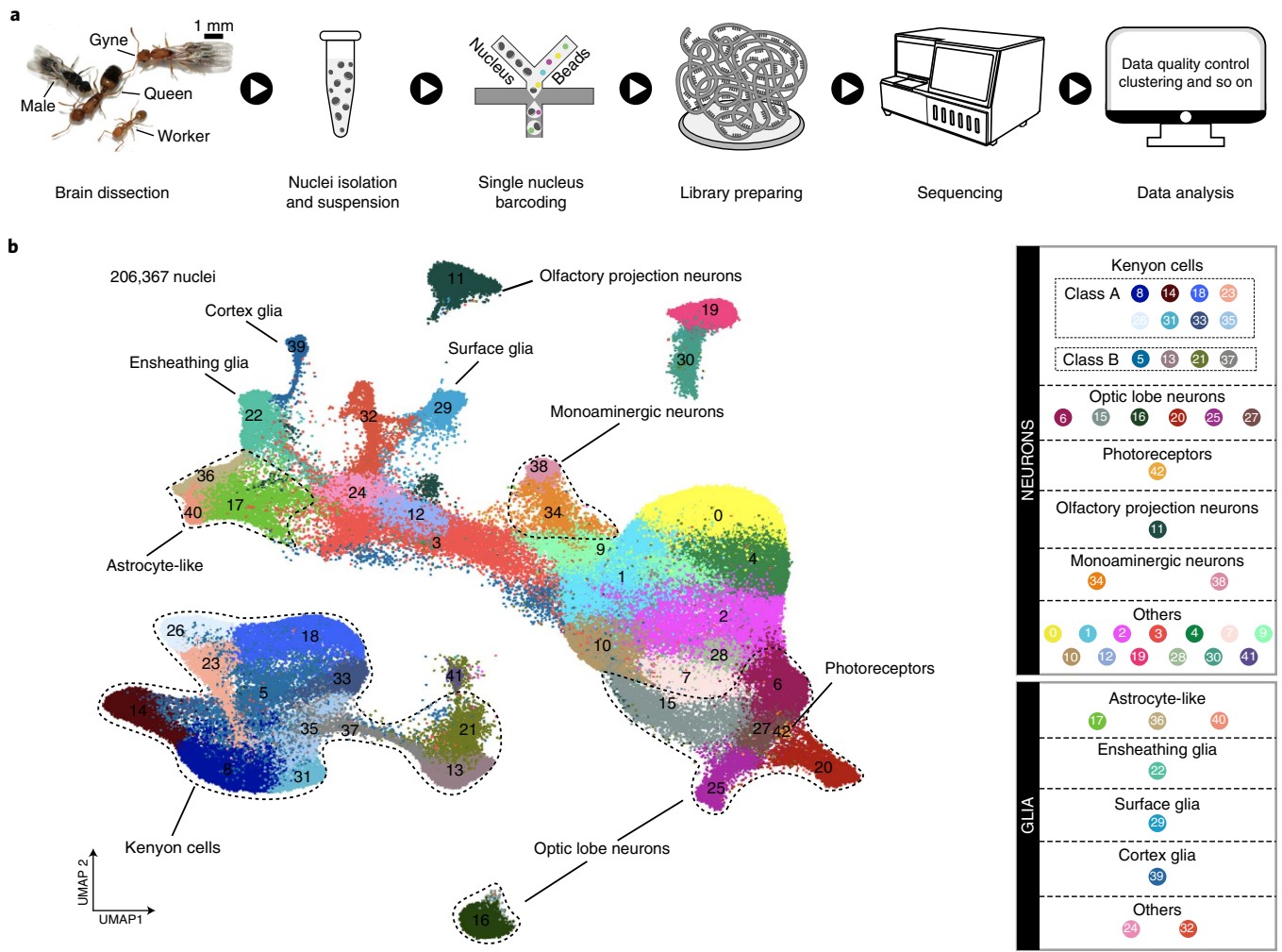

**Fig. 1 | Transcriptomic classification of cell types in ant brains. a**, The four adult phenotypes of *M. pharaonis* and a schematic overview of the overall experimental design. Four to five biological replicates for each adult phenotype were prepared for snRNA-seq. For a single biological replicate of an adult phenotype, nuclei for snRNA-seq were isolated from a pool of 30 to 50 whole brains. **b**, UMAP plot of the 43 cell clusters generated by grouping the 206,367 nuclei obtained from brains of workers, queens, gynes and males. Each dot represents one nucleus. See legend for numerical and colour coding. See also Supplementary Data 1 for the number of nuclei per cluster in each replicate of the four adult phenotypes.

associative learning and feeding behaviour, while those up-regulated in class-B KCs were enriched in taxis movement, circadian rhythm and neurogenesis (Fig. 3e).

To explore the evolutionary origin of the *Monomorium* KC subtypes, we next assessed the transcriptional similarity of KC clusters across *M. pharaonis*, *H. saltator*[18], *Apis mellifera* (honeybee)[35] and *D. melanogaster*[24,27,36] (Extended Data Fig. 4b,c). *Harpegnathos* lives in small colonies and represents an early-branching ant lineage that has been separated from *Monomorium* for at least 130 million years[37,38], while the honeybee independently evolved advanced superorganismal caste differentiation after the divergence of bees and ants roughly 160 million years ago[39]. In spite of these huge phylogenetic distances, we found that the *Monomorium* class-A and class-B KCs were very similar to two distinct groups of KC clusters in *Harpegnathos* and the honeybee (Fig. 3f and Extended Data Fig. 5a–c). This suggests that these two main KC classes evolved before the emergence of complex social life in the Hymenoptera. It is also notable that the three subclasses of class-A KCs were probably established early in ant evolution, because the *Monomorium* KCA-1, KCA-2 and KCA-3 were most similar to three distinct *Harpegnathos* class-A KC clusters (Fig. 3f). However, in contrast to

class-A, the relationships of the class-B KC clusters across the three hymenopteran species were less clear (Fig. 3f), which indicates that the class-B KCs probably underwent independent diversification in these three distantly related hymenopteran lineages.

In adult *Drosophila* brains, KCs are classified into three subtypes (γ, α'/β' and α/β) on the basis of their axonal projection patterns[40]. Consistent with the substantial morphological differences of the mushroom bodies between ants and flies[26], most *Monomorium* KC clusters showed low similarity to the three *Drosophila* KC subtypes. However, it was intriguing to see that the *Monomorium* c13 and c21 KCs showed high transcriptional similarity to the *Drosophila* α'/β' KCs (Fig. 3f), as validated by three independent *Drosophila* datasets[24,27,36] with AUROC scores over 0.9 (Extended Data Fig. 5d). In fact, many marker genes were shared by *Drosophila* α'/β' KCs and *Monomorium* c13/c21 KCs, such as *msi*, *Rbp6* and *dlg1* (Extended Data Fig. 5e,f), indicating that the *Monomorium* c13/c21 KCs may account for similar functions to *Drosophila* α'/β' KCs that are important for adult life unrelated to sociality.

**Insect behaviour regulation by conserved OL cells.** The OL neurons formed another large cell population, varying from 3 to 28% of

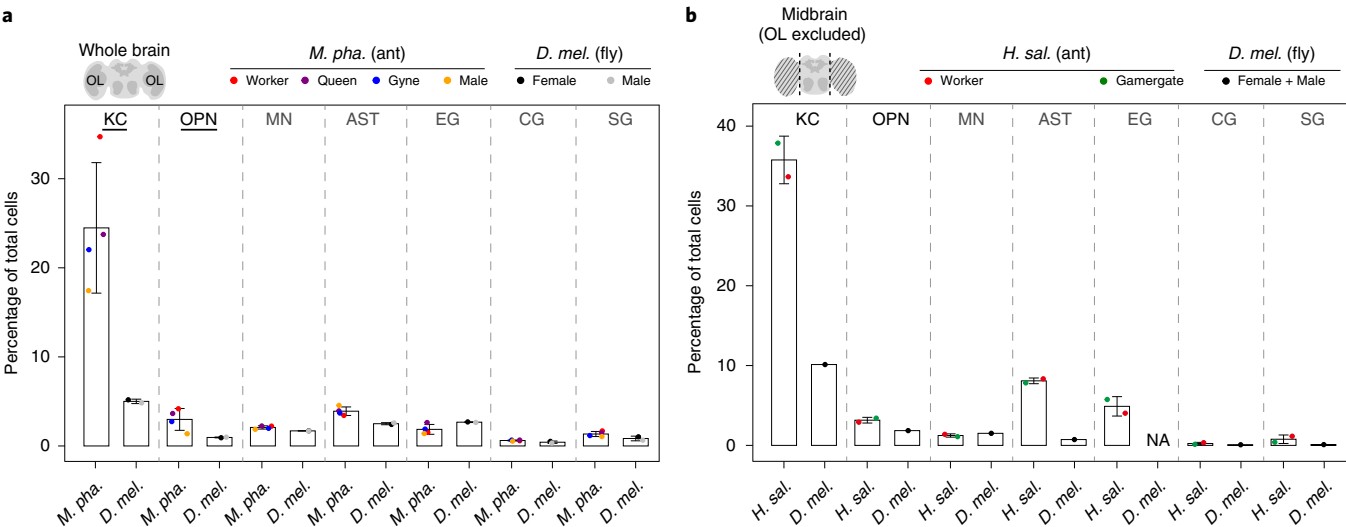

**Fig. 2 | Cell compositional differences between brains of adult ants and flies. a,b,** Abundance of different cell types relative to total cells in entire brains of *Monomorium* and *Drosophila*[24] (**a**) and in midbrains of *Harpegnathos*[18] and *Drosophila*[27] (**b**). *M. pha.*, *M. pharaonis*; *H. sal.*, *H. saltator* and *D. mel.*, *D. melanogaster*. The schematic brains on the top left corners illustrate the anatomical differences between a whole brain and a midbrain. In the plots, each dot presents the relative abundance of a cell type in an ant adult phenotype or in a *Drosophila* sex. The relative abundance of a focal cell type in a specific phenotype or sex was measured as the percentage of cells belonging to the focal cell type out of the total number of cells in a specific phenotype or sex after combining cells from all biological replicates (or all libraries) of the ant phenotype or *Drosophila* sex. Accordingly, bars are the corresponding means ± s.d. across ant adult phenotypes ($n = 4$ for *Monomorium* and 2 for *Harpegnathos*) or *Drosophila* sexes ($n = 2$). Cell types with significant abundance difference assessed by scCODA and with a more than twofold change in relative abundance between species are underlined in **a**. Note that the original *Drosophila* midbrain dataset[27] mixed cells from both sexes and that EG could not be detected in this dataset (Extended Data Fig. 3), so the scCODA assessment was not available for the midbrain datasets. MN, monoaminergic neuron; AST, astrocyte; EG, ensheathing glia; CG, cortex glia; SG, surface glia and NA, not available. See also Supplementary Data 4 for data related to this figure.

all cells in adult *Monomorium* brains. All *Monomorium* OL clusters showed high transcriptional similarity to at least one *Drosophila* OL cell type[41,42], indicating functional conservation of these OL neurons in insects (Fig. 4a,b and Extended Data Fig. 6a–c).

All *Monomorium* OL clusters displayed the highest abundances in male brains across the four adult phenotypes, and among them, c16 was most male-biased (Fig. 4c). This population of neurons, which occupies roughly 6% of the male brain cells, was completely absent in worker brains and was 2.5 and 5.8 times less represented in brains of gynes and queens, respectively (Fig. 4d). The DEGs up-regulated in c16 were enriched by genes involved in *Drosophila* male courtship behaviour (Fig. 4b), such as *Nlg2*, *tipE* and *DopEcR* (Fig. 4e). *Nlg2*-deficient male flies express less female-directed courtship and lower aggression to other males[43]. The *Monomorium* cl6 neurons showed the highest transcriptional similarity with *Drosophila* T4/T5 neurons, which mediate motion detection required for successful mating[44,45] and also preferentially express *Nlg2* (Fig. 4a and Extended Data Fig. 6d,e). In addition, the T4/T5 neurons are located near the lobula plate of *Drosophila* OLs[46] (Fig. 4a), in almost the same location that we observed for the c16 neurons in *Monomorium* by RNA ISHs of *Nlg2* (Fig. 4f). Our ISH assessments also confirmed the absence of c16 in workers and its higher abundance in males compared to gynes (Fig. 4f). These results indicate that this population of OL neurons probably play a conserved role in regulating male mating behaviour in insects regardless of sociality and that they are particularly important in the highly specialized males of ants.

Among the female phenotypes, gynes had the highest representation for all OL clusters. The only exception was c20, which was equally abundant in queens (Fig. 4d), suggesting that some vision-related functions are retained in mature egg-laying queens even though they mostly operate in the dark nest environment. The up-regulated DEGs in c20 were significantly enriched for

genes involved in *Drosophila* circadian rhythm regulation (Fig. 4b), such as *ort*, *qvr*, *GABA-B-R3* and *wake* (Fig. 4e). In addition, c20 displayed the highest transcriptional similarity to *Drosophila* lamina monopolar cells (LMCs), which reside near the lamina of *Drosophila* OLs[46], as confirmed by RNA ISH of *GABA-B-R3* and found to be also true in *Monomorium* (Fig. 4g). The LMCs in insects can dynamically optimize visual perception over a wide range of light levels[47–50], indicating that these neurons are probably involved in circadian behaviour by responding to light. Consistent with the significant contraction of all OL clusters except for c20 in queens, we propose that most vision-related functions have degenerated in mature *Monomorium* queens, while the retained sensitivity to light intensity changes allows queens to assess the optimal time for nest-budding dispersal and to quickly retreat to the dark inner nest on unexpected nest disturbance[51].

**Specialization and complementation of social brains.** To investigate the extent to which ant brains are differentiated among colony members, we compared relative cell-type abundances across the four adult *Monomorium* phenotypes. We found that almost all 43 cell clusters are present in all phenotypes, except that c16 (T4/T5 neurons) and c42 (ocellus photoreceptors) were absent in worker brains (Fig. 5a and Supplementary Data 1). However, the abundances of up to 86% (37/43) of cell clusters showed significant differences across phenotypes as assessed by scCODA[52] (Fig. 5b). The largest differences were observed between the sexes (males versus gynes/queens/workers), with 65–70% of cell clusters showing significant differences, followed by 49–56% differences between castes (workers versus gynes/queens). In contrast, we could not detect any cell clusters showing significant abundance differences between the sexes of *Drosophila* as assessed with two independent datasets[24,36] (Fig. 5c and Supplementary Data 5). These results confirm that the sexual and caste phenotypes of *Monomorium* are developmentally

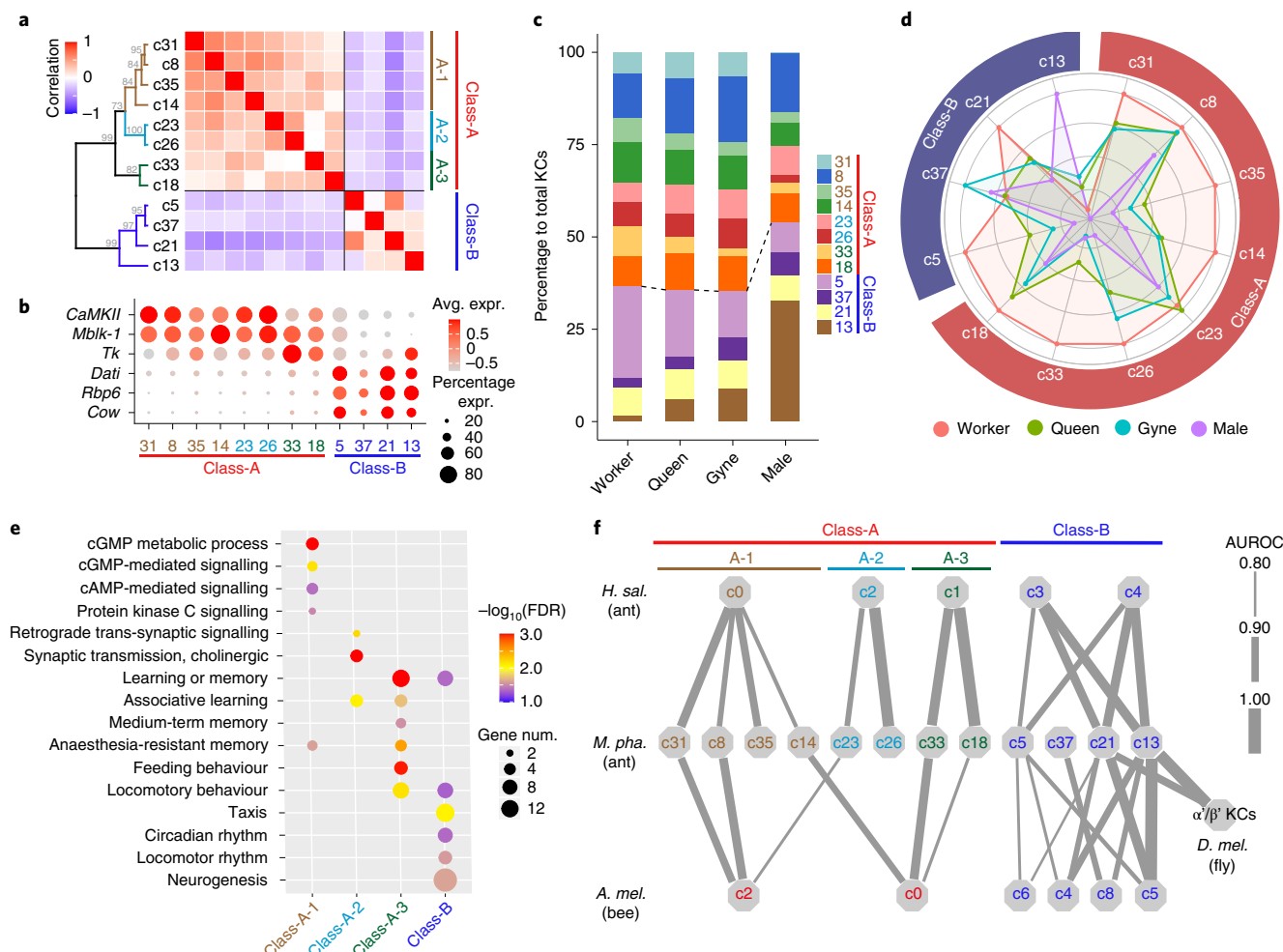

**Fig. 3 | The diversity of KCs in ant brains. a**, Pairwise Pearson correlations and hierarchical clustering of the 12 KC clusters in *M. pharaonis* brains based on gene expression, showing a clear division into two main classes (A and B). The grey numbers at the branches are bootstrap values. **b**, Dot plot showing the expression of representative DEGs between class-A and -B KCs. Dot colours represent average expression of a gene and dot sizes represent percentages of cells within each cluster expressing that gene. **c**, Bar plots showing the proportion of cells from each KC cluster against the total number of KCs in each of the four adult phenotypes, with the dashed line marking the boundary between class-A and -B KCs. **d**, Radar plot showing the variation in relative abundance of each KC subtype against total brain cells across *Monomorium* phenotypes. For each KC cluster, the mean across replicates for a phenotype was determined first and then divided by the maximum among the four phenotypes. See Supplementary Data 1 for the exact number of nuclei per KC cluster per phenotype. **e**, Representative GO terms enriched (FDR < 0.05) by the DEGs that were up-regulated in each KC (sub)class relative to the remaining KCs. Dot colours represent FDR values for each GO term, and dot sizes represent the number of DEGs associated with each GO term. **f**, Correspondence of KC clusters between *Monomorium* and *Harpegnathos/Apis/Drosophila* as predicted from the transcriptional similarities of orthologous genes by MetaNeighbor. A higher AUROC score means higher similarity. Each line links a *Monomorium* KC cluster to its top hit among the *Harpegnathos/Apis/Drosophila* KC clusters according to AUROC scores, with line thickness being proportional to the score. Only hits with AUROC > 0.80 are shown. A second hit is plotted as well when the difference between the top and second AUROC score was less than 0.05. AUROC scores for *Drosophila* α'/β' KCs are mean values across the three independent datasets (Extended Data Fig. 5d).

specialized to a high extent, and these specializations might have been resulted from the differential investment of a common set of cell types during development.

It appeared that the gyne and queen brains have a fairly generalized cell composition with moderate abundances for almost all cell clusters (Fig. 5d). By contrast, male brains had the highest abundances not only in all OL clusters, but also for two of the three astrocyte clusters, while they had the lowest abundances in OPNs and almost all KC-related clusters. These results clearly indicate that *Monomorium* males rely heavily on visually guided behaviours, even though nuptial flights appear to have lost in *M. pharaonis*[53]. Although mating in the laboratory happens easily within the same colony, these results made us speculate that *M. pharaonis* males have retained diurnal dispersal behaviour under natural conditions to prevent very close inbreeding. The worker brains were mostly characterized by cell-type preferences opposite to male brains, displaying the highest abundances in almost all KC clusters and the OPNs, but the lowest abundances in vision-related neurons, suggesting that learning, memorizing and processing of olfactory information are most important for worker behaviour. These data made us predict that worker brains should have the largest mushroom bodies and ALs while male brains should have the largest OLs in terms of relative volume, expectations that were confirmed by reconstructing the main neuropils of the four adult phenotypes with confocal microscopy image stacks (Fig. 5e, Extended Data Fig. 7 and Supplementary Data 6). We also observed an overall negative correlation between male versus gyne and worker versus gyne cell-type abundance changes, suggesting that male and worker brains are

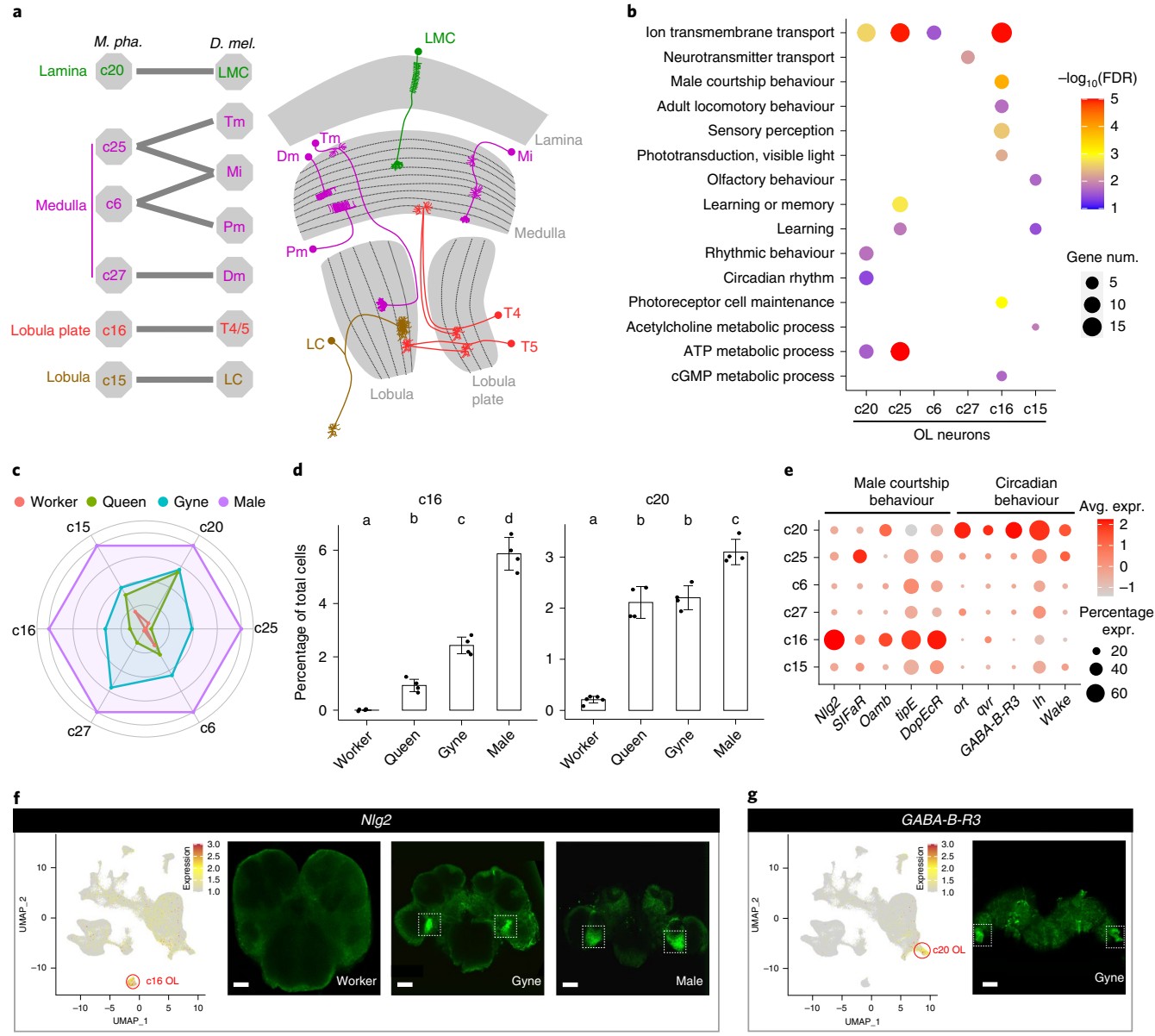

**Fig. 4 | The conserved OL neurons between *Monomorium* and *Drosophila*. a**, Annotation of the *Monomorium* OL clusters (left) based on transcriptional similarity comparison with two independent *Drosophila* OL single-cell datasets[41,42] (Extended Data Fig. 6) and a schematic diagram of the *Drosophila* OL (right) highlighting the OL cell types conserved in *Monomorium*. Tm, transmedullary neuron; Mi, medulla intrinsic neuron; Pm, proximal medulla neuron and LC, lobula columnar cells. **b**, Representative GO terms enriched (FDR < 0.05) by the up-regulated DEGs in each OL cluster relative to the remaining OL neurons. Dot colours represent FDR values for each GO term, and dot sizes represent the number of DEGs associated with each GO term. **c**, Radar plot showing the variation in relative abundance of each OL cell type against total brain cells across phenotypes. For each OL cluster, the mean across replicates of a phenotype was determined first and then divided by the maximum among the four phenotypes. **d**, Percentage of cells from c16 (left) and c20 (right) against the total number of brain cells in each adult phenotype. Each dot presents the biological replicate value of an adult phenotype (n = 5 for workers, 4 for queens, 4 for gynes and 4 for males), bars are means ± s.d. across replicates and lowercase letters assign bars to different groups that were significantly different as assessed by scCODA. **e**, Expression of representative top DEGs from c16 and c20 across all OL clusters. Dot colours represent average expression level of a gene and dot sizes represent percentages of cells within each cluster expressing that gene. **f,g**, UMAP plot and whole-mount RNA in situ detection of *Nlg2* (**f**) and *GABA-B-R3* (**g**) in the brains of focal adult phenotypes. The UMAP plots are coloured by gene expression (grey is low and red is high), with red circles indicating the cell clusters that preferentially expressed the focal marker genes. White dotted boxes indicate the positions of hybridization signals in the brain images. Scale bars in **f**; 40, 50 and 50 μm, respectively and **g**, 50 μm.

partially complementary to each other at cellular composition level in a *Monomorium* colony (Fig. 5f).

**Dopamine circuit remodelling induced by gyne insemination.** Insemination is a crucial single step for gynes to become queens with full reproductive functions because ants never re-mate later

in life[7]. A previous study has shown that gyne–queen role differentiation involves substantial brain anatomic changes and parallel remodelling of gene regulatory networks in *M. pharaonis*[21]. We detected significant abundance changes in 35% (15/43) of the cell clusters between gyne and queen brains (Fig. 6a), corroborating those previous findings[21] and suggesting that active neurogenesis

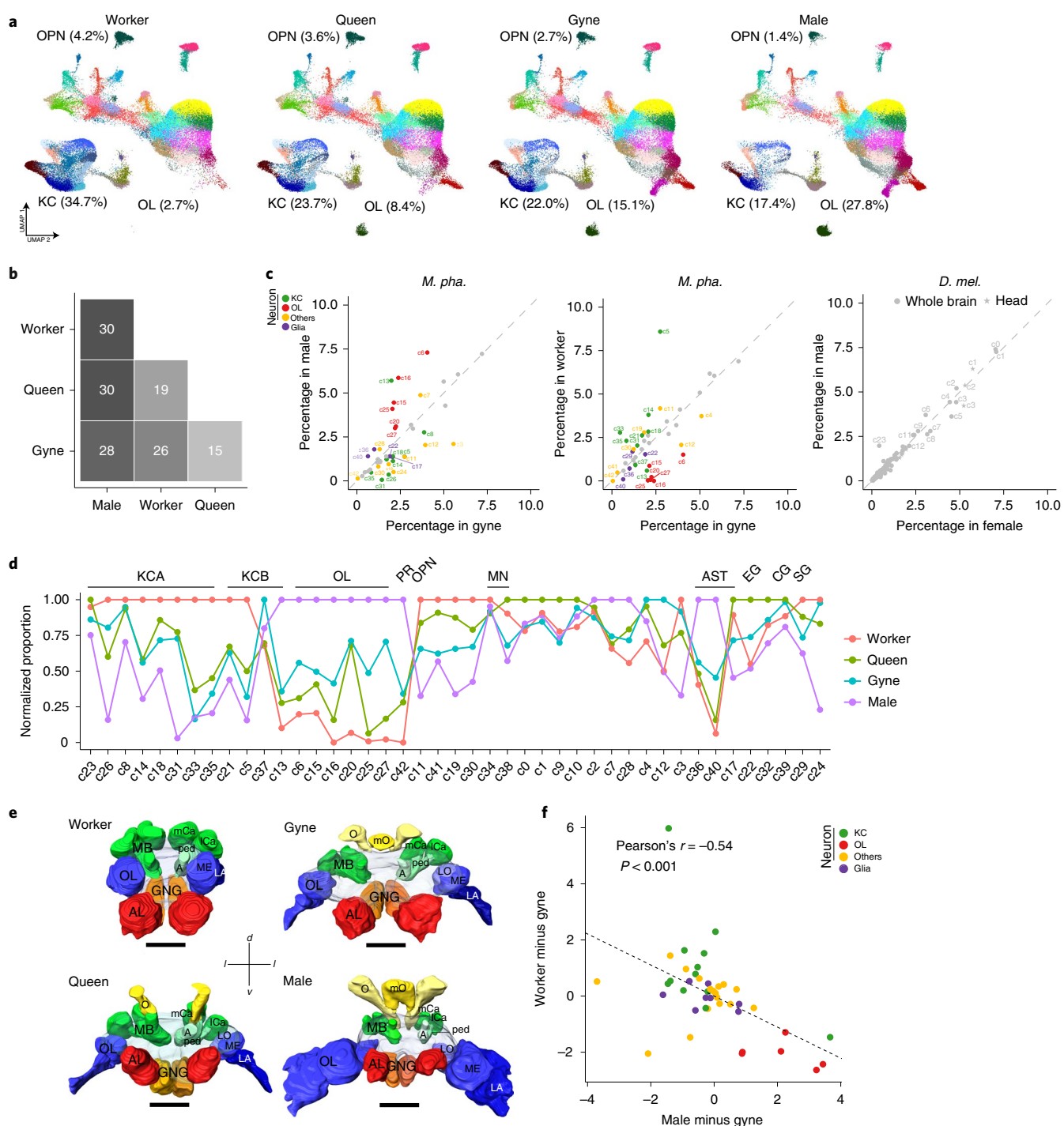

**Fig. 5 | Specialization and complementarity of *Monomorium* brains. a**, UMAP plots of the 43 clusters in the brains of workers, queens, gynes and males. Each dot represents one nucleus and is coloured according to cell cluster as in Fig. 1b. The relative abundances of KCs, OL neurons and OPNs against total number of brain cells in each adult phenotype are also presented. **b**, The number of cell clusters showing significant abundance differences, assessed by scCODA and with >1.3-fold changes, between any two of the four adult phenotypes. **c**, Cell clusters that displayed significant abundance differences between the sexes (left) and castes (middle) in *Monomorium* (corresponding to **b**), with the differences between female and male *Drosophila* brains[24] and heads[36] as controls (right). Coloured dots represent cell clusters with significant abundance differences and grey dots/stars represent those with no significant differences. **d**, The variation in relative abundance of each cell type against total brain cells across phenotypes. For each cell cluster, the phenotype-specific mean across replicates was determined first and then divided by the maximum among the four phenotypes. **e**, 3D brain reconstructions of a worker, queen, gyne and male using confocal microscopy image stacks (an anterior view). MB, mushroom body; mCa, medial calyx of MB; lCa, lateral calyx of MB; ped, peduncle of MB; A, alpha lobe of MB; LO, lobula of OL; ME, medulla of OL; LA, lamina of OL; AL, antennal lobe; GNG, gnathal ganglia; O, ocelli; mO, medial ocelli; d, dorsal; l, lateral; v, ventral. Scale bars, 100 μm. **f**, Scatter plot showing that the abundance differences per cell cluster between males and gynes are negatively correlated with the same abundance differences between workers and gynes. Each dot represents one of the 43 cell clusters.

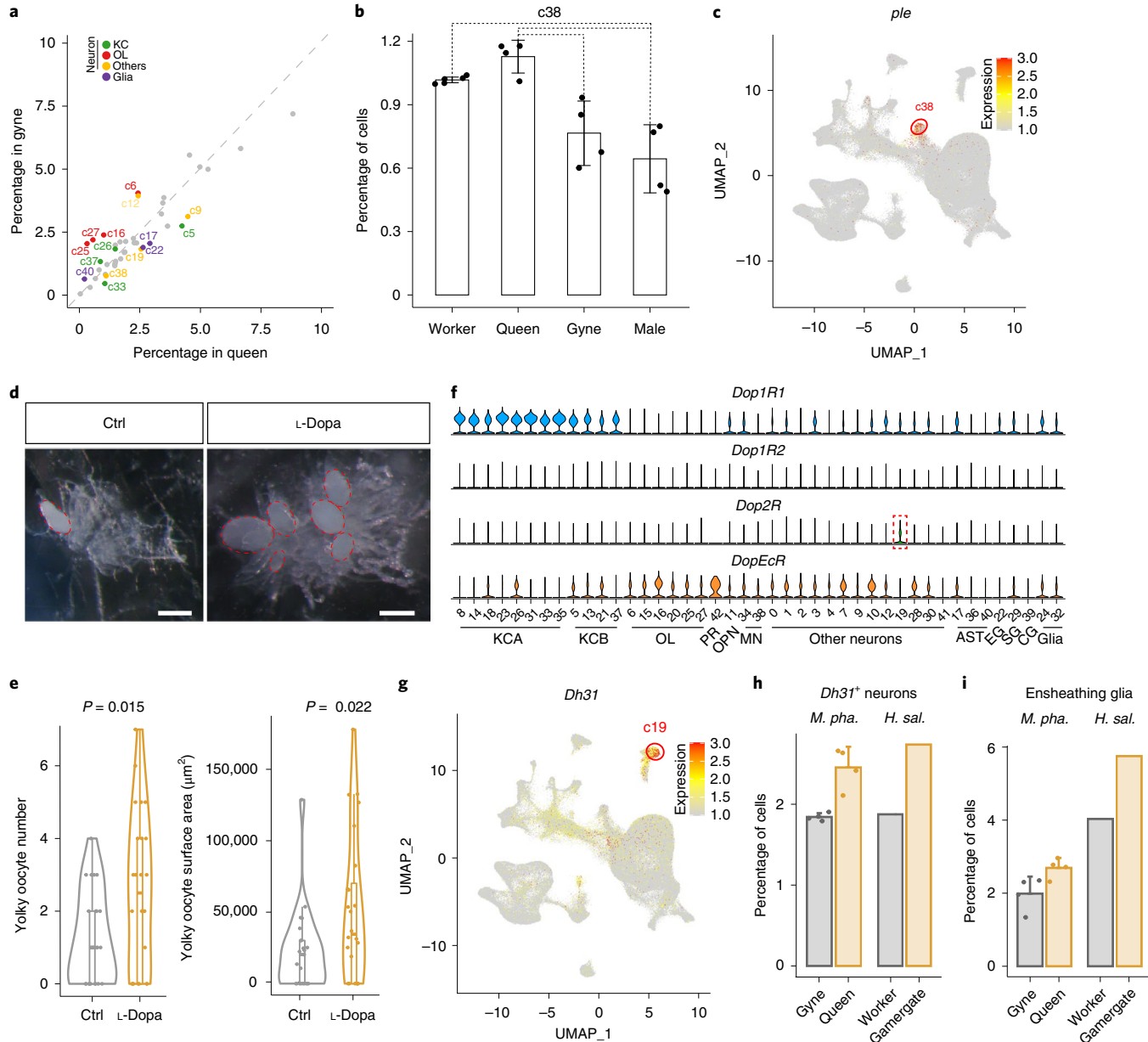

**Fig. 6 | Cell compositional differences between gyne and queen brains in *Monomorium*. a**, Cell clusters that display significant abundance differences, as assessed by scCODA and with >1.3-fold changes, between gyne and queen brains. Coloured dots represent cell clusters with significant abundance differences and grey dots represent those with no significant differences. **b**, The percentage of cells from c38 against total brain cells in each adult phenotype. Each dot represents the value of a phenotype-specific biological replicate ($n = 5$ for workers, 4 for queens, 4 for gynes and 4 for males), bars are means ± s.d. across replicates, and dotted lines indicate the comparison with significant differences in **a**. **c**, Expression level of *ple* across the 43 cell clusters. The UMAP plot is coloured by gene expression (grey is low and red is high) and the red circle indicates the cell cluster that preferentially expressed *ple*. **d**, Representative ovaries of control (Ctrl) and L-dopa treated gynes with yolky oocytes highlighted with red dotted ovals. Scale bar, 200 μm. **e**, Violin plots showing yolky oocyte number and total surface area in L-dopa treated gynes and control groups ($n = 24$ for both groups) with P values obtained from two-sided Student's t-tests. For all box plots inside the kernel density plots, the horizontal thick lines denote median values, the boxes show the range between the 25th and 75th percentiles and the whiskers represent 1.5× the interquartile range. **f**, Expression of the four dopamine receptors across the 43 cell clusters, with the dashed box highlighting the only cell cluster with a preferential expression of *Dop2R*. **g**, Expression level of *Dh31* across the 43 cell clusters, in similar notation to **c**. **h,i**, The convergent increases in relative abundance of *Dh31*+ neurons (**h**) and ensheathing glia (**i**) in *M. pharaonis* (accessed by scCODA and with >1.3-fold change) and *H. saltator* (accessed by Fisher's exact test with FDR < 0.001 and >1.3-fold change) in reproductively active females compared with uninseminated females. Each dot represents the value of a phenotype-specific biological replicate ($n = 4$ for gynes and queens) and bars are means ± s.d. across replicates in *M. pharaonis*.

and/or programmed cell death might occur during this role differentiation process in adult ant reproductives. However, given that the queens (3–6 months old posteclosion) were much older than the gynes (5–10 days old posteclosion), some of these cellular changes could also reflect age rather than effects induced by insemination.

Among the cell clusters with increased abundance in queens, c38 stands out as it preferentially expressed *ple* (encoding the

rate-limiting enzyme in dopamine synthesis[54]) and *DAT* (a dopamine transporter[55]), indicating that c38 primarily represents the dopaminergic neurons (Fig. 6b,c). The abundance increase of c38 in queens might therefore indicate an elevated activity of the dopamine circuit triggered by insemination. Previous studies of several ponerine ant species have found that dopamine titres are positively correlated with increased ovarian activity in reproductive workers (gamergates)[56,57]. We therefore tested whether dopamine administration would induce accelerated oocyte growth in *M. pharaonis* gynes by feeding 5-day-old gynes with 30 mg ml⁻¹ ʟ-dopa in 10% sucrose for 5 days. Compared to the control group, ʟ-dopa treated gynes had more yolky oocytes and the total areas of these yolky oocytes were significantly enlarged (Fig. 6d,e), confirming that dopamine has a gonadotrophic function during or immediately following the gyne–queen transition induced by insemination.

The effects of dopamine depend on the downstream neurons that express the dopamine receptors[58]. The *M. pharaonis* genome encodes *Dop1R1*, *Dop1R2*, *Dop2R* and *DopEcR* similar to *Drosophila* (Extended Data Fig. 8a). We found that *Dop2R* was preferentially expressed in the c19 neurons (Fig. 6f), the abundance of which was significantly increased in queens compared to gynes (Fig. 6a). This cell cluster is also characterized by the preferential expression of *Dh31*, *Prohormone-2*, *amon* and *7B2*, indicating that this dopamine-regulated cell cluster comprises a population of peptidergic neurons that mainly produce diuretic hormone (Fig. 6g). A recent study showed that neuronal knockdown of *Dh31* led to a statistically significant decrease in egg laying in *Drosophila*[59], indicating a potential role of this neuropeptide in ovulation. Overall, these results indicate that dopamine probably mediates insemination-induced gonadotrophic functions via the c19 *Dh31*-expressing neurons in *M. pharaonis*.

**Convergent cellular changes in reproductive role transitions.** While the gyne–queen transition in *M. pharaonis* represents a classic form of reproductive role differentiation[21], several ant species such as *H. saltator* have secondarily evolved reproductive role differentiation within the worker caste[60–62]. It was therefore interesting to observe that the abundances of the *Dh31*-expressing neurons and the ensheathing glia were increased both in gamergates compared to workers of *H. saltator* and in queens relative to gynes of *M. pharaonis* (Fig. 6h,i, Extended Data Fig. 8b and Supplementary Data 7). This reminds that gamergates have co-opted the reproductive role differentiation gene regulatory network normally expressed in the gyne–queen transition[21]. In particular, the ensheathing glia, which play a neuroprotective role in adult *Drosophila* brains[63], had the highest abundance in queens among the four adult phenotypes of *M. pharaonis* (Fig. 5d), suggesting a role in queen longevity. Ageing-associated decline of ensheathing glia has recently been reported in fruit flies and *H. saltator* workers, while the *H. saltator* gamergates are resistant to ageing with their ensheathing glia declining at a much slower rate[18]. Our finding of increased abundance of ensheathing glia in mature queens thus appears to corroborate the critical role of ensheathing glia for longevity in the reproductive castes of ants.

## Discussion

Our study generated a superorganismal brain cell atlas by profiling all brain cells of the full panel of adult phenotypes that typically make up an ant colony. We found that the ant mushroom body KCs are abundant and transcriptionally diverse relative to the KCs of *Drosophila*. We also identified conserved OL neurons that probably play crucial roles in visual courtship behaviour and circadian rhythm regulation in ants. Our results are consistent with advanced brain-level division of labour in superorganismal colonies and shed new light on neural mechanisms associated with the lifespan differences between workers and queens.

**Functional integration of superorganismal brains.** As we outlined at the start, we expected that the major evolutionary transition to superorganismal colony organization in ancestral ants should have selected for specialization of neural circuitry rather than bigger brains per se. Our study provides direct evidence to support this hypothesis, with high degrees of specialization being detectable in the brain cellular composition of all four adult phenotypes of *M. pharaonis* ants. We found that 41 out of 43 cell types could be detected in all four brain phenotypes, albeit in different abundances. In particular, workers and males have evolved extreme forms of brain specialization and with almost opposite cell-type preferences. Worker brains had the most abundant KCs and OPNs and the least abundant OL neurons, all biases that were opposite in male brains. These cellular differences were consistent with anatomical brain structures reflecting the distinct social and sexual specialization in these two phenotypes. Males are extremely short-lived and do not take part in any colony maintenance tasks, as their only function is to find and inseminate a virgin queen. Ant males therefore function as 'simple minded' but extremely targeted sperm vectors[7]. In sharp contrast, workers engage in all the colony tasks except reproduction and need multipurpose brains, consistent with the KCs and OPNs in workers being biased for processing complex information associated with nursing, foraging, colony defence and social communication.

Relative to these extremes, gynes and queens had intermediate abundances for almost all brain cell types. This probably reflects that both gynes and queens have maintained functional brain repertoires for a large subset of the social tasks normally done in more advanced ways by workers. Many ants may have retained generalist queen brain functions because they have solitary lives during colony founding, so they need to nurse a first brood and in some species even to forage[20]. However, finding relatively generalist brains in *M. pharaonis* gynes and queens is remarkable because this species has lost that ancestral independent colony founding behaviour and never needs to operate without worker assistance. However, *Monomorium* colonies have very many queens, some of which may fail to become inseminated. Such failed queens are known to survive, albeit for less time than inseminated queens, and perform worker-like behaviours[21], which may have selected for the maintenance of general cognitive abilities in the gyne/queen caste.

Overall, our results confirm the concept of complementary divergence in brain function between superorganismal colony members and strongly suggest that fine-tuned brain-level division of labour is an integrated part of sex, caste and reproductive role differentiation, in ways that are not expected to evolve in social systems where a variable number of colony members retain breeder potential even though they may first have helper roles. In many ways, the separate individual brains in colonies of ants such as *M. pharaonis* combine into a modularly coordinated super-neural organization maintained by advanced communication between colony members. Individual brains are continuously turned over when adult colony members hatch and die, but functional homeostasis and balanced interactions between modules continue, similar to how cells in a metazoan body are turned over without compromising overall body health, consistent with hypothetical comparisons by Wheeler more than a century ago[5]. The complementary functions of individual brains across the full panel of adult phenotypes are consistent with natural selection maximizing colony-level fitness, as expected for all superorganismal social insects, but not for animals that form societies without irreversible caste differentiation for life among all colony members[4,5,64].

**Neural effects on longevity/fecundity evoked by insemination.** Gynes and queens represent two subsequent functional states of the same reproductive female caste, separated by a single insemination event that induces substantial brain transcriptome remodelling resulting in remarkable shifts in behaviour[21]. The gene

regulatory network that mediates this reproductive role differentiation is insemination-specific rather than queen-specific, because it has been co-opted by distantly related ant species that secondarily shifted to reproduction via worker–gamergates rather than queens[21]. In the present study, we further explored this convergent evolutionary scenario across castes at the brain cell level. We found that there are parallel cellular shifts across these two caste-specific reproductive role transitions induced by insemination. In particular, a cluster of ensheathing glia with neuroprotective and anti-ageing functions was expanded in both *M. pharaonis* queens and *H. saltator* gamergates[18]. We therefore speculate that ensheathing glia modification might represent one of the proximate mechanisms that ancestrally prolonged queen longevity in ants and whose co-option secondarily extended worker lifespan when they became inseminated as gamergate reproductives. This quantitative reinforcement mechanism of particular neural modules in adulthood effectively decouples queen and worker ageing, so that extremely divergent caste-specific lifespans could evolve[65].

Insemination also induced the expansion of dopamine neurons and a cluster of downstream *Dop2R* expressing neurons in *M. pharaonis* queens, and the counterpart cell cluster in *H. saltator* was found to be convergently expanded in gamergates as well. Our experimental confirmation of the gonadotrophic function of dopamine via feeding *M. pharaonis* gynes with L-dopa suggests that dormant ovary maturation in gynes may be switched into an accelerating trajectory by elevating functionality of dopamine neurons. The downstream *Dop2R* neurons also preferentially expressed *Dh31*, a diuretic hormone known to regulate ovulation in flies[59]. The simultaneously expanded dopamine neurons and downstream *Dop2R* neurons may thus constitute an integral and conserved neural module to realize enhanced reproductive potential in ant queens well beyond the normal fertility levels of solitary insects.

## Methods

**Biological samples.** The original colony of *M. pharaonis* was collected in 2016 from a resident house in Mengla, Xishuangbanna, Yunnan Province, China, and split into hundreds of subcolonies in the laboratory in the subsequent years. All colonies were reared at 27 °C, 65% RH and a 12/12 h light/dark cycle. The rearing of gynes and males was induced in newly split colonies where inseminated and egg-laying queens were removed, and where easily recognizable male pupae were continuously removed to prevent the newly hatching gynes became inseminated. The eclosion date of males and gynes were recorded. The queens were collected from stable, mature colonies in which they were actively laying eggs. The demographic states of the colonies were frequently surveyed so the ages of queens could be estimated, albeit less accurately than the gynes and males. Workers were randomly collected from colonies, both inside and outside nests, so these samples covered both young (nursing) and old (foraging) workers. At the moment of dissection, gynes were 5–10 days posteclosion, queens were 3–6 months posteclosion and males were 3–14 days posteclosion, while the age of workers was not recorded. Four to five biological replicates were prepared for snRNA-seq for each of the four adult phenotypes (five for workers, four for queens, four for gynes and four for males). Nuclei for each single replicate were isolated from a pool of 30 to 50 whole brains for each specific category of adult phenotype.

**Brain dissection, nuclei isolation and snRNA-seq.** Ants were anaesthetized in a dissection dish on ice and washed with ethanol and PBS twice, after which brains were dissected in PBS on ice under a stereomicroscope (Nikon, SMZ645). We carefully removed the surrounding trachea (always present) and ocelli (absent in workers) after which ant brains were washed with 1 ml PBS to which 1 U μl⁻¹ RNase inhibitor was added. All brain samples were collected during the daytime (9:00 to 16:00).

The single nuclei were prepared by mechanical extraction. Specifically, for a single replicate of a specific adult phenotype, 30 to 50 whole brains were pooled and infiltrated together with lysis buffer (10 mM Tris-HCl, pH 7.4, 10 mM NaCl, 3 mM MgCl₂, 0.1% IGEPAL CA-630, 1% BSA, 1× protease inhibitor and 1 U μl⁻¹ RNase inhibitor) for 5 min, followed by being pestled with a 2-ml Dounce homogenizer set. Then the nuclei were filtered through a 30-μm cell strainer (Sysmex CellTrics) and pelleted by centrifugation at 3,000g for 8 min. The nuclei were then resuspended (buffer 1% BSA, 2 U μl⁻¹ RNase inhibitor and 6% ficoll in PBS) at a concentration of 1,000 nuclei per μl for single-nucleus library preparation.

The DNBelab C Series Single-Cell Library Prep Set (MGI Tech Co.) was used for the preparation of snRNA-seq libraries according to the manufacturer's

protocol (MGI Tech Co., Ltd). In brief, the single-nucleus suspensions were converted to barcoded snRNA-seq libraries through steps including droplet encapsulation, emulsion breakage, messenger RNA captured bead collection, reverse transcription, complementary DNA amplification and purification. Indexed sequencing libraries were constructed according to the manufacturer's protocol. The libraries were quantified by Qubit single-strand DNA Assay Kit (Invitrogen) and paired-end sequenced on the DNBSEQ platform at China National GeneBank (Shenzhen, China). Read 1 was 30-bp in length and contained a 10-bp cell barcode 1, a 10-bp cell barcode 2 and a 10 bp unique molecular identifier (UMI). Read 2 was 100-bp in length and represented the transcript sequence.

**snRNA-seq data processing and gene expression quantification.** Before read alignment, Drop-seq_tools (v.1.13)[66] was used to trim ploy(A) stretches, add cell and UMI barcodes to the reads, and remove the reads with barcodes that contained low-quality bases. The reads that passed quality control were then aligned to the *M. pharaonis* reference genome produced by Gao et al.[67] using STAR (v.2.6.1a_08-27)[68] with default parameters. To further ensure the accuracy of alignment, we used an in-house script to discard the reads that could be aligned to multiple positions of the reference genome, and to discard any spliced reads that spanned a gap >50 kb (because up to 99.5% of the *M. pharaonis* introns are shorter than 50 kb) or that detected splicing sites other than the canonical ones (that is, GT/AG, GC/AG and AT/AC). The TagReadWithGeneExon function of Drop-seq_tools was then used to add gene annotation tags to the aligned reads, and the DigitalExpression function to extract digital gene expression (DGE) data matrices (that is, the number of UMIs per gene per nucleus). Nuclei with fewer than 200 or more than 2,000 expressed genes, or with a high proportion (>1%) of UMI counts derived from mitochondrial genes were discarded.

**Integration, clustering and cell-type annotation.** After obtaining the filtered DGE matrices, we used Seurat (v.3.1.5)[69] for normalization, integration, dimension reduction, clustering, visualization and marker gene analysis in the R (v.3.6) environment. Specifically, the Seurat NormalizeData and FindVariableGene functions were first executed for each of the 17 samples (that is, the five worker replicates, four queen replicates, four gyne replicates and four male replicates), after which the 17 samples were integrated into a single dataset using FindIntergrationAnchors and IntegrateData (parameters dims = 20, anchor. features = 4,000) to correct for batch effects. The integrated dataset was then scaled, followed by dimensionality reduction with the RunPCA function. The first 25 PCs were used to construct a shared nearest neighbour network, and clusters were identified using the Louvain algorithm that was implemented in the Seurat FindClusters function. The resulting clusters were visualized using the UMAP method[70] by the RunUMAP function. Marker genes for each cluster were identified by the FindAllMarkers function with the Wilcoxon Rank Sum test (min.pct = 0.25, logfc.threshold = 0.25, test.use = 'wilcox', only.pos = TRUE).

To choose an appropriate resolution for clustering the 206,367 nuclei, we first generated different clustering versions using a series of resolutions (0.25–3.00 with a step of 0.25) and then manually checked the clustering results to find the versions that could best separate the main neuronal or glial cell types into different clusters. The examined cell types included KCs, OPNs, photoreceptors, monoaminergic neurons, astrocyte-like glia, ensheathing glia, cortex glia and surface glia, which have been well studied in insects with established marker genes as listed in Supplementary Data 1. We also required that all clusters in versions assessed as qualified contained nuclei from two or more of the 17 samples, to ensure that no cluster could result from batch effects. The combination of these considerations led us to finally choose the version generated by a resolution of 1.5 (see Fig. 1b for the cell-type annotations and Supplementary Data 1 for the number of nuclei per cluster per sample). The assignment of clusters to OL neurons, for which well-established marker genes are not available, was based on the transcriptional similarity of cell clusters between *Monomorium* and *Drosophila* and between *Monomorium* and *Harpegnathos* (see section Transcriptional similarity of cell clusters between species below for more details).

**Correlations between snRNA-seq and bulk RNA-seq data.** To evaluate the consistency of the gene expression quantification results obtained by snRNA-seq and conventional bulk RNA-seq (Extended Data Fig. 1e), we first generated pseudo-bulk data for each adult phenotype with the snRNA-seq data by accumulating the UMI counts by gene from all nuclei belonging to the same adult phenotype, after which the expression level of a given gene was calculated as CP10K (UMI counts per 10,000). Bulk whole brain RNA-seq reads of the four *M. pharaonis* adult phenotypes were retrieved from Wang et al.[6]. SOAPnuke (v.2.1.0)[71] was first used to assess sequencing quality and filter reads of low quality with parameters (-G -l 20 -q 0.2 -E 60 -5 1 -Q 2). Clean reads were then aligned to the *M. pharaonis* reference genome produced by Gao et al.[67] using Hisat2 (v.2.1.0)[72,73] with default parameters. Read count matrices (that is, the number of uniquely mapped reads per gene per sample) were then obtained by an in-house script. After accumulating read counts by gene from all replicates belonging to the same adult phenotype, the expression level of a given gene was calculated as CP10K (read counts per 10,000). Finally, the correlations between gene expression as quantified by the snRNA-seq data (pseudo-bulk) and the bulk RNA-seq data

were obtained as Pearson correlation coefficients after removing genes that had CP10K values <1 in both datasets and transforming the expression values to $\log_2$ [CP10K+1].

**Principal component analysis (PCA) and hierarchical clustering analyses of cell clusters.** For PCA presented in Extended Data Fig. 2c and hierarchical clustering analyses presented in Fig. 3a and Extended Data Fig. 4b,c, we generated a matrix representing the expression level of each gene in each cell cluster. These expression levels were calculated as $\frac{\sum_{i=1}^{n} u_i}{n} \times 10^6$, where $u_i$ is the percentage of UMI counts of a focal gene in each nucleus within the cell cluster, and $n$ is the total nuclei number of a focal cluster. In Extended Data Fig. 2c, we first filtered the genes with narrow variance (standard deviation of expression level <1 across the 43 cell clusters) and then performed a variance stabilizing transformation with the *vst* function provided by DESeq2 (v.1.22.0)[74]. We finally generated the PCA plot with the *vst*-transformed matrix using the plotPCA function provided by DESeq2. In Fig. 3a and Extended Data Fig. 4b,c, only the top 8,000 highly expressed genes across all KC clusters were used for hierarchical clustering analysis, which was achieved using the R package 'pvclust' (v.2.2.0)[75] with method.dist set to 'cor' and method.hclust set to 'Ward.D'. Confidence levels of branches were estimated by the bootstrapping-based method implemented in pvclust.

**Identification of DEGs between cell clusters.** The FindMarkers function of Seurat (v.3.1.5) was used to identify DEGs between two (or two groups of) cell clusters with the MAST model[76]. *P* values were adjusted for false discovery rate (FDR) following the Benjamini–Hochberg procedure[77]. A gene was retained as a significant DEG when reporting an FDR < 0.05, showing an expression fold change >1.25 and being expressed in >20% of cells in the up-regulated cell cluster.

**Orthologue identification.** All analyses involving cross-species comparisons were restricted to one-to-one orthologues between species, built with the reciprocal best hit approach according to the bit scores obtained from all-versus-all BLASTP (blast-2.2.26) alignment with parameters (-F F -e $1 \times 10^{-5}$).

Apart from *M. pharaonis*, other species used in the orthologue identification were *H. saltator*, *D. melanogaster*, *A. mellifica*, *Caenorhabditis elegans* and *Homo sapiens*. The gene sets of *H. saltator* (GCA_003227715.1_Hsal_v8.5) and *A. mellifica* (AJ489744) were downloaded from the National Center for Biotechnology Information (NCBI), and the gene sets of *D. melanogaster*, *C. elegans* and *H. sapiens* were obtained from Ensembl (release-100).

**Gene ontology (GO) annotations and enrichment analyses.** GO of the *M. pharaonis* protein-coding genes was assigned according to the GO annotation of their orthologues in *D. melanogaster*, *C. elegans* and *H. sapiens* obtained from the Ensembl database (release-100). One-to-one orthologues between *M. pharaonis* and *D. melanogaster*/*C. elegans*/*H. sapiens* were built as mentioned above. The GO annotation was assigned on the basis of the priority of *D. melanogaster* > *C. elegans* > *H. sapiens* when a *M. pharaonis* gene could find an orthologue in more than one species (that is according to the evolutionary distance against *M. pharaonis* from close to remote). The *M. pharaonis* genes that could not be annotated by the orthologous method were further aligned to the UniProt database (release-2020_04) using BLASTP with parameters (-F F -e $1 \times 10^{-5}$). The best hit of each query gene was then retained, on the basis of its BLASTP bit score, and the GO annotations of that best hit was assigned to the query gene. The combination of these two methods allowed us to assign GO annotation to 76% of the *M. pharaonis* protein-coding genes, which is considerably higher than the 56% reported by Gao et al.[67].

Fisher's exact tests were used to examine whether the up-regulated DEGs in a focal cell cluster (or a group of clusters) were significantly enriched in a specific GO term in relation to the background genes. This procedure compared the number of up-regulated DEGs annotated to this GO term, the number of up-regulated DEGs not annotated to this GO term, the number of background genes annotated to this GO term and the number of background genes not annotated to this GO term. The background genes were defined as all genes except for the up-regulated DEGs with mean expression level (CP10K) > 1 across the cell clusters of interest (for example, the 12 KC clusters in Fig. 3e or the six OL clusters in Fig. 4b). *P* values were adjusted by FDR following the Benjamini–Hochberg procedure[77], and GO terms with FDR < 0.05 and gene number ≥2 were considered to be significantly enriched.

**Single-cell datasets of other insects.** The single-cell datasets with cell-type annotation and gene expression information of each cell for the female and male whole brains of adult *D. melanogaster*[24], the female and male heads of adult *D. melanogaster*[36], the worker and gamergate midbrains of adult *H. saltator*[18] and the OLs of *D. melanogaster* (the main dataset from Kurmangaliyev et al.[41] and the adult dataset from Özel et al.[42]) were obtained from the original publications. Manual checks for possible mis-annotation of the cell clusters were performed for the *D. melanogaster* whole brain and *H. saltator* midbrain datasets before subsequent analyses (see Supplementary Data 2 for the cell numbers and final annotations for each cluster in each dataset). For the single-cell datasets of adult *D. melanogaster* midbrains[27] as well as the whole brain and mushroom bodies of adult *A. mellifera* workers[35], we obtained the digital-expression matrices and

performed clustering analysis with Seurat following the parameters mentioned in the original publications (see Extended Data Figs. 3 and 4c for the clustering results; see Supplementary Data 2 for the cell numbers and annotations for each cluster). The initial clustering result of the *D. melanogaster* midbrain data could not separate glial subtypes into independent clusters except for astrocytes, so we reclustered all glial cells using a resolution of 1.1 and identified cell clusters of astrocytes, surface glia and cortex glia (Extended Data Fig. 3c,d). However, ensheathing glia were probably absent in this dataset[27].

**Transcriptional similarity of cell clusters between species.** MetaNeighbor[25] was used to assess the pairwise transcriptional similarity of cell clusters between two species (for example, between *Monomorium* and *Drosophila* or between *Monomorium* and *Harpegnathos* as presented in Extended Data Fig. 2d,e). The MetaNeighbor framework calculates the correlations between all pairs of cells within and between datasets on the basis of the expression of a set of genes, and produces a score of mean AUROC to quantify the similarity of cell-cluster pairs[25]. To improve the performance of MetaNeighbor, we used pseudo-cell rather than single-cell expression as suggested in previous studies[78–80]. This approach reduces the impact of data sparsity: a typical feature of high-throughput single-cell sequencing data. A pseudo-cell was generated by merging the data of ten cells that were randomly picked from the total cells within a cell cluster without replacement. The UMI counts from these ten cells were then summed by gene, after which gene expression levels were calculated as CP10K (UMI counts per 10,000) in a pseudo-cell. The pseudo-cell expression matrix of each species was then standardized by z-transformation as $z = (x - m)/\text{s.d.}$, where $x$ is the expression level of a focal gene in a pseudo-cell, and $m$ and s.d. are the mean and the standard deviation of expression levels for the focal gene across all pseudo-cells. To maximize the differences between cell clusters, the orthologous genes that were identified as cell-cluster markers in at least one of the two compared species were used for the MetaNeighbor analysis. The correlations between pseudo-cell pairs were calculated as Pearson's coefficients. Pairwise AUROC scores for all cell clusters within and between species were visualized as a heatmap generated by the heatmap.2 function of the R package 'gplots', and the dendrogram of the cell clusters was generated by hierarchical clustering using the Ward's minimum variance method with the distance defined as 1-AUROC (Extended Data Fig. 2d,e).

By comparing all *Monomorium* cell clusters with all cell clusters defined in adult *Drosophila* whole brains (with OL cell types)[24] and in adult *Harpegnathos* midbrains (without OL cell types)[18] using the MetaNeighbor framework, we were also able to identify six *Monomorium* cell clusters that were derived from the OLs, because they clearly grouped with the *Drosophila* OL cell clusters in the *Monomorium*-versus-*Drosophila* tree and formed a single clade in the *Monomorium*-versus-*Harpegnathos* tree (Extended Data Fig. 2d,e).

**Interspecies comparisons of KCs and OL neurons.** To track the evolutionary origin of the *Monomorium* KC subtypes, we collected the KC clusters from two additional hymenopteran insects—the ant *H. saltator* that shares a common superorganismal ancestor with *Monomorium*[18] and the honeybee *A. mellifera* that belongs to the corbiculate bee lineage that independently evolved superorganismal colonies[35]. We also collected the *D. melanogaster* γ, α'/β' and α/β KCs from three independent studies[24,27,36] (Supplementary Data 2 and Extended Data Figs. 3 and 4). We then assessed the transcriptional similarity of KC clusters between any two of the four species (that is *M. pharaonis*, *H. saltator*, *A. mellifera* and *D. melanogaster*) with the MetaNeighbor framework as described above, except for limiting the pairwise AUROC calculation to KC clusters instead of all cell clusters and using the set of DEGs identified among the KC clusters to maximize the differences between KC subtypes. Pairwise similarities measured as AUROC scores for the focal cell clusters between species were visualized as heatmaps, which were generated with the R package 'ggplot2' (Extended Data Fig. 5a–d). We also used network plots to visualize the correspondence of KC clusters across species (Fig. 3f). These network plots were generated by Cytoscape (v.3.8.2)[81], with the clusters of *Monomorium* as source nodes, clusters of other species as target nodes and AUROC scores as edges. Only edges with AUROC >0.80 were shown. Each edge linked a *Monomorium* KC cluster to its top hit among the KC clusters in another species according to AUROC scores. A second hit also was plotted when the difference between the top and second AUROC score was less than 0.05.

To improve the annotation of the *Monomorium* OL neurons, we compared the *Monomorium* OL clusters with the *D. melanogaster* OL cell clusters generated by two independent single-cell studies that focused exclusively on the *D. melanogaster* OLs[41,42] (Supplementary Data 2), based on the MetaNeighbor framework as in the KC subtype analyses. The putative cell-type identity for the *Monomorium* OL clusters were summarized according to these two comparisons, which were generally consistent with each other (Extended Data Fig. 6a–c) and made us assign one *Monomorium* cluster (c20) as LMCs, three (c25, c6 and c27) as medulla neurons, one (c16) as lobula plate T4/T5 neurons and one (c15) as lobula columnar cells (Fig. 4a).

**Differential abundance testing between phenotypes and sexes.** scCODA (v.0.1.6)[52], a Bayesian model based on hierarchical Dirichlet-multinomial distribution, was used to identify cell clusters with credible abundance differences

between any two of the four *Monomorium* adult phenotypes when also taking the variation of the biological replicates into consideration. There are three important parameters to be considered when using scCODA. The first is the FDR level. In practice, an FDR level of 0.2 is deemed to be acceptable by the authors according to their applications of scCODA in five different real single-cell datasets[52]. The second is the Hamiltonian Monte Carlo (HMC) chain length, which is usually set according to the number of cell clusters. An HMC chain length of 800,000 with a burn-in of 10,000 was sufficient for our *Monomorium* dataset that contained 43 cell clusters. The third is the reference cell type, which is assumed to be unchanged in abundance across different samples. scCODA can automatically select an appropriate cell type as the reference or uses a prespecified reference cell type to identify compositional changes for the remaining cell types. As we did not have any previous knowledge about the best reference cell type for the four *Monomorium* adult phenotypes, the 'automatic reference selection' option was chosen. Finally, a cell cluster with the scCODA-inferred 'Final parameter' other than zero following the scCODA manual, and with >1.3-fold change in relative abundance, was considered to be showing a significant difference in abundance between two adult phenotypes (Fig. 5b,c and Supplementary Data 5).

The adult *D. melanogaster* whole brain dataset that contained five replicates for each sex[24] and the adult *D. melanogaster* head dataset that contained six replicates for each sex[36] (see Supplementary Data 2 for the number of cells per cluster per replicate) were used for the assessment of cell compositional differences between sexes in adult *Drosophila* flies. The same parameters as mentioned above for *Monomorium* were applied, which reported zero cell clusters as significant between female and male *Drosophila* flies in both datasets (Fig. 5b,c and Supplementary Data 5).

**Differential abundance testing between species.** The scCODA framework was also used to assess the cell compositional differences between the whole brains of adult *M. pharaonis* and adult *D. melanogaster* as presented in Fig. 2a. The examined cell types were KC, OPN, monoaminergic neuron, astrocyte, ensheathing glia, cortex glia and surface glia when the remaining cells were assigned as 'others'. Considering that the *M. pharaonis* and *D. melanogaster* datasets were obtained by two different protocols, which might have sampling biases for different cell types, we raised the criteria for defining cell types with credible abundance differences. Specifically, the scCODA assessment was conducted as comparing the four *M. pharaonis* adult phenotypes with the two *D. melanogaster* sexes (see Supplementary Data 4 for data used for scCODA analysis) using an FDR level of 0.1, an HMC chain length of 800,000 with a burn-in of 10,000 and the 'others' category as the reference cell type. Then a cell type with the scCODA-inferred 'Final parameter' other than zero and more than twofold change in relative abundance was considered to be sufficient for showing a significant abundance difference between species.

**Volumetric analysis of brain neuropils.** Volumetric analyses of major brain neuropils were performed for the four representative adult phenotypes of *M. pharaonis*. Confocal image stacks were used to reconstruct the brain neuropils, which were obtained from whole head preparations to preserve the orientation of the brain in the head capsule. The preparation and imaging of the whole head samples were adapted from Smolla et al.[82]. The bleached head samples were imaged using a point-scanning confocal and multiphoton microscope (SP5-X MP, Leica Microsystems) with a ×20 objective. The microscope images were imported into the AMIRA (v.6.4) software, where the main brain compartments (mushroom body, ocellus, antennal lobe, gnathal ganglion) were located, labelled and reconstructed in 3D (Fig. 5e). The volume data of the brain compartments were exported to a Microsoft Excel table (Supplementary Data 6), after which the relative volumes of brain compartments were calculated by dividing compartment volumes by entire brain volumes (Extended Data Fig. 7).

**ISH with tyramide signal amplification.** Anti-sense probes were synthesized to detect the mRNA of *Pka-C1* (XM_012678196) and *GABA-B-R3* (XM_012679823). The lengths of the probes were 500–600 nt. The following primers were used to obtain PCR amplification products from *M. pharaonis* brain cDNA:

Pka-C1_F: CGTTTCTCGTGTCGTTGCG
Pka-C1_T7R: GGATCCTAATACGACTCACTATAGGTGTGGCCCTTGATGTCGTTT
GABA-B-R3_F: TGAATAATACAGGCGTTGCG
GABA-B-R3_T7R: GGATCCTAATACGACTCACTATAGGTATGCTTTTGTGCTTGCGA

The PCR amplification products were purified with the TIANgel midi purification kit (Tiangen DP209) and used as DNA template. Anti-sense probes were synthesized using the DIG RNA labelling kit (Roche catalogue no. 11175025910) with a purified DNA template.

ISH with tyramide signal amplification was performed step by step as follows: (1) dissect ant brains in ice-cold PBS (prepared in nuclease free water, unless otherwise indicated) and rinse the brains in PBSTw (0.1% tween 20 in PBS) twice for 5 min. (2) Fix them in 4% PFA (paraformaldehyde, 0.1% DEPC treated) supplemented with 10% DMSO for 20 min at room temperature, followed by a triple PBSTw rinse for 5 min. (3) Rinse with ice-cold methanol for 5 min,

then with 3:1, 1:1, 1:3 methanol:PBSTw for 5 min each, then three times with PBSTw for 5 min. (4) Treat with 10 µg ml⁻¹ proteinse K for 20 min, then stop the reaction with 20 mg ml⁻¹ glycine rinse for 5 min, followed by three PBSTw rinses for 5 min. (5) Fix with 4% PFA (0.1% DEPC) for 20 min at room temperature, followed by four PBSTw rinses for 5 min. (6) Incubate with preheated in situ hybridization solution (ISHS) for 1 h at 55 °C. (7) Dissolve 4 µl (200 ng µl⁻¹) of probe in 200 µl of ISHS, heat at 80 °C for 3 min, then cool on ice for 5 min, then remove 200 µl of prehybridization ISHS and add the probe solution. (8) Incubate the brains with probe solution for 16 h at 55 °C. (9) Rinse them with preheated ISHS, 3:1, 1:1, 1:3 ISHS:PBSTw and PBSTw for 15 min each at 55 °C. (10) Rinse twice with PBSTw for 15 min at room temperature. (11) Incubate with PAT (1% triton X-100, 1% BSA (g ml⁻¹) in PBS) for 1 h at room temperature. (12) Incubate with the mouse anti-DIG primary antibody (1:400) for 2 h at room temperature. (13) Rinse three times with PBSTx (1% triton X-100 in PBS) for 15 min. (14) Incubate with poly-HRP conjugated anti-mouse IgG secondary antibody for 1 h at room temperature. (15) Rinse four times with PBS for 5 min. Thereafter, the tyramide signal amplification steps were conducted by strictly following the protocols provided in the Tyramide SuperBoost Kit (Alexa Fluor 488, poly-HRP conjugated goat anti-mouse IgG (Thermo Fisher B40912)). The stained brains were then imaged with a customized confocal microscope LSCM-1 (CASLIGHT, Suzhou Institute of Biomedical Engineering and Technology, Chinese Academy of Sciences).

**Hybridization chain reaction (HCR).** The *Nlg2* mRNA (XM_012685917) HCR probe set, amplifiers and buffers were commercially purchased from Molecular Instruments, Inc. We generally followed the HCR RNA-fluorescence ISH protocol provided by Molecular Instruments (www.molecularinstruments.com) for whole-mount fruit fly embryos in the HCR experiments, but with modifications for preparation steps of fixed whole-mount ant brains. Specifically, the ant brains were dissected in ice-cold PBS (prepared in nuclease free water), followed by two PBSTw (0.1% tween 20 in PBS) rinses for 5 min. The brains were then fixed in 4% PFA (paraformaldehyde, 0.1% DEPC treated) supplemented with 10% DMSO for 20 min at room temperature, followed by three PBSTw rinses of 5 min. Next, the brains were rinsed by ice-cold methanol for 5 min, then by 3:1, 1:1, 1:3 methanol:PBSTw for 5 min each, followed by two PBSTw rinses for 5 min. The brains were then treated with 5 µg ml⁻¹ proteinse K for 5 min, followed by three PBSTw rinses for 5 min. After that, the brains were fixed in 4% PFA (0.1% DEPC treated) for 20 min at room temperature, followed by five PBSTw rinses for 5 min. Thereafter, the detection and amplification steps were performed following the referenced protocol. We used Alexa Fluor 488 for the detection of *Nlg2*. The stained brains were imaged with customized confocal microscope LSCM-1 (CASLIGHT, Suzhou Institute of Biomedical Engineering and Technology, Chinese Academy of Sciences).

**Dopamine administration.** We dissolved 30 mg ml⁻¹ ʟ-dopa (3,4-di hydroxy-ʟ-phenylalanine, Sigma-Aldrich D9628-5G) and 10% (w/v) sucrose in distilled water with 0.9% HCl (w/v). One piece of Kimwipes paper was pressed to the bottom of a 15-ml conical tube (Falcon) and then soaked with ʟ-dopa solution. Five-day old gynes were introduced into the tube that was subsequently plugged with a cotton ball, pushed into the tube to offer roughly 5 ml of volumetric space to the ants. The ants were transferred to new tubes every 2 days. The ovaries of 10-day old gynes were then dissected for estimation of the number of yolky oocytes and the total surface area of yolky oocytes. Control gynes were collected from the same colony on the same day and were treated the same except that they were fed in 10% sucrose with 0.9% HCl (w/v).

To measure the number of yolky oocytes and the total surface area of yolky oocytes, the dissected ovaries were spread out, exposing all ovarioles, and then imaged with an Oplenic digital camera mounted to a Nikon SMZ800N stereomicroscope. Yolky oocytes are growing oocytes in the process of absorbing nutrients from haemolymph. They appeared as opaque, oval-shaped areas in the images as indicated in Fig. 6d with red dotted ovals. The total surface area of yolky oocytes in an ovary was estimated as the summed area of these ovals. A total of 24 individuals were measured for each group and all the images were analysed using EZ-MET software (x64, v.6.0.7543).

**Convergent cellular changes in *Monomorium* and *Harpegnathos*.** Cell clusters with significant changes in abundance between *Monomorium* gyne and queen brains were identified by the scCODA framework as described above (Fig. 6a). The correspondence of cell clusters between *Monomorium* and *Harpegnathos* was assessed with MetaNeighbor analysis as mentioned above (AUROC score >0.9), followed by manual check of cell-cluster marker genes in both species, after which the corresponding cell clusters in *Harpegnathos* were subjected to examination of abundance change between the worker and gamergate brains (Supplementary Data 7). Fisher's exact tests were used to assess the significance of abundance change for a given cell cluster between *Harpegnathos* gamergates and workers, by comparing the number of gamergate cells belonging to this cluster, the number gamergate cells not belonging to this cluster, the number of worker cells belonging to this cluster and the number of worker cells not belonging to this cluster. The raw *P* values were adjusted for FDR according to the Benjamini–Hochberg procedure,

and the cell clusters with FDR < 0.001 and >1.3-fold changes in relative abundance between gamergate and worker brains were considered to reflect a significant change. Finally, the cell clusters with consistent direction of significant change between *Monomorium* gyne and queen brains and between *Harpegnathos* worker and gamergate brains were considered as evidence for convergent change during reproductive role differentiation in these two distantly related ant species.

**Reporting summary.** Further information on research design is available in the Nature Research Reporting Summary linked to this article.

## Data availability

The raw snRNA-seq data of *M. pharaonis* generated in this study are deposited in NCBI Sequence Read Archive under BioProject accession no. PRJNA833256 and in the CNGB Nucleotide Sequence Archive under accession no. CNP0001472. The reference genome, gene models, functional annotations of protein-coding genes, gene expression matrix (the number of UMIs per gene per nucleus), full marker gene list of each cell cluster and all in-house scripts are deposited in the figshare repository[83].

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

## Acknowledgements

We thank R. Bonasio from University of Pennsylvania for providing the single-cell gene expression data of *H. saltator*. We thank X. Zhan, J. Li, H. Yu and H. Pan from BGI-Shenzhen for technical help. This work was supported by the National Natural Science Foundation of China (grant nos. 31970573 to G.Z. and 31900399 to W.L.), and the Shenzhen Key Laboratory of Single-Cell Omics (ZDSYS20190902093613831 to L.L.). G.Z. was also funded by a Villum Investigator grant (no. 25900) from the Villum Foundation.

## Author contributions

G.Z., Q.L., W.L. and C.L. conceived the study. J.Z. maintained the ant colonies. W.L., T.W., W.D. and W.J. performed sample collection and brain dissection. M.W., C.L., Z.W. and X.W. conducted the snRNA-seq experiments and sequencing. Y.L., M.W., Q.L., L.H. and L.W. performed quality control, integration, clustering and annotation of the snRNA-seq data. Q.L., P.Z., Y.Z., Q.G. and N.X. conducted comparative analyses across species and adult phenotypes. W.L., X.D., X.Z. and W.Z. performed the ISH/HCR and dopamine administration experiments. M.N. and B.H.D. conducted volumetric analysis of brain neuropils. L.L., X.X., H.L., H.Y. and J.W. contributed reagents/materials/computing resources. Q.L. and W.L. wrote the manuscript with the inputs from all authors. G.Z. and J.J.B. revised the manuscript. All authors read and approved the final manuscript.

## Competing interests

The authors declare no competing interests.

## Additional information

**Extended data** are available for this paper at https://doi.org/10.1038/s41559-022-01784-1.

**Correspondence and requests for materials** should be addressed to Chuanyu Liu, Guojie Zhang or Weiwei Liu.

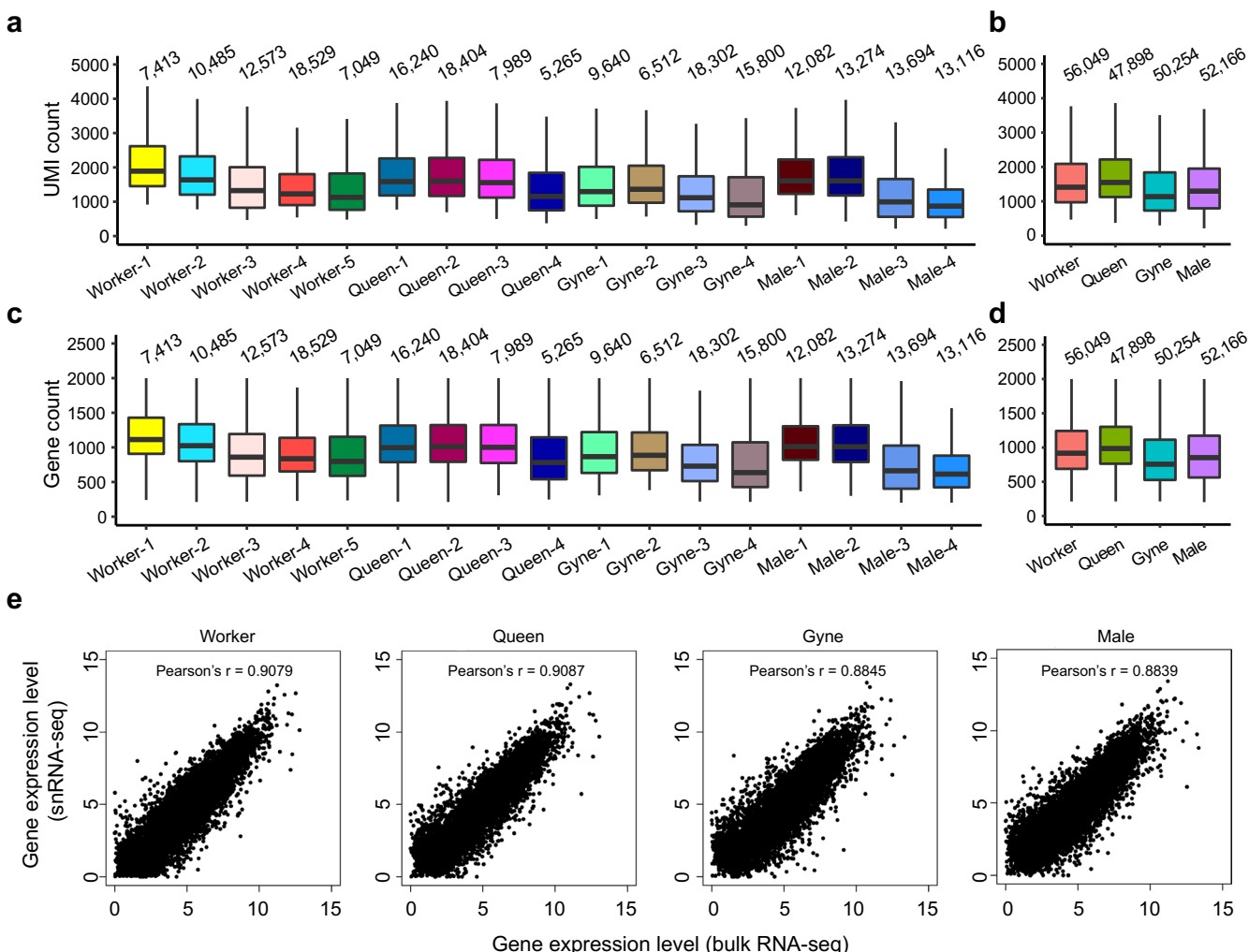

**Extended Data Fig. 1 | Quality control metrics of the *M. pharaonis* single-nucleus RNA-seq datasets. (a-d)** The number of transcripts (**a**, **b**) and genes (**c**, **d**) detected in the nuclei from each phenotype-specific biological replicate (**a**, **c**) and from each adult phenotype after combing biological replicates (**b**, **d**). The number of nuclei per category is shown above each box. For all box plots, the horizontal thick lines denote median values, the boxes show the range between the 25th and 75th percentile, and the whiskers represent 1.5× the interquartile range. (**e**) The correlation of gene expression between bulk RNA-seq data and snRNA-seq data.

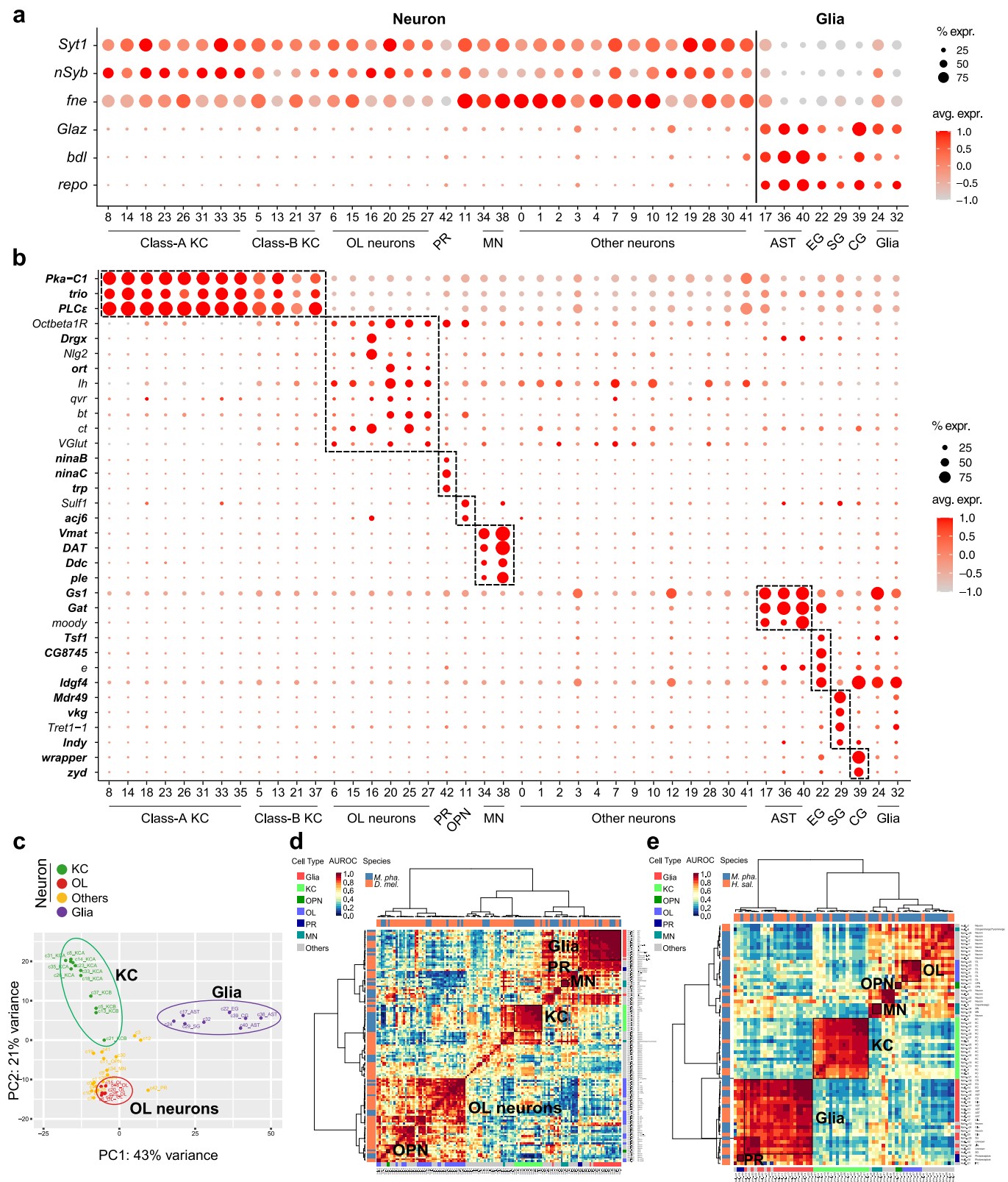

**Extended Data Fig. 2 | See next page for caption.**

**Extended Data Fig. 2 | Classification of the *M. pharaonis* clusters into major cell types.** (**a**) Dot plot showing the expression of neuronal (*Syt1*, *nSyb*, and *fne*) and glial markers (*Glaz*, *bdl*, and *repo*) across the *M. pharaonis* cell clusters. (**b**) Dot plot showing the expression of representative markers that define the major cell types. Gene symbols shown in bold font denotes known markers reported by previous studies in other insect species. Gene symbols shown in regular font denotes novel markers obtained from this study. (**c**) PCA based on average expression profile of each cluster. Clusters are coloured according to cell type category. (**d-e**) Pairwise transcriptional similarity (measured by AUROC scores) of cell clusters from *Monomorium* and *Drosophila* (**d**), and from *Monomorium* and *Harpegnathos* (**e**). The cell-cluster dendrogram trees were generated by hierarchical clustering using the Ward's minimum variance method with the distance defined as 1-AUROC. KC: Kenyon cell; OL: optic lobe; PR: photoreceptor; OPN: olfactory projection neuron; MN: monoaminergic neuron; AST: astrocyte; EG: ensheathing glia; CG: cortex glia; SG: surface glia.

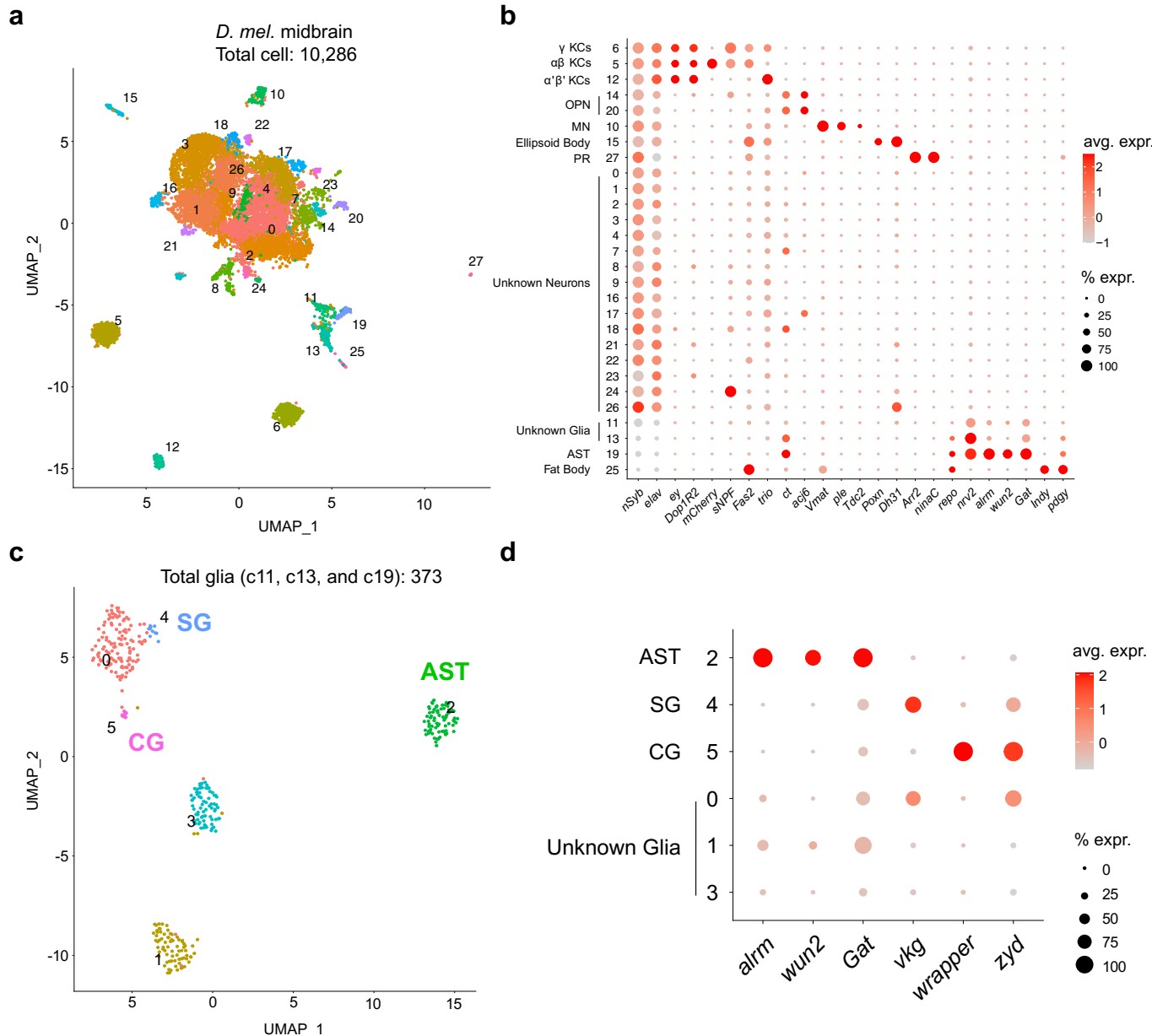

**Extended Data Fig. 3 | Re-analysis of the *Drosophila* midbrain single-cell dataset.** (**a**) UMAP plot showing the clustering result of 10,286 cells from Croset *et al*[27], which are grouped into 28 clusters. Each dot represents one cell and dots are colored according to cluster identity. (**b**) Dot plot showing the expression of representative markers that define the known cell types in *Drosophila* brains. (**c**) Re-clustering of the glial clusters (c11, c13 and c19) identified three known glial subtypes. (**d**) Dot plot showing the expression of representative markers that define the known glial subtypes. KC: Kenyon cell; OPN: olfactory projection neuron; MN: monoaminergic neuron; PR: photoreceptor; AST: astrocyte; CG: cortex glia; SG: surface glia.

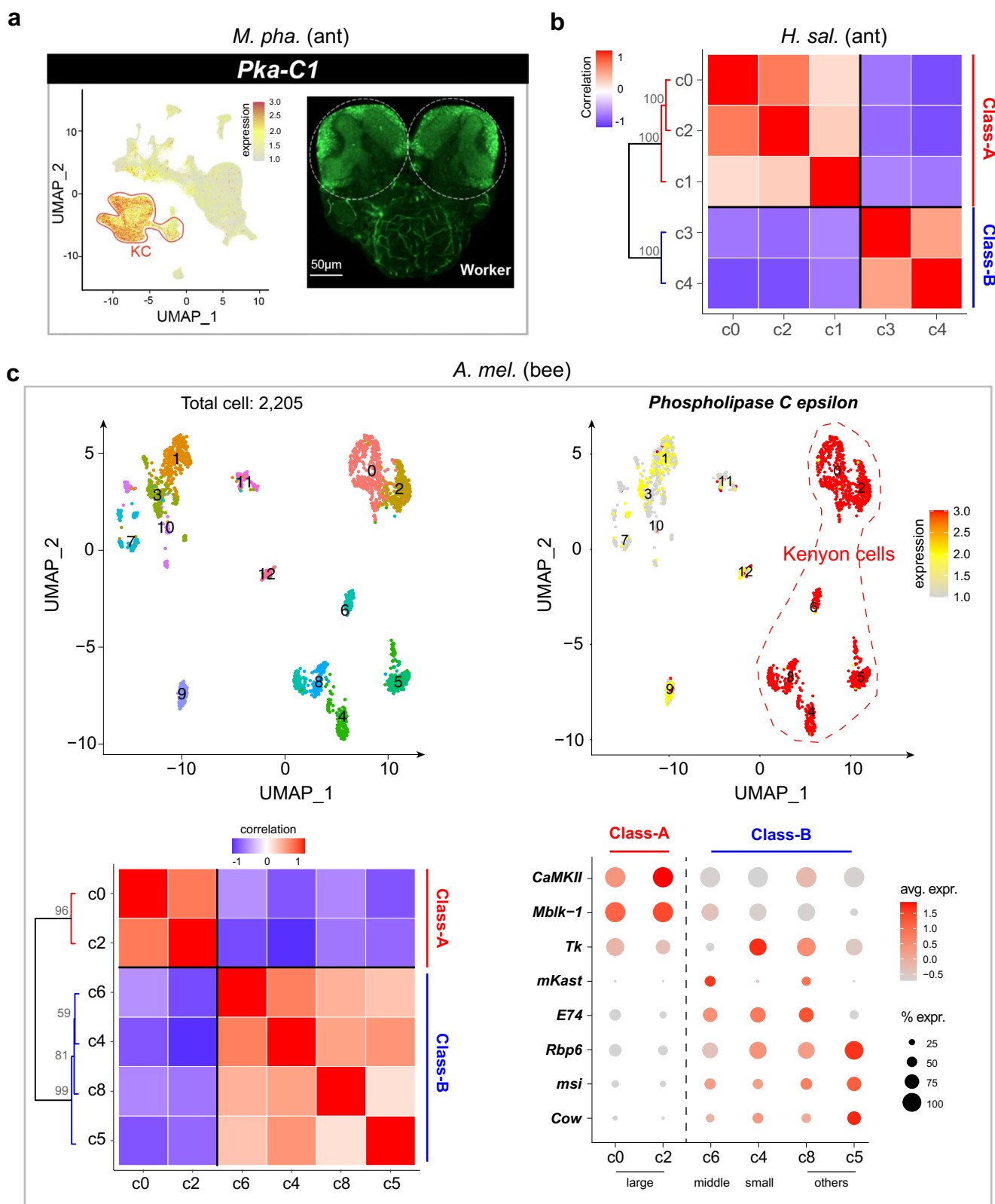

**Extended Data Fig. 4 | See next page for caption.**

**Extended Data Fig. 4 | Kenyon cells in three hymenopteran insects.** (**a**) Expression of the KC marker *Pka-C1* across all cell clusters (left) and whole-mount RNA detection of *Pka-C1* by *in situ* hybridization (right) in a *M. pharaonis* worker brain. UMAP plot is colored by gene expression (grey is low and red is high) with red solid line indicating the KC clusters. White dotted circles indicate paired mushroom bodies with strong hybridization signal. (**b**) Pairwise Pearson correlations and hierarchical clustering of the *H. saltator* KC clusters based on gene expression, showing a clear division into two major classes. The gray numbers at the branches are confidence values based on bootstrap method. (**c**) Re-analysis of the *A. mellifera* single-cell dataset. Top left: UMAP plot showing the clustering result of the 2,205 cells from Traniello *et al*[35], which are grouped into 13 clusters. Top right: Expression of the honeybee KC marker *Phospholipase C epsilon* (*PLCε*) across the 2,205 cells. Dashed line indicates the six clusters that preferentially expressed *PLCε*. Bottom left: Pairwise Pearson correlations and hierarchical clustering of the six honeybee KC clusters in similar notation as panel **b**. Bottom right: Dot plot showing the expression of representative markers that define the known KC subtypes in honeybee brains.

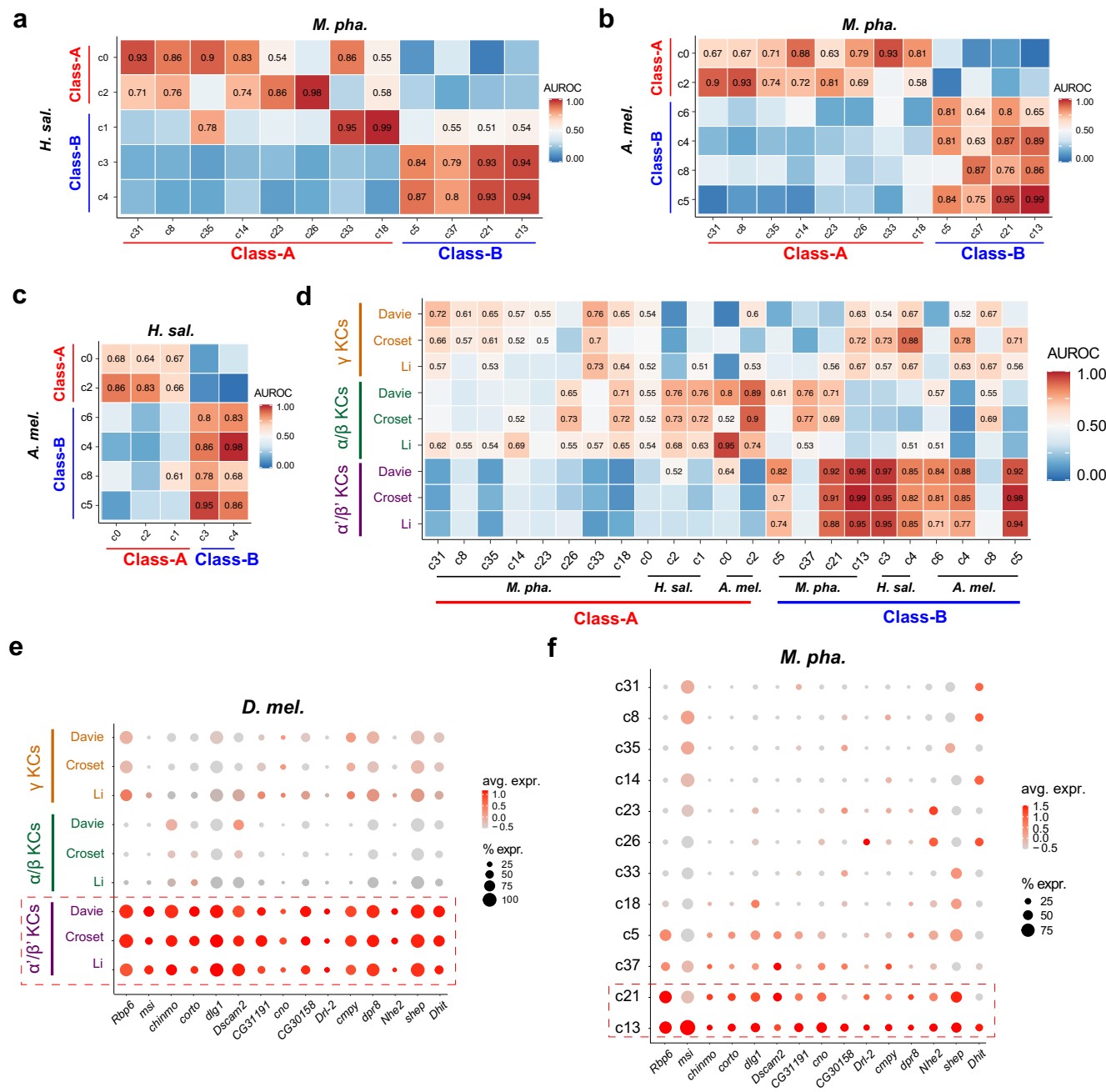

**Extended Data Fig. 5 | Comparison of Kenyon cells across species.** (**a**-**c**) Pairwise AUROC scores showing the cross-species transcriptional similarity of the KC subtypes from three hymenopteran insects (*M. pharaonis*, *H. saltator* and *A. mellifera*). Comparisons with AUROC scores > 0.5 are presented as exact values. (**d**) Pairwise AUROC scores showing the cross-species transcriptional similarity of hymenopteran and *Drosophila* KC subtypes. *Drosophila* KCs from three independent studies, namely Davie *et al*[24], Croset *et al*[27] and Li *et al*[36], were used for the analysis (see also Supplementary Data 2). Comparisons with AUROC scores > 0.5 are presented as exact values. (**e**-**f**) Dot plots showing the expression of representative shared DEGs up-regulated in *Drosophila* α'/β' KCs (**e**) and *Monomorium* c13/c21 KCs (**f**) in relative to the remaining KC subtypes.

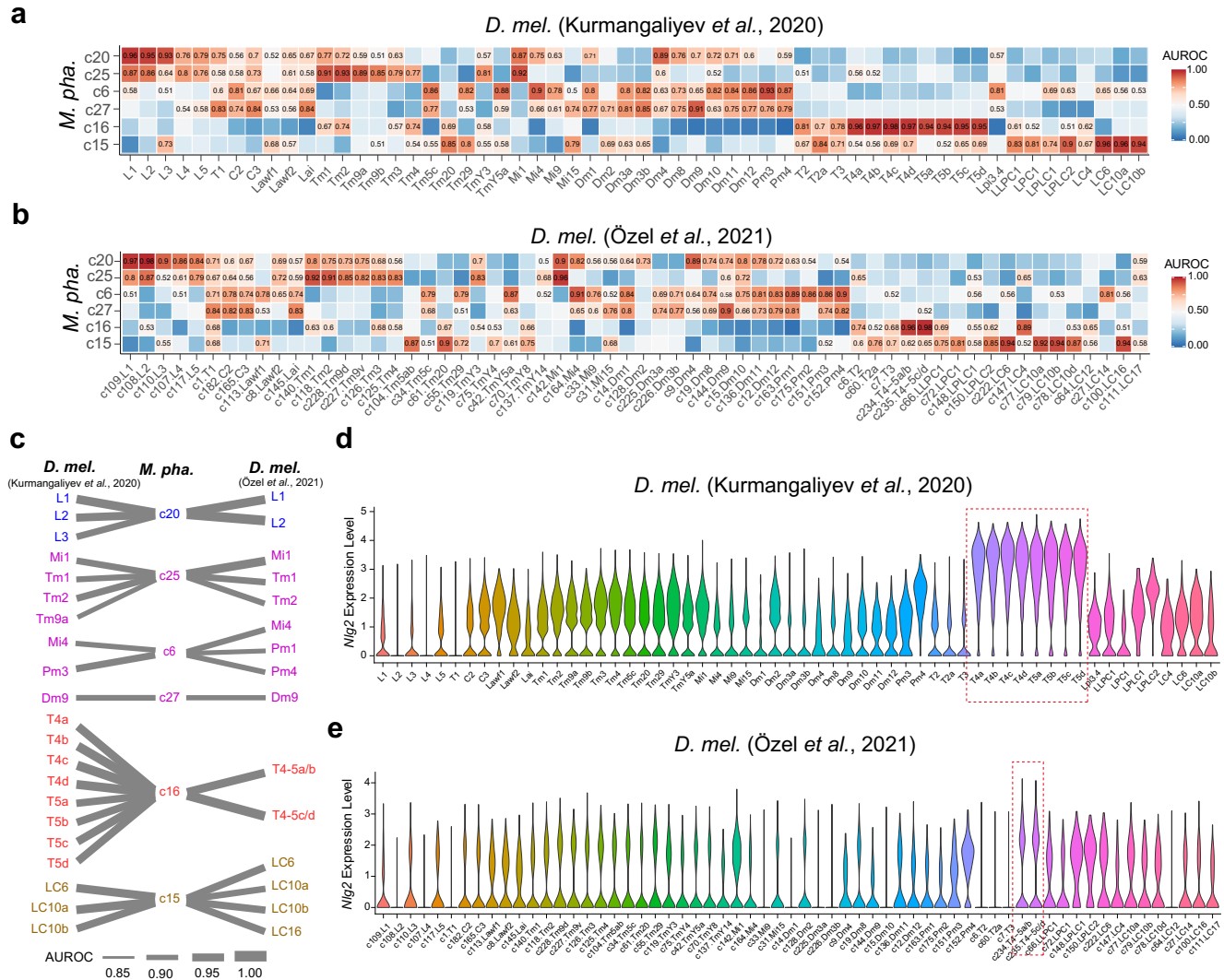

**Extended Data Fig. 6 | Comparison of optic lobe neurons across species.** (**a**, **b**) Pairwise AUROC scores showing the cross-species transcriptional similarity of OL clusters between *Monomorium* and *Drosophila*. *Drosophila* OL cell clusters from two independent studies that focus exclusively on the *Drosophila* optic lobes, namely Kurmangaliyev *et al*[41] and Özel *et al*[42], were used for the analyses. Comparisons with AUROC scores > 0.5 are presented as exact values. (**c**) Correspondence of OL clusters between *Monomorium* and *Drosophila* as predicted by the AUROC scores in panel **a** and panel **b**. Each line links a *Monomorium* OL cluster to its top hit among the *Drosophila* OL clusters according to AUROC scores, with line thickness being proportional to the score. A second hit is plotted as well, when the difference between the top and second AUROC score was less than 0.05. (**d**, **e**) Violin plots showing the expression of *Nlg2* across the *Drosophila* OL clusters. Dashed boxes indicate the T4/T5 neurons.

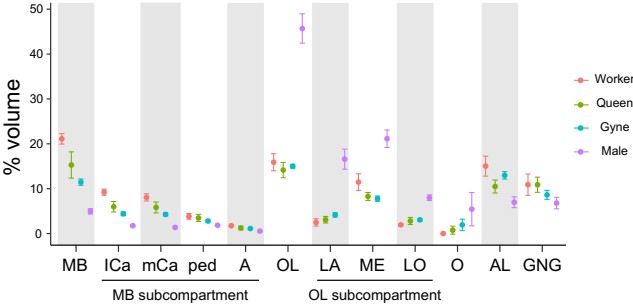

**Extended Data Fig. 7 | Relative volume of neuropil in different adult phenotypes.** The relative volume of a neuropil in an individual brain was calculated by dividing the volume of the neuropil with the entire brain volume (n = 5 for worker, 7 for queen, 6 for gyne and 5 for male). Data are presented as mean ± s.d. across replicates. MB: mushroom body; mCa: medial calyx of MB; lCa: lateral calyx of MB; ped: peduncle of MB; A: alpha lobe of MB; OL: optic lobe; LA: lamina of OL; ME: medulla of OL; LO: lobula of OL; O: ocelli; AL: antennal lobe; GNG: gnathal ganglia.

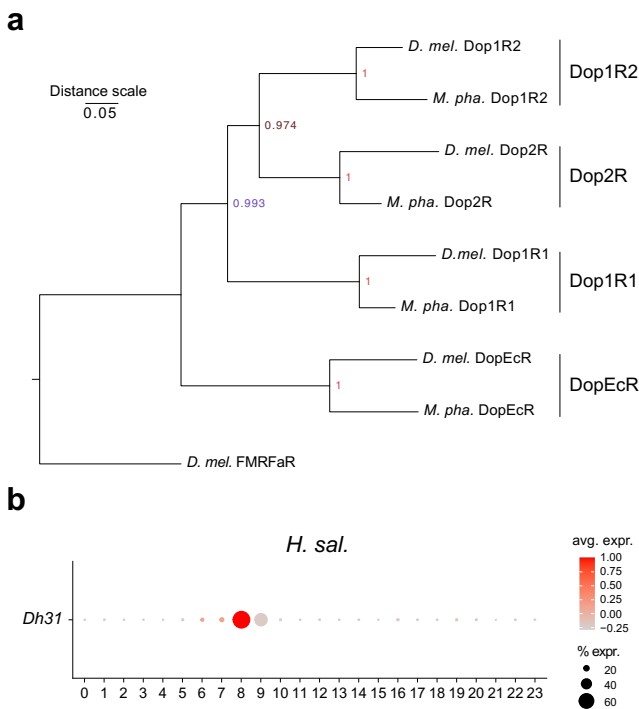

**Extended Data Fig. 8 | Phylogenetic analysis of dopamine receptors and expression analysis of *Dh31*.** (**a**) Phylogenetic relationship of the dopamine receptors from *M. pharaonis* and *D. melanogaster*, indicating that the *M. pharaonis* genome encodes four distinct dopamine receptors as observed in *D. melanogaster*. The *D. melanogaster* FMRFaR protein is used as the outgroup. The phylogenetic tree was built with the GPR domain sequences after alignment by MUSCLE (v.3.8.31) and with the neighbor-joining method implemented in MEGAX. The reliability of the tree was estimated with 1,000 bootstrap replications. (**b**) Dot plot showing the expression of *Dh31* across all the *H. saltator* cell clusters defined by Sheng *et al*[18]. Shade of dot represents mean expression within cluster, and size of dot represents percentage of cells within the cluster expressing that gene.

# Reporting Summary

## Statistics

For all statistical analyses, confirm that the following items are present in the figure legend, table legend, main text, or Methods section.

| n/a | Confirmed | |
|---|---|---|
| ☐ | ☒ | The exact sample size (*n*) for each experimental group/condition, given as a discrete number and unit of measurement |
| ☒ | ☐ | A statement on whether measurements were taken from distinct samples or whether the same sample was measured repeatedly |
| ☐ | ☒ | The statistical test(s) used AND whether they are one- or two-sided *Only common tests should be described solely by name; describe more complex techniques in the Methods section.* |
| ☒ | ☐ | A description of all covariates tested |
| ☐ | ☒ | A description of any assumptions or corrections, such as tests of normality and adjustment for multiple comparisons |
| ☐ | ☒ | A full description of the statistical parameters including central tendency (e.g. means) or other basic estimates (e.g. regression coefficient) AND variation (e.g. standard deviation) or associated estimates of uncertainty (e.g. confidence intervals) |
| ☐ | ☒ | For null hypothesis testing, the test statistic (e.g. *F*, *t*, *r*) with confidence intervals, effect sizes, degrees of freedom and *P* value noted *Give P values as exact values whenever suitable.* |
| ☒ | ☐ | For Bayesian analysis, information on the choice of priors and Markov chain Monte Carlo settings |
| ☒ | ☐ | For hierarchical and complex designs, identification of the appropriate level for tests and full reporting of outcomes |
| ☐ | ☒ | Estimates of effect sizes (e.g. Cohen's *d*, Pearson's *r*), indicating how they were calculated |

*Our web collection on statistics for biologists contains articles on many of the points above.*

## Software and code

Policy information about availability of computer code

| Data collection | No software was used. |
|---|---|
| Data analysis | Softwares used to analyze the data were described in details in the Methods section of the manuscript and listed below: Drop-seq_tools-1.13, STAR (2.6.1a_08-27), Seurat (v3.1.5), SOAPnuke (v 2.1.0), Hisat2 (v 2.1.0), DESeq2 (v1.22.0), BLASTP (blast-2.2.26), MetaNeighbor, Cytoscape (v 3.8.2), scCODA (v 0.1.6), AMIRA (v6.4), EZ-MET (x64, 6.0.7543), R (v.3.6) and R (v4.0.5). |

For manuscripts utilizing custom algorithms or software that are central to the research but not yet described in published literature, software must be made available to editors and reviewers. We strongly encourage code deposition in a community repository (e.g. GitHub). See the Nature Portfolio guidelines for submitting code & software for further information.

## Data

Policy information about availability of data

All manuscripts must include a data availability statement. This statement should provide the following information, where applicable:

- Accession codes, unique identifiers, or web links for publicly available datasets
- A description of any restrictions on data availability
- For clinical datasets or third party data, please ensure that the statement adheres to our policy

The raw snRNA-seq data of M. pharaonis generated in this study are deposited in the CNGB Nucleotide Sequence Archive (CNSA) with accession number CNP0001472. The reference genome, gene models, functional annotations of protein-coding genes, full marker gene list of each cell cluster, and all in-house scripts are deposited in the figshare repository under the link https://doi.org/10.6084/m9.figshare.16616353.

# Field-specific reporting

Please select the one below that is the best fit for your research. If you are not sure, read the appropriate sections before making your selection.

☒ Life sciences ☐ Behavioural & social sciences ☐ Ecological, evolutionary & environmental sciences

For a reference copy of the document with all sections, see nature.com/documents/nr-reporting-summary-flat.pdf

# Life sciences study design

All studies must disclose on these points even when the disclosure is negative.

| | |
|---|---|
| Sample size | We set 4 to 5 replicates for each of the four adult phenotypes of Monomorium pharaonis (i.e. workers, queens, gynes or males), as this number of replicate is suitable for controlling variation resulting from different bathes of library construction or sequencing. We obtained a total of 206,367 high-quality nuclei from the four adult phenotypes. This is 1.3 to 4 times the estimated cell number of 50,000 – 150,000 in a single individual ant brain, and thus is expected to be sufficient for capturing most cell types in the ant brain. |
| Data exclusions | No data were excluded. |
| Replication | Four to five replicates were analyzed for each adult phenotype to ensure the reproducibility of the findings. |
| Randomization | Samples were allocated into different groups according to their adult phenotype identity (i.e. workers, queens, gynes or males). |
| Blinding | Blinding was not relevant to this study because the adult phenotype identity (i.e. workers, queens, gynes or males) of each collected sample was very clear. |

# Reporting for specific materials, systems and methods

We require information from authors about some types of materials, experimental systems and methods used in many studies. Here, indicate whether each material, system or method listed is relevant to your study. If you are not sure if a list item applies to your research, read the appropriate section before selecting a response.

## Materials & experimental systems

| n/a | Involved in the study |
|---|---|
| ☒ | Antibodies |
| ☒ | Eukaryotic cell lines |
| ☒ | Palaeontology and archaeology |
| ☐ | ☒ Animals and other organisms |
| ☒ | Human research participants |
| ☒ | Clinical data |
| ☒ | Dual use research of concern |

## Methods

| n/a | Involved in the study |
|---|---|
| ☒ | ChIP-seq |
| ☒ | Flow cytometry |
| ☒ | MRI-based neuroimaging |

# Animals and other organisms

Policy information about studies involving animals; ARRIVE guidelines recommended for reporting animal research

| | |
|---|---|
| Laboratory animals | The original colony of Monomorium pharaonis was collected in 2016 from a resident house in Mengla, Xishuangbanna, Yunnan Province, China, and split into hundreds of sub-colonies in the lab in the subsequent years. All colonies were reared at 27℃, 65% RH and a 12/12 hr light/dark cycle. The rearing of gynes and males was induced in newly split colonies where inseminated and egg-laying queens were removed, and where easily recognizable male pupae were continuously removed to prevent that newly hatching gynes became inseminated. The eclosion date of males and gynes were recorded. The queens were collected from stable, mature colonies in which they were actively laying eggs. The demographic states of the colonies were frequently surveyed, so the ages of queens could be estimated, albeit less accurately than the gynes and males. Workers were randomly collected from colonies, both inside and outside of nests, so these samples covered both young (nursing) and old (foraging) workers. At the moment of dissection, gynes were 5-10 days post-eclosion, queens were 3-6 months post-eclosion, and males were 3-14 days post-eclosion, while the age of workers was not recorded. |
| Wild animals | The study did not involve wild animals. |
| Field-collected samples | The study did not involve samples collected from the field. |
| Ethics oversight | No ethical approval was required, as the studied species is an ant species that is commonly found around the world and can be easily maintained in lab. |

Note that full information on the approval of the study protocol must also be provided in the manuscript.

