## [Peer Review File. · Nature Ecology & Evolution]

Peer Review Information

Journal: Nature Ecology & Evolution

Manuscript Title: A single-cell transcriptomic atlas tracking the neural basis of division of labor in an ant superorganism

Corresponding author name(s): Chuanyu Liu, Guojie Zhang, Weiwei Liu

Editorial Notes:

Reviewer Comments & Decisions:

Decision Letter, initial version:
--

24th November 2021

Dear Guojie,

Your manuscript entitled "A single-cell transcriptomic atlas tracking the neural basis of division of labor in an ant superorganism" has now been seen by 4 reviewers, whose comments are attached. The reviewers have raised a number of concerns which will need to be addressed before we can offer publication in Nature Ecology & Evolution. We will therefore need to see your responses to the criticisms raised and to some editorial concerns, along with a revised manuscript, before we can reach a final decision regarding publication.

We therefore invite you to revise your manuscript taking into account all reviewer and editor comments. Please highlight all changes in the manuscript text file in Microsoft Word format.

* Include a "Response to reviewers" document detailing, point-by-point, how you addressed each

reviewer comment. If no action was taken to address a point, you must provide a compelling argument. This response will be sent back to the reviewers along with the revised manuscript.

* If you have not done so already please begin to revise your manuscript so that it conforms to our Article format instructions at <http://www.nature.com/natecolevol/info/final-submission>. Refer also to any guidelines provided in this letter.

[REDACTED]

Nature Ecology & Evolution is committed to improving transparency in authorship. As part of our efforts in this direction, we are now requesting that all authors identified as 'corresponding author' on published papers create and link their Open Researcher and Contributor Identifier (ORCID) with their account on the Manuscript Tracking System (MTS), prior to acceptance. ORCID helps the scientific community achieve unambiguous attribution of all scholarly contributions. You can create and link your ORCID from the home page of the MTS by clicking on 'Modify my Springer Nature account'. For more information please visit www.springernature.com/orcid.

[REDACTED]

Reviewer expertise:

Reviewer #1: evolution of sociality in insects

Reviewer #2: single-cell RNA-Seq in insects, neurogenetics

2Reviewer #3: neurobiology of social behaviours in invertebrates

Reviewer #4: single-cell RNA-Seq in insects, neurogenetics

Reviewers' comments:

Reviewer #1 (Remarks to the Author):

This study uses scRNA seq to characterize the relative neuroanatomy and functional neural cell diversity of eusocial versus solitary insects. These study questions, the study system, and the approach are broadly interesting.

The manuscript starts by emphasizing a goal of assessing the social brain hypothesis, whether social versus solitary organisms invest relatively more in neuroanatomy or function. Much of the analyses (Figs. 2,4,5) emphasizes comparison of eusocial ants and solitary fruit flies. Recent scRNA papers have stressed that there are many potential biases introduced at each processing and analysis step, depending exactly on the details. It is problematic to compare datasets between species that were collected and analyzed with different protocols. This said, even though the samples were clearly collected and processed in different labs and with different protocols, the ms was not clear about whether the *Drosophila* dataset was re-analyzed by the authors. The text suggested it was not: L156-158 "The *Monomorium* KCs could be divided into 12 distinct clusters, while a previous study detected five KC clusters in *H. saltator* -- still a higher number than the three KC clusters that could be identified in *Drosophila*". Since clustering algorithms are completely dependent on details of the analysis, including parameter choice, inferred cluster number is somewhat arbitrary and it does not necessarily reflect biological differences (this is hinted at in the Methods, L648-649 "The resolution parameter for clustering was set to 1.5 to produce an appropriate number of clusters without over-splitting"). On this note, I could find no statistical support (e.g., bootstrap support) for any of the hierarchical clustering analysis (Fig. 3a). Generally, throughout the ms, there were often strong statements made with little or no statistical support provided.

Several analysis details also seem problematic. For example, why were age categories in the *Drosophila* dataset lumped together in Figure 2, likely contributing to relatively very little variation between pools? Why is a Student's t-test used when more appropriate tests (GLM) are used later (L798-804)? More generally, using scRNA seq seems like a rather roundabout way of estimating the relative volume of neuropils within the brain. A more straightforward way would be to directly quantify relative brain volume e.g., with confocal microscopy (as shown in figure 5e) or with microCT. Directly quantifying cell number would also be more straightforward (e.g., Godfrey and Gronenberg 2019 *J Comp Phys*).

Even more biologically important, *Drosophila* and *Monomorium/Harpegnathos* differ in many ways besides just sociality, so it is not reasonable to suggest that any differences (e.g., in the relative size of certain neuropils, or in scRNA seq clusters, etc.) are caused by differences in social organization, or even statistically associated with differences in social organization. A much broader comparative study is necessary to make and support these claims. These claims are repeatedly made throughout the ms,

3including in the Discussion, although later in the Discussion the authors acknowledge "...enlargement of mushroom bodies" may be a "pre-adaptation rather than an outcome of sociality", and "direct comparisons with close hymenopteran outgroups will be needed". *Drosophila* also has many well-known derived features, so it does not make sense to suggest that *Drosophila* represents the insect ancestral state and *Monomorium* the derived state (L477 "Many of the evolutionarily derived KC subtypes in ants..."; Abstract L40 "...most newly derived subtypes enriched in worker brains"..).

Another important issue is variation within the worker caste. Many transcriptomic studies in honeybees and ants based on whole body, head, or brain tissue have shown the gene expression profiles vary dramatically within the worker caste dependent on worker age and task (i.e. nurse versus forager). This "age polyethism" corresponding to large transcriptomic changes has also been shown in *M. pharaonis*. Further details are necessary regarding how workers were collected (inside the nest, outside the nest?), and worker age or at least task should be controlled to account for and to understand variation within the worker caste.

Further sampling details are also needed. For example, precisely how many individual brains were pooled per replicate? What were the mentioned batch effects? Where did the biological replicates come from, i.e. what does "biological replicates" mean in the context of this study? Do all samples from a single colony, and if so, what biological variation do the replicates capture? Were the gynes uniseminated? If they were not physically separated from males, they would be expected to be inseminated, since *M. pharaonis* virgin gynes mate with males soon after eclosion. All of these details are expected to have large effects on transcriptomic profiles.

Reviewer #2 (Remarks to the Author):

I have really enjoyed reading this manuscript. Li, Wang, Zhang, Liu et al. have sought to explore the "social brain hypothesis", or the idea that sociality drives the evolution of larger brains. They have done it in a non-conventional (from the mammalian point of view) system - the pharaoh ant. They generated a large single-nucleus data set for the brains of ants of different sexes and social roles. They use these data to argue that sociality does not necessarily require a larger brain (the brains of the solitary *Drosophila* and the social ants seem to have approximately the same number of neurons). Instead, sociality may be associated with the emergence of a brain whose cell type composition can be specified during the development to adapt to a predetermined social role later. I think this is a really interesting take. I believe that the manuscript is a great fit for NEE because it tackles an interesting evolutionary question and comes with an extensive data set that will be useful for the social insect community and beyond. Therefore, I recommend the manuscript for publication after my concerns below have been addressed.

MAJOR POINTS:

1. Throughout the text, the authors compare ant brains to the brain of *Drosophila* and draw conclusions about the differences between solitary and social insects (e.g. one of the results sections

4is called “Differences between social and solitary insect brains”). This is not always justified as the families to which flies and ants belong are quite divergent, and it is impossible to say whether the differences between these two groups are linked to sociality or not without comparing ants to more closely related solitary species (e.g. solitary wasps or bees). The authors do acknowledge this at one point in the discussion, but I believe that they have to clearly state throughout the text that ants and flies are very different species, and the connection between sociality and the expansion or contraction of cell types in ants when compared to *Drosophila* is merely suggestive. Clearly stating this will not diminish the importance of the paper in any way, since the most interesting conclusions come from comparing the cell type composition between different castes and sexes of *Monomorium*.

2. Many conclusions of the paper are based on comparing relative abundances of cell types in different sexes or castes, which the authors do by fitting quasibinomial generalized linear models to the frequencies of individual cell types. Strictly speaking, this is not the most appropriate statistical approach. Cell type proportions are compositional data, meaning that an increase in the proportion of one cell type will always lead to the decrease in the proportion of other cell types - something that the current model does not account for. Dirichlet multinomial models are commonly used in metagenomics to work with compositional data, and these models have been recently implemented for estimating differential abundances in single-cell data, too (e.g. <https://www.biorxiv.org/content/10.1101/2020.12.14.422688v2.abstract>). Alternatively, especially given the “continuity” between the clusters in the ant data, the authors may consider applying clustering-free kNN-based differential abundance testing, e.g. as described by Dann et al. 2021 (PMID: 34594043). The latter approach seems to be quite user-friendly, and it is also implemented in R, which seems to be preferred by the authors.

3. In the section describing KC types, I am not entirely convinced by the conclusions drawn from the comparison to *Drosophila*. What makes me doubtful is the fact that the honeybee literature seems to have provided solid evidence that class II neurons are homologous to the *Drosophila* gamma. Conversely, large-type KCs are later-born neurons (based on their cell body position inside the calyces in the honeybee), so they should presumably correspond to either a'b' or ab in *Drosophila*. Given that the results presented in this paper contradict the current knowledge, I suggest being especially careful in presenting the evidence. For example, AUROC scores do not seem very high to me, except the edges between c13/c21 and a'b'. In extended fig. 2e, I see a lot of 0.7-0.8 scores outside the diagonal line. Second, which *Drosophila* data set did the authors use? Davie et al. may not be the best for KCs as KCs exhibit strong batch effects in those data (see Fig. S3 of Davie et al.). Do the authors get the same result with Davie et al. and Croset et al. data? Third, did the authors validate the correlation results by looking at individual marker genes known in *Drosophila*? For example, is *Imp*, which specifies the early KC fate, expressed in Class II or Class I neurons in *Monomorium*? Fourth, perhaps the authors could compare their data to the honeybee single-cell data published by Traniello et al. 2020 (PMID: 32080242). The published data set is rather small, but it is enriched in KCs, and the numbers should be sufficient to separate Class II from Class I KCs. Finally, the authors define their Class II cells as everything that does not express IKC markers. Technically, this can be a mixture of Class II, sKCs, and mKCs (the fact that ants have lost mKast does not mean that they have lost mKCs).

4. A large portion of the results section describes the cell types of the optic lobes, and some of these

5results are very cool, e.g. the absence of T4/T5 in workers. However, the authors appear to have used somewhat outdated *Drosophila* atlases to assign cell type identities to their ant clusters. Kurmangaliyev et al. 2020 (PMID: 33125872) and Özel, Simon et al. 2021 (PMID: 33149298) have published much more detailed and better annotated optic lobe atlases than the Davie et al. and Croset et al. data sets currently used by the authors. Frankly, I am not completely sure whether using the newer data will result in a major improvement given the large evolutionary distances between ants and flies and the fact that the better optic lobe atlases are better because they resolve lowly represented cell types, which may have not been well captured in the ant. Still, I would like to know whether the authors have tried to use the newer *Drosophila* data.

5. Finally, the link to longevity in the end and in the abstract feels somewhat artificial and separated from the rest of the manuscript. I feel that the rest of the conclusions are exciting on their own, and longevity could only be mentioned in passing.

MINOR POINTS:

Line 42: "generalized reminiscent" -> "generally reminiscent"

1st paragraph of Introduction: I would mention here that ants seem to have the same number of brain cells as flies (based on Godfrey et al., which is already cited elsewhere).

Line 83: Saying that only workers were sequenced in *Harpegnathos* is somewhat misleading, I would say that it was a comparison between workers and pseudo-queens.

Extended fig. 1e: Not very important, but I cannot help but wonder how a "negative control" would look like - would worker sn data correlate equally well with queen bulk data? Are these plots really informative?

Line 116: Given that many clusters form a "continuum", could the authors perhaps explain in the methods section in more detail how they chose the optimal clustering resolution?

Lines 216-221: It is not clear to me why KCs, even if they are processing visual information, should express molecules related to PR axon guidance. As far as I know, PRs do not directly synapse on KCs. The same goes for compound eye development. I am very intrigued that these GO terms are popping up here, but GO term analyses are very artifact-prone, especially in non-model species. Therefore, I suggest verifying which specific genes drive the enrichment of these GO-terms in these clusters, and whether this makes sense.

Lines 230-233: Downsampling is not enough to support the statement about the higher diversity of KC clusters in *Monomorium* because *Harpegnathos* data are not just represented by fewer cells, they are also much shallower (fewer UMIs / genes per cell), so they may not allow discriminating between closely related cell types.

Lines 280-294: The authors may add that T4/T5 perform motion detection in *Drosophila* - a task

6required for successful mating.

Lines 531-539: I am not sure I agree with the statement that “worker phenotypes did not exist before the ants made their major evolutionary transition to superorganismality, so novel genes and directional selection on their effects should be particularly expected for worker-expressed genes”. Primitively social species exhibit polyethism, meaning they go through phases where they do more “worker-like” and more “queen-like” things at different stages of their life. Thus, both the worker and the queen phenotype seem like secondary specializations to me, although I am open to other opinions.

Reviewer #3 (Remarks to the Author):

The authors present whole brain single-cell analysis of the pharaoh ant, interrogating cell composition across behavioral/reproductive phenotypes. The authors present ample sequencing data, and corroborate some of their findings with anatomical and physiological data. Results from this study will be of great interest to hymenopterists and may be important for research on the evolution of sociality more broadly. My comments are aimed at making this manuscript suitable for a broad audience.

Framing and Interpretation

Introduction: For the purposes of the data presented, I did not find the introduction of the social brain hypothesis to be convincing or satisfying. While the authors mention that “evidence for this hypothesis is, however, mixed” (line 65-66), they focus their introduction and references on the evolution of primate social systems. There is a wealth of literature discussing the social brain hypothesis in social insects in particular (see Farris 2016, Lihoreau 2012). It will be important for the authors to integrate more background on the study of the social brain hypothesis and social insects in particular into their introductory framing. As it stands right now, the jump from primates to ants too large and not well-motivated. While I find the authors’ position to look at brain specialization rather than enlargement (line 74) to be compelling, I do not think the Introduction or framing of the manuscript adequately demonstrates to a broad audience why it is important to study this particular species of ant (i.e. why is filling the gap mentioned in line 83-86 important to a more general audience?).

Figure 2: I recommend that the authors provide more background on the differences in social structure between *H. sal* and *M. pha* before the presentation of this figure (i.e. gamergates are not introduced until line 418-419). Without it, the significance of panels c and d are lost.

Throughout the text, the authors use the word “frequency” in a way that is imprecise, non-specific, or misleading, i.e. line 280 “clusters reached their highest frequency in male brains,” line 247-348, “male brains reached the highest frequencies not only in all OL clusters”, radar plot figure legends etc. Since the paper deals with developmental time, these phrases may imply claims that are not supported by the experimental design or results. Please edit throughout to avoid misinterpretation of results.

Please provide more references for the basic behavioral ecology, for example: lines 274-275 “ants rely less on visual stimuli than fruit flies”, lines “gynes depend to some extent on visual input before they are inseminated”, line 309, “assess the optimal time for nest-budding dispersal” etc, line 284 need to

7specify that this is for courtship in flies, not ants. References and background need to be strengthened throughout the text to give evidence to the authors' claims and improve data reporting.

Related to above, I found several of the claims regarding neural activity or behavior to be interesting ideas but poorly-supported by references or the data presented. Line 308-310 for example, "queens retain a good sensitivity to light but do not have good vision"—this is a very broad claim that is insufficiently discussed. What aspect of vision may not be good? Moreover, in line 300—the "rhythmic behavior" GO term seems very important to the author's claims, and it would be useful here to specify what annotations are included (for example, this term includes locomotor rhythm—is that also found in the data here?).

Specialization and complementation of social brains—I find these results to be very interesting, but I think some of the authors' claims are too strong or not given sufficient discussion/contextualization.

Line 345—"all of which reflect different use of the same set of cell types". More accurately, the data presented can only suggest this possibility.

Lines 352-361— I don't think the statements "worker brains are almost the opposite of male brains" (line 352) or "male and worker brains are functionally each other's mirror image" (line 360) are adequately supported by their data or discussed. For example, no functional data is presented. In the absence of supporting data, these statements are empty or meaningless, and are thus susceptible to misinterpretation. I think the authors can strengthen their discussion here by contextualizing their results in the larger understanding of the social brain hypothesis and social insects.

Line 416 and beyond: "the gyne-queen transition" and "worker-gamergate transition"—I think the putative neuroprotective results are interesting. However, from the Methods, it does not appear that the authors directly studied the transition between the gyne state and the queen state. The "transition" period suggests a developmental period between mating and before egg-laying, but I'm not sure from the methods that that is true of the animals used in this study (see comment below). The authors need to be very precise in language here that they are comparing gynes and queens, not directly probing the transition period. I do not think this takes away from their very interesting putative results on ensheathing glia and increased lifespan.

Line 545-546 "all phenotypes remain disposable and replaceable"—I think that their results more strongly support the opposite—that the neural phenotypes found in *M. pha* greatly complement each other, and are thus none are disposable because all are required to make up the whole. What do the authors mean by this?

The Discussion should contextualize the main findings in the framework of the social brain hypothesis in social insects to strengthen the authors' arguments and make this study of interest to a broad audience.

Data presentation and methods

Replicates and animals used in the study: The authors should take care to be upfront, clear, and consistent in the presentation of their biological replicates in the main text and Materials and Methods.

8Information about the animals used in the study is spread across lines 106-107, lines 591-593, lines 602-603, lines 641-642, Extended Figure 1, etc. It's challenging to piece together the biological attributes of the animals used for study: i.e. how many n's in each caste is not revealed until line 641, and even then it is confusing as whether these n's are a subset of the "30-50 brains of the same adult phenotype" (lines 602-603) used for single nuclei isolation, or if each replicate is 30-50 brains, or if each replicate is a single individual (Extended Data 1). Please clarify. Additionally, what is the mating status of gynes/males?

I found the statements in lines 436-437 and 562-564 to be very bold and not sufficiently supported. Can the authors please comment on how ages of the samples were determined (line 592-693), typical lifespan for the different phenotypes, how they ensured that gynes didn't mate, and briefly discuss any confounds that might exist in scenarios where age was not recorded or age within a phenotype varied greatly (i.e. queens).

Line 627: is the in-house script publicly available?

Figure 2 lines 171-172: are the dots averages from individuals in each phenotype, or pooled samples? The bars here (e.g. KC bars) imply a spread that is greater than the data points shown. Please specify in the figure legend if samples were pooled and specify in the methods if individual ants were barcoded.

Figure 3: is cluster 31 also missing from males, or just very low? The latter seems to be what is suggested in Figure 4, and if that is accurate, an internal reference to Figure 4 could be useful here.

Figure 4: the righthand side of panel A is not adequately explained in either the text or figure legend. There may be some typos here too (does $Dm = Dm9$?)

Figure 4d: the dots here seem to represent individuals—please clarify in legend and methods if so (see above comment for Figure 2).

Dopamine administration: because this manipulation seems very important to the authors' claims about the gonadotrophic effects of dopamine, they should expand on identification of yolky oocytes in the methods and provide images.

Reviewer #4 (Remarks to the Author):

In this manuscript, the authors performed single-nucleus RNAseq for adult ant brains from males, gynes, workers, and queens. Through thorough analyses, the authors compared the brain cell composition and cell states among these four castes, as well as with brain cells from *Drosophila* and another ant species *Harpegnathos saltator*. A number of interesting observations were reported. This study stands as the first comprehensive single-cell transcriptomic profiling of brain cells from an ant colony, and will be an important addition for understanding the insect brain function and evolution. However, I have several major concerns about data analysis and data interpretation that the authors

9should address.

1. Line 156-159: The Authors should reconsider this statement. It is difficult to make a solid conclusion just based on cluster number, without functional validation. Under different resolutions (Leiden resolution for example), a group of cells can be assigned as one cluster or can be many. See recent discussion on Fly Cell Atlas, Figure S3 (Li, Janssens et al., BioRxiv). Also, sub-clustering analysis can commonly reveal more cell subtypes. Total cell number also affects the cluster number.

2. Fig 2c, d: When comparing cell compositions between ant and fly brains, two important things should be noted: First, current study used snRNAseq and Davie et al. 2018 used scRNAseq. scRNAseq, compared with snRNAseq, may have sampling bias. For example, glia cells tend to be more difficult to be isolated for scRNAseq than neurons. Second, when comparing a certain cell type, for example OPNs, the Authors should check carefully whether all OPNs have been annotated from referred datasets. For instance, it is possible that only a fraction of sequenced OPNs are annotated as OPN clusters due to the limit of available markers. This will cause a problem when making a conclusion based on the annotated cell number, but not the "real cell number".

3. Line 218-221: it is interesting that c13 shows eye-related function genes, but it should be noted that many axon guidance genes are conserved across different brain regions; so the GO term "photoreceptor cell axon guidance" doesn't necessarily mean it is specifically linked to eye functions, but may be linked with axon guidance of other types of neurons. Sometimes, or in most cases, GO terms cannot precisely predict a cell's function. It is also not clear to me how a KC cell type shows compound eye development features. Did the Authors mean retinal cells, like photoreceptors, pigment cells, and cone cells, or optical lobe neurons?

4. Line 344: the recent Fly Cell Atlas preprint (Li, Janssens et al, BioRxiv) also compared male and female cells between male and female fly heads. The Authors should refer to this newer and more comprehensive dataset for comparison.

5. Figure 5d, and line 348: photoreceptors (PRs) are normally excluded from dissected brain (at least in flies), that's why PRs are not detected in fly brain atlas data (Davie et al 2018), but in head atlas data (Fly Cell Atlas); It is possible that the detected PRs from ant brains are mostly reflect dissection bias, but not real cell type composition. The authors should validate this part.

6. Fig 6a, line 385: it was concluded that a large fraction of clusters showed cell frequency changes from gynes to queens, like cluster 25, 27 etc. Since the insemination happened to adult gynes, where are these cell frequency changes coming from? Is there neurogenesis or programmed neuronal death during the transition from gynes to queens? If yes, more evidence or clarification should be provided.

7. In the result section of "Dopamine circuit remodeling...", it is not very clear to me in which level the gyne and queen neurons are compared. Normally, there are two levels of comparison between two closely-related scRNAseq datasets, like current data of gynes and queens, cell composition level and gene expression level. In the cell composition level, it seems, for some clusters, there are significant changes (this is related to above point 6 and should be addressed). In the gene expression level, even if one cluster contains the same fraction of cells from gynes and queens, they may still show

10significant differences of certain genes' expression (see similar analysis in Figure 6A-D in Fly Cell Atlas, Li, Janssens et al). The Authors need to clearly distinguish these two types of comparison.

*****END*****

Author Rebuttal to Initial commentsReviewer #1 (Remarks to the Author):

This study uses scRNA seq to characterize the relative neuroanatomy and functional neural cell diversity of eusocial versus solitary insects. These study questions, the study system, and the approach are broadly interesting.

The manuscript starts by emphasizing a goal of assessing the social brain hypothesis, whether social versus solitary organisms invest relatively more in neuroanatomy or function. Much of the analyses (Figs. 2,4,5) emphasizes comparison of eusocial ants and solitary fruit flies. Recent scRNA papers have stressed that there are many potential biases introduced at each processing and analysis step, depending exactly on the details. It is problematic to compare datasets between species that were collected and analyzed with different protocols. This said, even though the samples were clearly collected and processed in different labs and with different protocols, the ms was not clear about whether the *Drosophila* dataset was re-analyzed by the authors. The text suggested it was not: L156-158 “The *Monomorium* KCs could be divided into 12 distinct clusters, while a previous study detected five KC clusters in *H. saltator* -- still a higher number than the three KC clusters that could be identified in *Drosophila*”. Since clustering algorithms are completely dependent on details of the analysis, including parameter choice, inferred cluster number is somewhat arbitrary and it does not necessarily reflect biological differences (this is hinted at in the Methods, L648-649 “The resolution parameter for clustering was set to 1.5 to produce an appropriate number of clusters without over-splitting”). On this note, I could find no statistical support (e.g., bootstrap support) for any of the hierarchical clustering analysis (Fig. 3a). Generally, throughout the ms, there were often strong statements made with little or no statistical support provided.

Response: We appreciate the reviewer’s interest and advice. We realize that even though we could use our own pipeline to re-analyze the published data it is still hard to avoid all bias introduced by different experimental protocols. We therefore toned down the conclusion from the cross-species comparison in the revision. Specifically, we have removed the statements that the ants have more KC clusters than *Drosophila*, that *Monomorium* ants have more KC clusters than *Harpegnathos* ants, and that the diversity of OL neurons in *Monomorium* is lower than in *Drosophila*. Discussion points related to these findings have been modified accordingly.

In addition, we have added bootstrap analysis for the hierarchical clustering analyses and provided bootstrap supporting values in Fig. 3a and Extended Data Fig. 4.

Several analysis details also seem problematic.

- For example, why were age categories in the *Drosophila* dataset lumped together in Figure 2, likely contributing to relatively very little variation between pools?

Response: The age categories in the *Drosophila* dataset were not lumped together, but not shown in Figure 2. In this figure, each dot presents the relative abundance of a cell type in an ant adult phenotype or in a *Drosophila* sex. The relative abundance of a focal cell type in a specific phenotype or sex, was measured as the percentage of cells belonging to the focal cell type out of the total number of cells in a specific phenotype or sex, after combining cells from all biological replicates (or all libraries) of the ant phenotype or *Drosophila* sex. Accordingly, bars are the corresponding means \pm s.d. across ant adult phenotypes or *Drosophila* sexes. We have now clarified this in the legend of the new Figure 2.

Please also note that we have removed the old Fig. 2a and 2b, which compared the ratio of glia vs neurons between ants and flies in the revision, because we found that the ratio differences between ants and flies were not substantial enough to be unambiguous in light of the possible bias due to original protocols discussed above.

- Why is a Student's t-test used when more appropriate tests (GLM) are used later (L798-804)?

Response: We apologize for this mistake. We used a GLM test in the old Fig. 2c but mistakenly wrote Student's t-test in the legend. In the revision, we applied a more appropriate approach based on the Dirichlet multinomial model to do the test. The tool we used is named scCODA, which is specifically designed for compositional single-cell data analysis and published in Nature Communications recently¹. We have updated this panel accordingly (please see new Fig. 2a). The conclusion remains the same.

- More generally, using scRNA seq seems like a rather roundabout way of estimating the relative volume of neuropils within the brain. A more straightforward way would be to directly quantify relative brain volume e.g., with confocal microscopy (as shown in figure 5e) or with microCT. Directly quantifying cell number would also be more straightforward (e.g., Godfrey and Gronenberg 2019 J Comp Phys).

Response: The objective of this study was to interrogate the brain cell compositional differences rather than the volumetric differences of neuropils between castes/sexes. We agree that the direct quantification of relative brain volume or cell number could also provide information of brain differences between castes/sexes, but those methods lack the power to conduct comparisons at the cell type (or even subtype) resolution and do not help when exploring the biological function of an interested cell type. The main purpose of presenting the relative neuropil volume results in Fig. 5e was to provide additional evidence supporting that our single-cell dataset did reliably capture the brain differences between castes/sexes.

Even more biologically important, *Drosophila* and *Monomorium/Harpegnathos* differ in many ways besides just sociality, so it is not reasonable to suggest that any differences (e.g., in the relative size of certain neuropils, or in scRNA seq clusters, etc.) are caused by differences in social organization, or even statistically associated with differences in social organization. A much broader comparative study is necessary to make and support these claims. These claims are repeatedly made throughout the ms, including in the Discussion, although later in the Discussion the authors acknowledge "...enlargement of mushroom bodies" may be a "pre-adaptation rather than an outcome of sociality", and "direct comparisons with close hymenopteran outgroups will be needed". *Drosophila* also has many well-known derived features, so it does not make sense to suggest that *Drosophila* represents the insect ancestral state and *Monomorium* the derived state (L477 "Many of the evolutionarily derived KC subtypes in ants..."; Abstract L40 "...most newly derived subtypes enriched in worker brains"...).

Response: We agree with this criticism and have revised statements relevant to this issue throughout the manuscript. For example, in the abstract, we modified "...most newly derived subtypes enriched in worker brains" into "...most subtypes being enriched in worker brains" (see line 40 in the revised manuscript). In results section, the subtitle "Differences between social and solitary insect brains" is changed into "Cell compositional differences between ant and fly brains" (see line 162). In discussion section, we modified "Many of the evolutionarily derived KC subtypes in ants..." into "Many of the lineage-specific KC subtypes in ants..." (see lines 496-497).

We also deleted the sentence “Because the degree of KC diversification is even more pronounced in socially advanced *M. pharaonis* than in socially less complex and phylogenetically basal *H. saltator*, it is tempting to hypothesize that the transition from solitary to social life was directly associated with expansion of brain regions involved in communications”.

Another important issue is variation within the worker caste. Many transcriptomic studies in honeybees and ants based on whole body, head, or brain tissue have shown the gene expression profiles vary dramatically within the worker caste dependent on worker age and task (i.e. nurse versus forager). This “age polyethism” corresponding to large transcriptomic changes has also been shown in *M. pharaonis*. Further details are necessary regarding how workers were collected (inside the nest, outside the nest?), and worker age or at least task should be controlled to account for and to understand variation within the worker caste.

Response: Since *M. pharaonis* workers are monomorphic, we cannot tell by external morphology whether a worker is a nurse or forager and it is technically challenging to track the ages of workers and their allocated tasks, we therefore collected mixed samples of workers which should represent the entire spectrum of different age groups of workers. We have added a description of this selection procedure in “Biological samples” under “Materials and Methods” in the revision (see lines 601-618). In this study, we focused on revealing how brains of different castes/sexes within a superorganismal colony are differentiated, rather than on the age-dynamics of neural plastic changes happening in workers as they age. The worker “age polyethism” topic is surely worthy of in-depth investigation, but was beyond the scope of our current study.

Further sampling details are also needed. For example, precisely how many individual brains were pooled per replicate? What were the mentioned batch effects? Where did the biological replicates come from, i.e. what does “biological replicates” mean in the context of this study? Do all samples from a single colony, and if so, what biological variation do the replicates capture? Were the gynes unseminated? If they were not physically separated from males, they would be expected to be inseminated, since *M. pharaonis* virgin gynes mate with males soon after eclosion. All of these details are expected to have large effects on transcriptomic profiles.

Response: These sampling details have now been further clarified in the “Materials and Methods” section of the revision (see lines 601-618). Specifically, we pooled 30-50 brains per replicate for each adult phenotype. All samples were collected from sub-colonies split from an original colony collected from Xishuangbanna, Yunnan province, China, in 2016. The gynes and males were collected from sub-colonies in which the inseminated egg-laying queens were removed. The males were separated from gynes in the pupal stage, so we are sure that gynes were un-inseminated. The replicates of the same adult phenotypes are expected to capture the variations between samples collected from different sub-colonies as well as the batch effects resulted from the differences in handling personnel, reagent lots and sequencing lots.

Reviewer #2 (Remarks to the Author):

I have really enjoyed reading this manuscript. Li, Wang, Zhang, Liu *et al.* have sought to explore the “social brain hypothesis”, or the idea that sociality drives the evolution of larger brains. They

have done it in a non-conventional (from the mammalian point of view) system - the pharaoh ant. They generated a large single-nucleus data set for the brains of ants of different sexes and social roles. They use these data to argue that sociality does not necessarily require a larger brain (the brains of the solitary *Drosophila* and the social ants seem to have approximately the same number of neurons). Instead, sociality may be associated with the emergence of a brain whose cell type composition can be specified during the development to adapt to a predetermined social role later. I think this is a really interesting take. I believe that the manuscript is a great fit for NEE because it tackles an interesting evolutionary question and comes with an extensive data set that will be useful for the social insect community and beyond. Therefore, I recommend the manuscript for publication after my concerns below have been addressed.

MAJOR POINTS:

1. Throughout the text, the authors compare ant brains to the brain of *Drosophila* and draw conclusions about the differences between solitary and social insects (e.g. one of the results sections is called “Differences between social and solitary insect brains”). This is not always justified as the families to which flies and ants belong are quite divergent, and it is impossible to say whether the differences between these two groups are linked to sociality or not without comparing ants to more closely related solitary species (e.g. solitary wasps or bees). The authors do acknowledge this at one point in the discussion, but I believe that they have to clearly state throughout the text that ants and flies are very different species, and the connection between sociality and the expansion or contraction of cell types in ants when compared to *Drosophila* is merely suggestive. Clearly stating this will not diminish the importance of the paper in any way, since the most interesting conclusions come from comparing the cell type composition between different castes and sexes of *Monomorium*.

Response: This comment is similar to a point raised by reviewer 1. We agree and have revised relevant text throughout the manuscript.

2. Many conclusions of the paper are based on comparing relative abundances of cell types in different sexes or castes, which the authors do by fitting quasibinomial generalized linear models to the frequencies of individual cell types. Strictly speaking, this is not the most appropriate statistical approach. Cell type proportions are compositional data, meaning that an increase in the proportion of one cell type will always lead to the decrease in the proportion of other cell types - something that the current model does not account for. Dirichlet multinomial models are commonly used in metagenomics to work with compositional data, and these models have been recently implemented for estimating differential abundances in single-cell data, too (e.g. <https://www.biorxiv.org/content/10.1101/2020.12.14.422688v2.abstract>). Alternatively, especially given the “continuity” between the clusters in the ant data, the authors may consider applying clustering-free kNN-based differential abundance testing, e.g. as described by Dann et al. 2021 (PMID: 34594043). The latter approach seems to be quite user-friendly, and it is also implemented in R, which seems to be preferred by the authors.

Response: Thank you for the helpful suggestions and the recommendation of the cutting-edge tools. We agree that statistical approaches based on Dirichlet multinomial models are more suitable for compositional data. Therefore, we implemented the recommended tool scCODA¹ to reassess the cell compositional differences between adult phenotypes in the revision.

We note that there are three important parameters to be considered when using scCODA. The first one is the FDR level. Unlike our previous GLM method which calculated an FDR for each cell cluster, scCODA reports cell clusters with credible abundance changes between samples according to an inferred value (zero, no statistically credible effect; non-zero, statistically credible effect) at a given FDR level. In practice, an FDR level of 0.2 is shown to be acceptable by the authors according to their applications of scCODA in five different real single-cell datasets (see Fig. 3 and related supplementary tables in their original paper¹). The second important parameter is the Hamiltonian Monte Carlo (HMC) chain length, which is usually set according to the number of cell clusters. An HMC chain length of 800,000 is sufficient for our dataset which contains 43 cell clusters. Finally, choosing the right reference (which is assumed to be unchanged in abundance in different samples) is important for all compositional analyses². scCODA can automatically select an appropriate cell type as the reference or uses a pre-specified reference cell type to identify compositional changes for the rest cell types. As we do not have any prior knowledge about the best reference cell type for the four phenotypes of *Monomorium*, we generated two versions of results with scCODA based on two reference cell type selection strategies. In the first version, we used the ‘automatic reference selection’ method provided by scCODA, while in the second version, we specified c34 as the reference cell type for all comparisons, as c34 showed the least variation in relative abundance across the four adult phenotypes (see Fig. 5d in the revised manuscript). For both versions, FDR level was set to 0.2, the HMC chain length was set to 800,000 with burn-in of 10,000, and the fold change in relative abundance was set to > 1.3 .

In general, we found that scCODA yielded similar results as our previous quasibinomial GLM method (see Table 1 and 2 below). In the meanwhile, scCODA reported zero cell clusters as significant between female and male *Drosophila* flies based on datasets from two independent studies^{3,4} (see Table 3 below). Therefore, the conclusions of our previous manuscript are still valid. We have updated all related figures and tables with the scCODA results according to the ‘automatic reference selection’ version (please see Fig. 5b-c, Fig. 6a and Supplementary Table 4).

Table 1 | Cell clusters with credible abundance changes between *Monomorium* adult phenotypes obtained by scCODA (automatic reference selection) and our previous quasi-binomial GLM method.

Comparison	No. of clusters with credible changes			Specific in scCODA	Specific in GLM
	scCODA	GLM	Overlap		
Worker VS Queen	19	18	18	c3	-
Worker VS Gyne	26	25	24	c22, c36	c38
Worker VS Male	30	30	30	-	-
Queen VS Gyne	15	15	13	c17, c26	c35, c41
Queen VS Male	30	30	29	c32	c29
Gyne VS Male	28	25	25	c17, c21, c22	-

Table 2 | Cell clusters with credible abundance changes between *Monomorium* adult phenotypes obtained by scCODA (using c34 as reference) and our previous quasi-binomial GLM method.

Comparison	No. of clusters with credible changes			Specific in scCODA	Specific in GLM
	scCODA	GLM	Overlap		
Worker VS Queen	18	18	17	c3	c28
Worker VS Gyne	25	25	24	c36	c38

Worker VS Male	30	30	30	-	-
Queen VS Gyne	15	15	13	c27, c26	c35, c41
Queen VS Male	29	30	28	c32	c9, c29
Gyne VS Male	28	25	25	c17, c21, c22	-

Table 3 | Cell clusters with credible abundance changes between *Drosophila* sexes obtained by scCODA (automatic reference selection) and our previous quasi-binomial GLM method.

Dataset	No. of clusters with credible changes			Specific in scCODA	Specific in GLM
	scCODA	GLM	Overlap		
Whole brains (Davie et al.) ³	0	5	0	-	c23, c32, c33, c74, c85
Heads (Li et al.) ⁴	0	0	0	-	-

Because we obtained many interesting differences that needed interpretation at the cell cluster level, we decided to refrain from even more massive further analyses of the differences between castes/sexes using the clustering-free method Milo⁵, but the recommendation is appreciated.

3. In the section describing KC types, I am not entirely convinced by the conclusions drawn from the comparison to *Drosophila*. What makes me doubtful is the fact that the honeybee literature seems to have provided solid evidence that class II neurons are homologous to the *Drosophila* gamma. Conversely, large-type KCs are later-born neurons (based on their cell body position inside the calyces in the honeybee), so they should presumably correspond to either a 'b' or ab in *Drosophila*. Given that the results presented in this paper contradict the current knowledge, I suggest being especially careful in presenting the evidence. For example, AUROC scores do not seem very high to me, except the edges between c13/c21 and a'b'. In extended fig. 2e, I see a lot of 0.7-0.8 scores outside the diagonal line. Second, which *Drosophila* data set did the authors use? Davie *et al.* may not be the best for KCs as KCs exhibit strong batch effects in those data (see Fig. S3 of Davie *et al.*). Do the authors get the same result with Davie *et al.* and Croset *et al.* data? Third, did the authors validate the correlation results by looking at individual marker genes known in *Drosophila*? For example, is *Imp*, which specifies the early KC fate, expressed in Class II or Class I neurons in *Monomorium*? Fourth, perhaps the authors could compare their data to the honeybee single-cell data published by Traniello *et al.* 2020 (PMID: 32080242). The published data set is rather small, but it is enriched in KCs, and the numbers should be sufficient to separate Class II from Class I KCs. Finally, the authors define their Class II cells as everything that does not express IKC markers. Technically, this can be a mixture of Class II, sKCs, and mKCs (the fact that ants have lost mKast does not mean that they have lost mKCs).

Response: We agree that previous honeybee literature has provided evidence supporting that the hymenopteran class-II KCs are homologous to the *Drosophila* γ KCs, mainly based on birth order and dendritic morphology⁶⁻⁸. Here, like in other single-cell comparative studies, our cross-species cell cluster comparisons were based on transcriptomic similarity, which assumes that a higher similarity score suggests a higher functional similarity of two cell clusters in two different species.

We only used the dataset from Davie *et al.*³ in the previous version of our manuscript. In the revision, we used two additional *Drosophila* datasets (the midbrain dataset from Croset *et al.*⁹ and

the head dataset from Li *et al.*⁴) to examine the similarity between the ant and *Drosophila* KCs. As shown in the new Extended Data Fig. 5d, the *Drosophila* γ KCs from these two additional datasets do not show higher than 0.8 AUROC scores to any of the *Monomorium* KC clusters as estimated with the Davie dataset. In addition, the *Drosophila* γ KCs from Croset *et al* showed comparable scores to *Monomorium* c33 (0.70) and c13 KCs (0.72). Due to these relatively low similarity scores, we have decided to remove the links between the *Drosophila* γ KCs and the *Monomorium* KCs in the revised manuscript. However, we found that the similarity links between *Drosophila* α/β' KCs and *Monomorium* c21/c13 KCs are solid enough, as confirmed by the three independent *Drosophila* datasets which all reported a high AUROC score over 0.9 (see new Extended Data Fig. 5d). Besides, there are many marker genes shared by *Drosophila* α/β' KCs and *Monomorium* c21/c13 KCs, such as *msi*, *Rbp6* and *dlg1* (see new Extended Data Fig. 5e,f). This implies that the *Drosophila* α/β' KCs and the *Monomorium* c21/c13 KCs may account for some similar functions in the adult stages of these two distantly related insect lineages.

We also checked the expression of *Imp*. Interestingly, while *Imp* specifies the early KC fate during development¹⁰ and retains higher expression levels in adult γ KCs than α/β' and α/β KCs in *Drosophila*³, we found that the expression levels of *Imp* were quite low in all KC clusters in both ant species (see Fig. 1 below). This suggests that *Imp* is not that important in the mature KCs in adult ant brains. But we could not rule out the possibility that *Imp* also specifies the early KC fate in ants, which requires the examination of *Imp* expression in larval ant samples.

Fig. 1 | Expression of *Imp* across the KC clusters in *M. pharaonis*, *H. saltator* and *D. melanogaster*. The *Drosophila* plot was generated with data from Davie *et al*. Plots with data from Croset *et al* and Li *et al* show similar patterns and are thus not shown.

Finally, we compared our ant data with the honeybee data from Traniello *et al* (2020)¹¹ as suggested - a dataset containing a total of 2,205 cells obtained from two libraries. Because one library was constructed exclusively from the mushroom bodies (the other was from a whole brain), it has a good representation of the honeybee KCs, though the total captured cell number is rather small. We found that six out of the 13 honeybee cell clusters could be assigned as KCs according to the expression of the honeybee KC marker *PLCe*¹² (see new Extended Data Fig. 4c). After careful examination of the expression of the honeybee KC subtypes marker genes, we could clearly assign two clusters (c0 and c2) as class-I large-type (IKCs), one (c6) as middle-type (mKCs), and one (c4) as small-type KCs (sKCs), while one of remaining two clusters (c8) likely represents a mixture of mKCs and sKCs and the other (c5) presumably represents class-II KCs. Of note, the annotation of c5 as class-II KCs is still speculative, as no marker gene has been reported for the honeybee class-II KCs. Interestingly, the six honeybee KC clusters were also separated into two distinct groups based on gene expression patterns (see new Extended Data Fig. 4c), with one group comprising solely the IKC clusters (c0 and c2). Furthermore, by assessing the transcriptional similarity between the ant and honeybee KC clusters, we found that the two groups of honeybee KCs corresponded clearly to the two groups of ant KCs that we defined before (see new Fig. 3f and new Extended Data Fig. 5b,c). Accordingly, the reviewer is correct that our previously defined

'class-I' KCs (c31, c8, c35, c14, c23, c26, c33 and c18) in *Monomorium* are indeed IKCs, while the rest (c5, c37, c21 and c13) is a mixture of sKCs, mKCs and IICs. Secondly, these new results suggest that hymenopteran IKCs are functionally distinct from the other KC subtypes in adults, whereas the earliest born IICs share some similarities with the latest born s+mKCs. This is consistent with that sKCs, mKCs and IICs are traditionally called compact KCs while IKCs are referred to as noncompact KCs according to their distinct cell body sizes in adult mushroom bodies¹³.

We have modified Fig. 3f and corresponding statements in the main text according to these new results (see lines 196-249 in revised manuscript).

4. A large portion of the results section describes the cell types of the optic lobes, and some of these results are very cool, e.g. the absence of T4/T5 in workers. However, the authors appear to have used somewhat outdated *Drosophila* atlases to assign cell type identities to their ant clusters. Kurmangaliyev *et al.* 2020 (PMID: 33125872) and Özel, Simon *et al.* 2021 (PMID: 33149298) have published much more detailed and better annotated optic lobe atlases than the Davie *et al.* and Croset *et al.* data sets currently used by the authors. Frankly, I am not completely sure whether using the newer data will result in a major improvement given the large evolutionary distances between ants and flies and the fact that the better optic lobe atlases are better because they resolve lowly represented cell types, which may have not been well captured in the ant. Still, I would like to know whether the authors have tried to use the newer *Drosophila* data.

Response: Thank you for the recommendation of the two latest *Drosophila* optic lobe datasets. We found that comparing our *Monomorium* data to the adult optic lobe dataset from Özel *et al.* (2021)¹⁴ and the main dataset from Kurmangaliyev *et al.* (2020)¹⁵ did improve the cell type annotation of the *Monomorium* OL neurons. For example, according to the updated analysis results, we could observe that (i) the *Monomorium* c20 only shows high similarity scores to L1/L2/L3 subtypes (AUROC scores > 0.9) of the *Drosophila* lamina monopolar cells (LMCs) but not to the L4/L5 subtypes, whereas L1/L2/L3/L4/L5 formed a single cluster in Davie *et al.* and thus this kind of detail was masked; (ii) the *Monomorium* c25 shows high similarity scores exclusively to the *Drosophila* medulla neurons such as Medulla intrinsic neurons (Mi) and Transmedullary neurons (Tm), whereas our old result also linked c25 to the LMCs, and (iii) the *Monomorium* c15 shows high similarity (AUROC scores > 0.9) to the *Drosophila* lobula columnar (LC) cells, whereas we previously linked c15 to *Drosophila* T2a neurons.

We have updated Fig. 4a in the main text according to these new results as summarized in the new Extended Data Fig. 6a-c. We note also that these improved results do not affect our previous conclusions in the main text, because the cell-type identity of c16 (T4/T5 neurons) and c20 (LMCs) that were highlighted in our previous manuscript are unchanged.

5. Finally, the link to longevity in the end and in the abstract feels somewhat artificial and separated from the rest of the manuscript. I feel that the rest of the conclusions are exciting on their own, and longevity could only be mentioned in passing.

Response: We have modified the abstract, and shortened the results and discussions related to longevity in the revised manuscript. Now we mention the finding related to longevity in the context of cell compositional changes between gynes and queens, which also involve changes that might facilitate fecundity (please see lines 43-48, lines 422-439 and lines 572-598 in the revision).

MINOR POINTS:

Line 42: “generalized reminiscent” -> “generally reminiscent”

Response: We think that the word “generalized” is correct, as the “generalized” here for gyne and queen brains is relative to the highly “specialized” cell composition of the worker and male brains. We have added a comma between “generalized” and “reminiscent” in the revision to make this sentence more readable (please see line 43 in the revision).

1st paragraph of Introduction: I would mention here that ants seem to have the same number of brain cells as flies (based on Godfrey et al., which is already cited elsewhere).

Response: Yes, Godfrey surveyed representative species of ants and reported that brain cell numbers of ants ranged from 50,000 to 150,000, which is comparable to *Drosophila*. We have now referred to this result in the 1st paragraph of introduction in the revision (please see lines 68-69).

Line 83: Saying that only workers were sequenced in *Harpegnathos* is somewhat misleading, I would say that it was a comparison between workers and pseudo-queens.

Response: We have now changed the description to “So far, only a single ant species, *Harpegnathos saltator*, has been interrogated at the single-cell level and only for the midbrains of workers and gamergates (inseminated and reproductively active workers)” (please see lines 89-91 in the revised manuscript).

Extended fig. 1e: Not very important, but I cannot help but wonder how a “negative control” would look like - would worker sn data correlate equally well with queen bulk data? Are these plots really informative?

Response: Yes, the negative controls (e.g. worker sn data vs queen bulk data) yielded equally good correlation coefficients as observed in the positive controls (e.g. worker sn data vs worker bulk data) (see Fig. 2 below). But this finding is not that surprising, because the global gene expression patterns of the four adult phenotypes are quite similar to each other (see Fig. 3 below). After all, the fact that only a subset of genes show significant differential expression between adult phenotypes is less likely to affect the global expression correlation.

But these plots are informative. If we cannot observe an acceptable correlation coefficient between the sn data and the high-quality bulk data, we would consider that the quality of the sn data must be problematic, either due to RNA degeneration of the target tissues or due to the failure of library construction. So, we intend to keep Extended Data Fig. 1e in the manuscript.

Fig. 2 | The correlation of gene expression between bulk RNA-seq data and snRNA-seq data.

Fig. 3 | The correlation of bulk brain gene expression between any two of the four *Monomorium* adult phenotypes.

Line 116: Given that many clusters form a “continuum”, could the authors perhaps explain in the methods section in more detail how they chose the optimal clustering resolution?

Response: We think that the “continuum” of clusters is a typical feature of the UMAP visualizing method, as UMAP preserves more of the global data structure (i.e. the intercluster relationships) than other methods like t-SNE¹⁶. That means cell clusters with similar transcriptional patterns (e.g. the KC subtypes or the glial subtypes) will appear close to each other in the UMAP space. It is also notable that there are limitations for showing high-dimensional data in a 2D view, as it would inevitably lead to the overlap of some clusters in the plot. Therefore, we also present a 3D view of the whole atlas in GIF format (please see Supplementary Data 1).

Regarding the optimal clustering resolution, as stated in a recent Tutorial published in Nature protocols, there are no set rules for determining the optimal parameters, and researchers must typically make informed decisions that depend on the single-cell dataset at hand¹⁷. What we did to determine the optimal clustering resolution was a bit laborious. We first generated different versions using a series of resolutions (0.25 - 3 with a step of 0.25). Then we manually checked the clustering results of each version in order to find out the versions that could separate the major neuronal or glial cell types into different clusters. The manually examined cell types included Kenyon cells, olfactory projection neurons, photoreceptors, monoaminergic neurons, astrocyte-like glia, ensheathing glia, cortex glia and surface glia, as these cell types have been well studied in insects with well-established marker genes. We also required that all of the cell clusters in a qualified version must be supported by nuclei from two or more of the 17 samples (5 worker replicates + 4 queen replicates + 4 gyne replicates + 4 male replicates), ensuring that no cluster might be resulted from batch effects. The combination of these considerations led us to finally choose the version generated by a resolution of 1.5 (see Fig. 1b for the cell type annotations and new Supplementary Table 1 for the number of nuclei per cluster per sample). We have now provided these additional explanations in the Methods section (please see lines 676-686 in the revised manuscript).

Lines 216-221: It is not clear to me why KCs, even if they are processing visual information, should express molecules related to PR axon guidance. As far as I know, PRs do not directly synapse on KCs. The same goes for compound eye development. I am very intrigued that these GO terms are popping up here, but GO term analyses are very artifact-prone, especially in non-model species. Therefore, I suggest verifying which specific genes drive the enrichment of these GO-terms in these clusters, and whether this makes sense.

Response: We carefully examined the up-regulated genes for c13, which drive these GO terms. For the GO term “photoreceptor cell axon guidance”, the genes involved include (names for *Drosophila* homologs are used) *dyschronic*, *Fasciclin 2*, *Moesin*, *Nhe2*, *chiffon*, *hts* and *β -Spec*. The functions of these genes are found in photoreceptors as well as in other tissues. For the GO terms “eye development”, “eye morphogenesis”, “photoreceptor cell differentiation”, the genes involved include *canoe*, *musashi*, *dyschronic*, *Fasciclin 2*, *prospero*, *shibire*, *Moesin*, *Nhe2*, *Mob2*, *cnk*, *Rap1*, *APC-like*, *lola*, *rdx*, *Liprin-alpha*, *nej* and *p120ctn*. These genes generally regulate cell proliferation, cell differentiation, cell signaling, development and morphogenesis in a variety of tissues. Particularly, *Mob2*, *APC-like* and *lola* have reported functions in the eye. We agree that the GO term description is too ambiguous and could be the result of artifacts. We have therefore removed the statements of GO terms related to c13 KCs in the revised manuscript.

Lines 230-233: Downsampling is not enough to support the statement about the higher diversity of KC clusters in *Monomorium* because *Harpegnathos* data are not just represented by fewer cells, they are also much shallower (fewer UMIs / genes per cell), so they may not allow discriminating between closely related cell types.

Response: We appreciate this comment and have removed the statement “The higher number of IKC clusters in *Monomorium* relative to *Harpegnathos* suggests a further diversification of the class-I KCs in the *Monomorium* lineage, which was also supported by a down-sampled analysis with only 18k *Monomorium* worker nuclei.” in the revised manuscript.

Lines 280-294: The authors may add that T4/T5 perform motion detection in *Drosophila* - a task required for successful mating.

Response: Thanks for the suggestion. We have added this information in the revision (please see line 284).

Lines 531-539: I am not sure I agree with the statement that “worker phenotypes did not exist before the ants made their major evolutionary transition to superorganismality, so novel genes and directional selection on their effects should be particularly expected for worker-expressed genes”. Primitively social species exhibit polyethism, meaning they go through phases where they do more “worker-like” and more “queen-like” things at different stages of their life. Thus, both the worker and the queen phenotype seem like secondary specializations to me, although I am open to other opinions.

Response: What we mean here is workers in the sense of irreversibly differentiated non-reproductive caste phenotypes early in development. As we write, this is the defining characteristic of superorganismality as originally proposed by William Morton Wheeler. The polyethism that the

reviewer refers to should not be referred to in terms of workers and queens because, as the reviewer writes these differences are continuous, similar to vertebrate cooperative breeders – it is just role differentiation in the adult stage without deeper developmental background.

Reviewer #3 (Remarks to the Author):

The authors present whole brain single-cell analysis of the pharaoh ant, interrogating cell composition across behavioral/reproductive phenotypes. The authors present ample sequencing data, and corroborate some of their findings with anatomical and physiological data. Results from this study will be of great interest to hymenopterists and may be important for research on the evolution of sociality more broadly. My comments are aimed at making this manuscript suitable for a broad audience.

Framing and Interpretation

Introduction: For the purposes of the data presented, I did not find the introduction of the social brain hypothesis to be convincing or satisfying. While the authors mention that “evidence for this hypothesis is, however, mixed” (line 65-66), they focus their introduction and references on the evolution of primate social systems. There is a wealth of literature discussing the social brain hypothesis in social insects in particular (see Farris 2016, Lihoreau 2012). It will be important for the authors to integrate more background on the study of the social brain hypothesis and social insects in particular into their introductory framing. As it stands right now, the jump from primates to ants too large and not well-motivated. While I find the authors’ position to look at brain specialization rather than enlargement (line 74) to be compelling, I do not think the Introduction or framing of the manuscript adequately demonstrates to a broad audience why it is important to study this particular species of ant (i.e. why is filling the gap mentioned in line 83-86 important to a more general audience?).

Response: We have followed the reviewer’s advice and now integrated more background information on insect social brains in the introduction section. We modified the first paragraph in the revision and elaborated on how insect social systems are different from those of primates. We also included the recent results comparing brain cell numbers across Hymenopteran species and in a wide range of other animals. We put our argument into a general framework asking whether and to what extent sociality drive brain evolution. Please see the first paragraph in Introduction and the first paragraph in Discussion of the revised manuscript.

Figure 2: I recommend that the authors provide more background on the differences in social structure between *H. sal* and *M. pha* before the presentation of this figure (i.e. gamergates are not introduced until line 418-419). Without it, the significance of panels c and d are lost.

Response: We have modified a sentence in the Introduction section to introduce gamergates as inseminated and reproductively active workers in *Harpegnathos saltator* (please see lines 89-91 in the revision). We think that this information is sufficient for interpreting the data present in Figure 2, because this figure aims to highlight the cell compositional differences between ant and fly brains, rather than the variation between ant adult phenotypes or adult *Drosophila* sexes.

Throughout the text, the authors use the word “frequency” in a way that is imprecise, non-specific, or misleading, i.e. line 280 “clusters reached their highest frequency in male brains,” line 247-348, “male brains reached the highest frequencies not only in all OL clusters”, radar plot figure legends etc. Since the paper deals with developmental time, these phrases may imply claims that are not supported by the experimental design or results. Please edit throughout to avoid misinterpretation of results.

Response: Sorry for the imprecise use of “frequency”. We have replaced it either by “abundance” or by “relative abundance” according to the context throughout the manuscript.

Please provide more references for the basic behavioral ecology, for example: lines 274-275 “ants rely less on visual stimuli than fruit flies”, line 296 “gynes depend to some extent on visual input before they are inseminated”, line 309, “assess the optimal time for nest-budding dispersal” etc, line 284 need to specify that this is for courtship in flies, not ants. References and background need to be strengthened throughout the text to give evidence to the authors’ claims and improve data reporting.

Response: We have deleted the statement “the diversity of OL neurons in *Monomorium* is lower than in *Drosophila*” in the revision, as we find that, according to the latest studies that focus exclusively on *Drosophila* optic lobes^{14,15}, the optic lobe neurons are very complex and comprise more than one hundred neuronal subtypes, which are really challenging to be fully resolved in a whole brain cell atlas. Accordingly, we have deleted “ants rely less on visual stimuli than fruit flies” in the revision. We have also deleted the statement “gynes depend to some extent on visual input before they are inseminated”, because this is an inference and has not been explicitly studied.

Regarding line 309, we have now added a reference (Tay *et al.* (2015)¹⁸) to support our hypothesis that pharaoh ant queens may retain sensitivity to light intensity changes for responding to nest disturbances and relocating to new nest sites (please see line 307 in the revision). Also, from our observation, we found that when pharaoh ant nests are disturbed and exposed to light, queens would crazily search for a dark place and aggregate there, consistent with the inference that queens are sensitive to light intensity changes. Regarding line 284, we have specified that “the DEGs up-regulated in c16 were enriched by genes involved in *Drosophila* male courtship behavior” (please see line 281 in the revision).

Related to above, I found several of the claims regarding neural activity or behavior to be interesting ideas but poorly-supported by references or the data presented. Line 308-310 for example, “queens retain a good sensitivity to light but do not have good vision”—this is a very broad claim that is insufficiently discussed. What aspect of vision may not be good? Moreover, in line 300—the “rhythmic behavior” GO term seems very important to the author’s claims, and it would be useful here to specify what annotations are included (for example, this term includes locomotor rhythm—is that also found in the data here?).

Response: Given the significant contraction of all optic lobe cell clusters except for c20 (LMCs) in queen brains, we would expect that most vision-related functions degenerate compared to newly hatched uninseminated gynes. This loss of function makes general sense because inseminated ant queens normally live in permanent darkness. However, it is really difficult to discuss what aspect of vision may be affected, because it is still unclear what kind of vision young gynes of pharaoh ants exactly have. To make the description more precise, we have modified “queens retain a good

sensitivity to light but do not have good vision” to “most vision-related functions have degenerated in mature *Monomorium* queens, while the retained sensitivity to light intensity changes allows queens to assess the optimal time for nest-budding dispersal and to quickly retreat to the dark inner nest upon unexpected nest disturbance” (please see lines 304-307 in the revision).

The up-regulated genes driving the enrichment of GO term “rhythmic behavior” include *ort*, *qvr*, *GABA-B-R3*, *lh* and *wake*, all of which are responsible for regulating fly circadian rhythm. The GO term “locomotor rhythm” was not enriched by the c20 up-regulated genes. Therefore, we have modified “rhythmic behavior” to “circadian rhythm” to avoid confusion (see also line 297 in the revision).

Specialization and complementation of social brains—I find these results to be very interesting, but I think some of the authors’ claims are too strong or not given sufficient discussion/contextualization.

Line 345—“all of which reflect different use of the same set of cell types”. More accurately, the data presented can only suggest this possibility.

Response: We agree. We have modified the sentence to “These results confirm that the sexual and caste phenotypes of *Monomorium* ants are developmentally specialized to a remarkably high extent, and these specializations might have been resulted from the differential investment of a common set of cell types during pre-adult development.”

Lines 352-361—I don’t think the statements “worker brains are almost the opposite of male brains” (line 352) or “male and worker brains are functionally each other’s mirror image” (line 360) are adequately supported by their data or discussed. For example, no functional data is presented. In the absence of supporting data, these statements are empty or meaningless, and are thus susceptible to misinterpretation. I think the authors can strengthen their discussion here by contextualizing their results in the larger understanding of the social brain hypothesis and social insects.

Response: We agree. To make these descriptions more precise, we have modified the statement “worker brains are almost the opposite of male brains” to “worker brains were mostly characterized by cell-type preferences opposite to male brains”. We have also modified the statement “male and worker brains are functionally each other’s mirror image” to “male and worker brains are partially complementary to each other at cellular composition level in a *Monomorium* colony”. These revised statements avoid referring functions, but are instead focused on cell compositional differences between worker and male brains that could be supported by our data (please see lines 353-354 and lines 362-363 in the revision).

In the revised manuscript, we begin our discussion by reviewing the debate on the role of sociality in driving brain evolution and expand on the point where our study fits in the framework. We now emphasize the prospect of the new approach that we developed to accelerate studies to resolve this discussion (please see lines 461-484 in the revision).

Line 416 and beyond: “the gyne-queen transition” and “worker-gamergate transition”—I think the putative neuroprotective results are interesting. However, from the Methods, it does not appear that the authors directly studied the transition between the gyne state and the queen state. The

“transition” period suggests a developmental period between mating and before egg-laying, but I’m not sure from the methods that that is true of the animals used in this study (see comment below). The authors need to be very precise in language here that they are comparing gynes and queens, not directly probing the transition period. I do not think this takes away from their very interesting putative results on ensheathing glia and increased lifespan.

Response: We agree that the usage of the term “transition” was imprecise. We have now edited throughout the manuscript to avoid this kind of confusion.

Line 545-546 “all phenotypes remain disposable and replaceable”—I think that their results more strongly support the opposite—that the neural phenotypes found in *M. pha* greatly complement each other, and are thus none are disposable because all are required to make up the whole. What do the authors mean by this?

Response: What we meant by the sentence is that individual colony members within each phenotype are disposable, but we can see that our previous phrasing was not very clear. We have rephrased this paragraph. Rather than the ‘irreplaceable’ aspect, we have now emphasized the analogy with how cells in bodies turn over, as that was, we think, what inspired Wheeler. Please see lines 556-570 in the revised manuscript.

The Discussion should contextualize the main findings in the framework of the social brain hypothesis in social insects to strengthen the authors’ arguments and make this study of interest to a broad audience.

Response: We have now integrated our discussion into the framework of “social brain hypothesis” and reviewed the debate concerning the role of sociality in driving insect brain evolution. Please see lines 461-484 in the revised manuscript.

Data presentation and methods

Replicates and animals used in the study: The authors should take care to be upfront, clear, and consistent in the presentation of their biological replicates in the main text and Materials and Methods. Information about the animals used in the study is spread across lines 106-107, lines 591-593, lines 602-603, lines 641-642, Extended Figure 1, etc. It’s challenging to piece together the biological attributes of the animals used for study: i.e. how many n’s in each caste is not revealed until line 641, and even then it is confusing as whether these n’s are a subset of the “30-50 brains of the same adult phenotype” (lines 602-603) used for single nuclei isolation, or if each replicate is 30-50 brains, or if each replicate is a single individual (Extended Data 1). Please clarify. Additionally, what is the mating status of gynes/males?

Response: We have provided more details concerning the biological samples and replicates in the “Materials and Methods” section in the revised manuscript (please see lines 601-618). Briefly, whole brain snRNA-seq was performed for four to five biological replicates of each adult phenotype – gynes (n = 4), queens (n = 4), males (n = 4) and workers (n = 5). For a single biological replicate of an adult phenotype, nuclei for snRNA-seq were isolated from a pool of 30 to 50 whole brains. We have now also added this information in the first sentence of the Result section and in the legend of Figure 1.

Gynes and males were collected from colonies where the original (inseminated and fertile) queens were removed. Gynes and males were separated since the late pupal stage, so we could be sure they were virgins when sampled. These details have now been added in Materials and Methods as well (see lines 601-618), hopefully clarifying what we mean with mating status.

I found the statements in lines 436-437 and 562-564 to be very bold and not sufficiently supported. Can the authors please comment on how ages of the samples were determined (line 592-693), typical lifespan for the different phenotypes, how they ensured that gynes didn't mate, and briefly discuss any confounds that might exist in scenarios where age was not recorded or age within a phenotype varied greatly (i.e. queens).

Response: We can see that these two statements were too bold, so we have deleted the statement “We thus hypothesize that enforcement of ensheathing glia may have been a key mechanism by which the lifespan of inseminated queens increased relative to unmated workers in the ancestral ants” in the Result section, and toned down the second statement as “We speculate that ensheathing glia modification might represent one of the proximate mechanisms that ancestrally prolonged queen longevity in ants and whose co-option secondarily extended worker lifespan when they became inseminated as gamergate reproductives.” in the Discussion section (please see lines 583-586 in the revision).

As already indicated above, the gynes and males were obtained from newly split colonies where inseminated egg-laying queens were removed. Male pupae were continuously removed to prevent newly hatching gynes became inseminated. The pupal eclosion dates of males and gynes were recorded, so their age could be specified in intervals of 3-14 days and 5-10 days, respectively. Queens were collected from stable, mature colonies in which they had been actively laying eggs. *Monomorium pharaonis* queens usually live for 6-8 months in stable conditions¹⁹, so queen age in our samples was in that range but could not be better specified than 3-6 months. The lifespan of workers is around 10 weeks²⁰. Workers were randomly collected from colonies both inside and outside the nests, so our samples should have covered both young and old workers. These details have been added in Materials and Methods in the revision (please see lines 601 to 618).

To remind readers to be aware of age effect when interpreting our results, we have added the sentence “However, given that the queens (3-6 months old post-eclosion) were much older than the gynes (5-10 days old post-eclosion), some of these cellular changes could also reflect age rather than effects induced by insemination.” in the revision (see lines 391-394). In addition, we have re-structured the last section of the Results, in order to clarify that the age differences between our gyne and queen samples do not affect the inference about ensheathing glia playing a critical role in ant longevity (see lines 422-439 in the revision).

Line 627: is the in-house script publicly available?

Response: We have uploaded all the in-house scripts to the figshare repository under the link <https://doi.org/10.6084/m9.figshare.16616353>. We will make them publicly available along with other data in figshare upon acceptance.

Figure 2 lines 171-172: are the dots averages from individuals in each phenotype, or pooled samples? The bars here (e.g. KC bars) imply a spread that is greater than the data points shown.

Please specify in the figure legend if samples were pooled and specify in the methods if individual ants were barcoded.

Response: In figure 2, each dot presents the relative abundance of a cell type in an ant adult phenotype or in a *Drosophila* sex. The relative abundance of a focal cell type in a specific phenotype or sex, was measured as the percentage of cells belonging to the focal cell type out of the total number of cells in a specific phenotype or sex, after combining cells from all biological replicates (or all libraries) of the ant phenotype or *Drosophila* sex. Accordingly, bars are the corresponding means \pm s.d. across ant adult phenotypes or *Drosophila* sexes. We have added these statements in the legend of the new Figure 2 to facilitate understanding of this figure.

In this study, individual ants were not barcoded. Nuclei of a single replicate were isolated from a pool of 30 to 50 whole brains of the same phenotype. We have described the sample information more clearly in the Method section of the revised manuscript (see lines 601 to 618).

Figure 3: is cluster 31 also missing from males, or just very low? The latter seems to be what is suggested in Figure 4, and if that is accurate, an internal reference to Figure 4 could be useful here.

Response: Yes, the abundance of c31 KCs is very low rather than completely missing in males. We have now provided an internal reference to the new Supplementary Table 1 in the legend of Fig. 3, so that readers can find that the exact number of nuclei from each KC cluster in each adult phenotype.

Figure 4: the righthand side of panel A is not adequately explained in either the text or figure legend. There may be some typos here too (does Dm=Dm9?)

Response: In the revised manuscript, we have further improved the annotation of the *Monomorium* optic lobe neurons by comparing them to two latest *Drosophila* single-cell datasets that focus exclusively on the *Drosophila* optic lobes^{14,15}. Besides, we have provided more explanations and spelled out all the abbreviations in the legend of the new Fig. 4a.

Figure 4d: the dots here seem to represent individuals—please clarify in legend and methods if so (see above comment for Figure 2).

Response: One dot in Fig. 4d represent one biological replicate for a specific adult phenotype. We have clarified this in the legend of Fig. 4d in the revised manuscript.

Dopamine administration: because this manipulation seems very important to the authors' claims about the gonadotrophic effects of dopamine, they should expand on identification of yolky oocytes in the methods and provide images.

Response: We have supplemented the representative photos (see new Fig. 6d) and method descriptions related to the identification of yolky oocytes (see lines 966-973) in the revision. Briefly, to measure the number of yolky oocytes and the total surface area of yolky oocytes, the dissected ovaries were spread out, exposing all ovarioles, and then imaged with an Oplenic digital camera mounted to a Nikon SMZ800N stereomicroscope. Yolky oocytes are growing oocytes in the process of absorbing nutrients from hemolymph. They appeared as opaque, oval-shaped areas in the images as indicated in Fig. 6d by red dotted ovals. The total surface area of yolky oocytes

in an ovary was estimated as the summed area of these ovals. A total of 24 individuals were measured for each group and all the images were analyzed using EZ-MET software (x64, 6.0.7543).

Reviewer #4 (Remarks to the Author):

In this manuscript, the authors performed single-nucleus RNAseq for adult ant brains from males, gynes, workers, and queens. Through thorough analyses, the authors compared the brain cell composition and cell states among these four castes, as well as with brain cells from *Drosophila* and another ant species *Harpegnathos saltator*. A number of interesting observations were reported. This study stands as the first comprehensive single-cell transcriptomic profiling of brain cells from an ant colony, and will be an important addition for understanding the insect brain function and evolution. However, I have several major concerns about data analysis and data interpretation that the authors should address.

1. Line 156-159: The Authors should reconsider this statement. It is difficult to make a solid conclusion just based on cluster number, without functional validation. Under different resolutions (Leiden resolution for example), a group of cells can be assigned as one cluster or can be many. See recent discussion on Fly Cell Atlas, Figure S3 (Li, Janssens *et al.*, BioRxiv). Also, sub-clustering analysis can commonly reveal more cell subtypes. Total cell number also affects the cluster number.

Response: We thank the reviewer's advice on this. We have deleted the statement that the ants have more KC clusters than *Drosophila* in the revised manuscript. We also avoid drawing conclusions from comparing the number of cell clusters between species throughout the revised manuscript.

2. Fig 2c, d: When comparing cell compositions between ant and fly brains, two important things should be noted: First, current study used snRNAseq and Davie *et al.* 2018 used scRNAseq. scRNAseq, compared with snRNAseq, may have sampling bias. For example, glia cells tend to be more difficult to be isolated for scRNAseq than neurons. Second, when comparing a certain cell type, for example OPNs, the Authors should check carefully whether all OPNs have been annotated from referred datasets. For instance, it is possible that only a fraction of sequenced OPNs are annotated as OPN clusters due to the limit of available markers. This will cause a problem when making a conclusion based on the annotated cell number, but not the "real cell number".

Response: We agree that scRNA-seq may have sampling bias compared with snRNA-seq. We therefore have removed the statement that ants possess a somewhat higher ratio of glia vs neurons than *Drosophila* in the revised manuscript, as the ratio differences between ants and flies are indeed not that dramatic. Accordingly, we have discarded the old Fig. 2a and 2b. In addition, we removed the abundance comparison of optic lobe neurons between ant and *Drosophila* in Fig. 2, as we find that, according to the latest studies that focus exclusively on *Drosophila* optic lobes^{14,15}, the optic lobe neurons are very complex and comprise more than one hundred neuronal subtypes, which are really challenging to be fully resolved in a whole brain cell atlas.

But we have kept the abundance comparisons of KC, OPN, MN, AST, EN, CG and SG, because these neuronal or glial subtypes have been well studied in insects with established marker genes, so that they are relatively easy to be identified in a single-cell atlas. In addition, we have performed

manual check for the datasets used for comparison as suggested, to ensure that no cell clusters corresponding to KC, OPN, MN, AST, EN, CG and SG are missed. For example, this led us to identify two previously unannotated clusters (c9 and c10) as MNs and OPNs, respectively, in the *Harpegnathos* dataset from Sheng *et al* (2020)²¹, and to successfully annotate the glial subtypes of CG and SG that were originally not found in the *Drosophila* midbrain dataset from Croset *et al* (2018)⁹. Please see the sheets S2-1 (Sheng), S2-2 (Davie) and S2-7 (Croset) in the new Supplementary Table 2 for the curated annotations of all the public datasets used for comparison in Figure 2.

Overall, we believe that our findings about the higher abundances of KCs and OPNs in ants compared with *Drosophila* are very solid, because: 1) both the KCs and OPNs are not rare cell types but have notable abundances in both ants, 2) the abundance differences between ant and fly are quite dramatic (**KC**: ~24% in *Monomorium* whole brain vs ~5% in *Drosophila* whole brain, FC = 4.8; ~36% in *Harpegnathos* midbrain vs ~10% in *Drosophila* midbrain, FC = 3.8. **OPN**: ~3.0% in *Monomorium* whole brain vs ~1.0% in *Drosophila* whole brain, FC = 3; ~3.2% in *Harpegnathos* midbrain vs ~1.8% in *Drosophila* midbrain, FC = 1.78), 3) the more abundant KCs and OPNs in ant brains compared to fly brains are supported by the single-cell data collected from two different ant species, two different brain regions (whole brain and midbrain) and even two different platforms (snRNA-seq for *Monomorium* and scRNA-seq for *Harpegnathos* and *Drosophila*), and most importantly, 4) the higher abundances of KCs and OPNs in ants are well in line with the expected adaptations of ants to social and ground life that rely heavily on olfactory communication, which is less important in *Drosophila*.

3. Line 218-221: it is interesting that c13 shows eye-related function genes, but it should be noted that many axon guidance genes are conserved across different brain regions; so the GO term "photoreceptor cell axon guidance" doesn't necessarily mean it is specifically linked to eye functions, but may be linked with axon guidance of other types of neurons. Sometimes, or in most cases, GO terms cannot precisely predict a cell's function. It is also not clear to me how a KC cell type shows compound eye development features. Did the Authors mean retinal cells, like photoreceptors, pigment cells, and cone cells, or optical lobe neurons?

Response: The reviewer is right on this. It seems the genes in GO term 'photoreceptor cell axon guidance' do not really participate in photoreceptor/eye functions. For this GO term and others like "eye development", "eye morphogenesis" and "photoreceptor cell differentiation", the genes involved include *canoe*, *musashi*, *dyschronic*, *Fasciclin*, *prospero*, *shibire*, *Moesin*, *Nhe2*, *Mob2*, *cnk*, *Rap1*, *APC-like*, *lola*, *rdx*, *Liprin-alpha*, *nej* and *p120ctn*. These genes generally regulate cell proliferation, cell differentiation, cell signaling, development and morphogenesis in a variety of tissues including eyes. To avoid this confusion, we have deleted the functional descriptions of c13 KCs in the revised manuscript.

4. Line 344: the recent Fly Cell Atlas preprint (Li, Janssens *et al*, BioRxiv) also compared male and female cells between male and female fly heads. The Authors should refer to this newer and more comprehensive dataset for comparison.

Response: Thank you for the recommendation of this latest and comprehensive *Drosophila* dataset. We have conducted the same analysis to identify cellular compositional differences between male and female fly heads. As estimated with the whole brain dataset from Davie *et al* (2018), none of

the 82 cell clusters in the head dataset could be considered as showing credible abundance changes between sexes using the same criteria as in our ant dataset (see new Fig. 5c).

Please note that we have adopted a more advanced method, the scCODA framework¹ which is specifically designed for compositional single-cell data analysis (as recommended by Reviewer#2), to identify cell clusters with credible abundance differences between samples during manuscript revision. Please see line 845-871 in the methods for more details.

5. Figure 5d, and line 348: photoreceptors (PRs) are normally excluded from dissected brain (at least in flies), that's why PRs are not detected in fly brain atlas data (Davie *et al* 2018), but in head atlas data (Fly Cell Atlas); It is possible that the detected PRs from ant brains are mostly reflect dissection bias, but not real cell type composition. The authors should validate this part.

Response: The PRs could be from the ocelli that are not completely removed in the gyne, queen and male brains during dissection. In fact, there is a cluster annotated as ocelli in the *Drosophila* midbrain cell atlas (Croset *et al* 2018)⁹ and a cluster annotated as photoreceptors in the *Drosophila* whole brain cell atlas (Davie *et al* 2018)³, and their marker genes are quite similar. We have now added an explanation for the source of PRs in the first section of Results (see line 134 in the revision).

6. Fig 6a, line 385: it was concluded that a large fraction of clusters showed cell frequency changes from gynes to queens, like cluster 25, 27 etc. Since the insemination happened to adult gynes, where are these cell frequency changes coming from? Is there neurogenesis or programmed neuronal death during the transition from gynes to queens? If yes, more evidence or clarification should be provided.

Response: Given that queens are former adult gynes, the observed changes in cell-type abundance between gyne and queen brains could only be explained by the existence of neurogenesis and/or programmed neuronal death during the transition from gynes to queens, following insemination. Actually, neurogenesis as well as programmed cell death have been reported to occur in adults in many other insect lineages²²⁻²⁴, so it is not surprising that these cellular processes also exist in adult *Monomorium* gynes/queens. We have now added “.....suggesting that active neurogenesis and/or programmed cell death might occur during this role differentiation process in adult ant reproductives” in the revised manuscript, to help readers understand the potential mechanisms underlying the cell compositional changes between gynes and queens (please see lines 390-391).

7. In the result section of “Dopamine circuit remodeling...”, it is not very clear to me in which level the gyne and queen neurons are compared. Normally, there are two levels of comparison between two closely-related scRNAseq datasets, like current data of gynes and queens, cell composition level and gene expression level. In the cell composition level, it seems, for some clusters, there are significant changes (this is related to above point 6 and should be addressed). In the gene expression level, even if one cluster contains the same fraction of cells from gynes and queens, they may still show significant differences of certain genes' expression (see similar analysis in Figure 6A-D in Fly Cell Atlas, Li, Janssens *et al*). The Authors need to clearly distinguish these two types of comparison.

Response: We compared gyne and queen brains at the cell composition level. We have clarified the confusion in the corresponding section by emphasizing that the observed changes refer to cell abundance differences.

References

- 1 Buttner, M., Ostner, J., Muller, C. L., Theis, F. J. & Schubert, B. scCODA is a Bayesian model for compositional single-cell data analysis. *Nat Commun* **12**, 6876, doi:10.1038/s41467-021-27150-6 (2021).
- 2 Aitchison, J. The statistical analysis of compositional data. *Journal of the Royal Statistical Society: Series B (Methodological)* **44**, 139-160 (1982).
- 3 Davie, K. *et al.* A Single-Cell Transcriptome Atlas of the Aging *Drosophila* Brain. *Cell* **174**, 982-998 e920, doi:10.1016/j.cell.2018.05.057 (2018).
- 4 Li, H. *et al.* Fly Cell Atlas: a single-cell transcriptomic atlas of the adult fruit fly. *bioRxiv* (2021).
- 5 Dann, E., Henderson, N. C., Teichmann, S. A., Morgan, M. D. & Marioni, J. C. Differential abundance testing on single-cell data using k-nearest neighbor graphs. *Nature biotechnology*, doi:10.1038/s41587-021-01033-z (2021).
- 6 Fahrbach, S. E. Structure of the mushroom bodies of the insect brain. *Annual review of entomology* **51**, 209-232, doi:10.1146/annurev.ento.51.110104.150954 (2006).
- 7 Farris, S. M., Abrams, A. I. & Strausfeld, N. J. Development and morphology of class II Kenyon cells in the mushroom bodies of the honey bee, *Apis mellifera*. *The Journal of comparative neurology* **474**, 325-339, doi:10.1002/cne.20146 (2004).
- 8 Strausfeld, N. J., Sinakevitch, I. & Vilinsky, I. The mushroom bodies of *Drosophila melanogaster*: an immunocytochemical and golgi study of Kenyon cell organization in the calyces and lobes. *Microsc Res Tech* **62**, 151-169, doi:10.1002/jemt.10368 (2003).
- 9 Crosset, V., Treiber, C. D. & Waddell, S. Cellular diversity in the *Drosophila* midbrain revealed by single-cell transcriptomics. *Elife* **7**, doi:10.7554/eLife.34550 (2018).
- 10 Liu, Z. *et al.* Opposing intrinsic temporal gradients guide neural stem cell production of varied neuronal fates. *Science* **350**, 317-320, doi:10.1126/science.aad1886 (2015).
- 11 Traniello, I. M. *et al.* Meta-analysis of honey bee neurogenomic response links Deformed wing virus type A to precocious behavioral maturation. *Sci Rep* **10**, 3101, doi:10.1038/s41598-020-59808-4 (2020).
- 12 Suenami, S. *et al.* Analysis of the Differentiation of Kenyon Cell Subtypes Using Three Mushroom Body-Preferential Genes during Metamorphosis in the Honeybee (*Apis mellifera* L.). *PloS one* **11**, e0157841, doi:10.1371/journal.pone.0157841 (2016).
- 13 Suenami, S., Oya, S., Kohno, H. & Kubo, T. Kenyon Cell Subtypes/Populations in the Honeybee Mushroom Bodies: Possible Function Based on Their Gene Expression Profiles, Differentiation, Possible Evolution, and Application of Genome Editing. *Front Psychol* **9**, 1717, doi:10.3389/fpsyg.2018.01717 (2018).
- 14 Ozel, M. N. *et al.* Neuronal diversity and convergence in a visual system developmental atlas. *Nature* **589**, 88-95, doi:10.1038/s41586-020-2879-3 (2021).
- 15 Kumrangaliyev, Y. Z., Yoo, J., Valdes-Aleman, J., Sanfilippo, P. & Zipursky, S. L. Transcriptional Programs of Circuit Assembly in the *Drosophila* Visual System. *Neuron* **108**, 1045-1057 e1046, doi:10.1016/j.neuron.2020.10.006 (2020).
- 16 Becht, E. *et al.* Dimensionality reduction for visualizing single-cell data using UMAP. *Nature biotechnology*, doi:10.1038/nbt.4314 (2018).
- 17 Andrews, T. S., Kiselev, V. Y., McCarthy, D. & Hemberg, M. Tutorial: guidelines for the computational analysis of single-cell RNA sequencing data. *Nature protocols* **16**, 1-9, doi:10.1038/s41596-020-00409-w (2021).
- 18 Tay, J. W. & Lee, C. Y. Induced Disturbances Cause *Monomorium pharaonis* (Hymenoptera: Formicidae) Nest Relocation. *J Econ Entomol* **108**, 1237-1242, doi:10.1093/jee/fov079 (2015).
- 19 Pontieri, L. & Linksvayer, T. A. *Monomorium*. In: Starr C. (eds) Encyclopedia of Social Insects. *Springer, Cham.*, doi:10.1007/978-3-319-90306-4_171-1 (2019).
- 20 Pontieri, L. & Linksvayer, T. A. in *Encyclopedia of Social Insects* (ed Christopher Starr) 1-6 (Springer International Publishing, 2019).
- 21 Sheng, L. *et al.* Social reprogramming in ants induces longevity-associated glia remodeling. *Sci Adv* **6**, eaba9869, doi:10.1126/sciadv.aba9869 (2020).

- 22 Scotto Lomassese, S. *et al.* Influence of environmental stimulation on neurogenesis in the adult insect brain. *J Neurobiol* **45**, 162-171 (2000).
- 23 Trebels, B. *et al.* Adult neurogenesis in the mushroom bodies of red flour beetles (*Tribolium castaneum*, HERBST) is influenced by the olfactory environment. *Sci Rep* **10**, 1090, doi:10.1038/s41598-020-57639-x (2020).
- 24 Giraldo, Y. M. *et al.* Lifespan behavioural and neural resilience in a social insect. *Proceedings. Biological sciences / The Royal Society* **283**, doi:10.1098/rspb.2015.2603 (2016).

Decision Letter, first revision:

23rd March 2022

Dear Guojie,

Thank you for submitting your revised manuscript "A single-cell transcriptomic atlas tracking the neural basis of division of labor in an ant superorganism" (NATECOLEVOL-211014802A). It has now been seen again by the original reviewers and their comments are below (Reviewer #3 was not available to review but we have checked responses to their comments in house). The reviewers find that the paper has improved in revision, and therefore we'll be happy in principle to publish it in Nature Ecology & Evolution, pending minor revisions to satisfy the reviewers' final requests and to comply with our editorial and formatting guidelines.

You will see that Reviewer #1 feels strongly against framing the study around the social brain hypothesis. We understand that the main focus of the study is division of labour in ants and the title and abstract reflect that well. On balance, we think it is fine to keep the ant/fly comparison in the results but we agree with the reviewer that you need to de-emphasise the social brain hypothesis in the introduction and discussion. For example, the two sections should not open with that and if you want to keep some discussion around that theme in the middle of the introduction and discussion, it should be shortened and you will need to state very clearly that differences between ants and flies cannot be directly attributed to social status.

Before you revise your manuscript, please email us a copy of the file in an editable format (Microsoft Word or LaTeX). We will then perform detailed checks on your paper and will send you a checklist detailing our editorial and formatting requirements in about a week. Please do not upload the final materials and make any revisions until you receive this additional information from us.

[REDACTED]

Reviewer #1 (Remarks to the Author):

The revised manuscript is improved over the original submission. Unfortunately, the authors have tried to maintain their original focus (testing the Social Brain Hypothesis), even though all reviewers pointed out this focus was flawed and the comparisons conducted in the study do not enable this test:

All reviewers pointed out that the comparison between ants and flies is flawed for a range of reasons, that motivating the study primarily by the Social Brain Hypothesis is flawed, as is attempting to use the comparison between fly and ant brains to understand neural changes associated with the evolution

35of superorganisms. In their Response letter, the authors acknowledged and agreed with each of these issues. Unfortunately, in the revision, the authors have still motivated their study with the Social Brain Hypothesis, the idea that the evolution of superorganismality might be associated with the evolution of larger brains. As all reviewers pointed out, the current dataset does not allow assessment of this hypothesis.

The authors seem to understand this mismatch between their study design and their stated motivation (focus on Social Brain Hypothesis) too: In the introduction, they state L96-97 “we combined the power of massively parallel single-nucleus RNA sequencing (snRNA-seq) technology with the unique biology of the pharaoh ant *Monomorium pharaonis* to interrogate the neural correlates underlying division of labor and reproductive specialization.” And L110-L11 “This allowed us to map important aspects of the multi-brain complementarity of functional coordination within superorganismal pharaoh ant colonies.” Here, when the authors are describing what they did and their goals, there is no mention of comparison with flies and a goal of understanding the evolution of superorganismality, but rather with identifying neural correlates of division of labor and reproductive specialization, i.e. the within ant colony comparisons.

In my opinion, to be publishable, the manuscript would need to be rewritten so that stated motivation and discussion is actually in line with what was done.

Reviewer #2 (Remarks to the Author):

The authors have done a remarkable job addressing my comments and the comments of the other reviewers! The manuscript as it stands now is a very interesting and cohesive story and I am convinced it will be a great fit for the journal.

I only have one remaining comment. Perhaps I am missing something, but I do not fully understand the argument in lines 502-512. The authors state correctly that recent phylogenetic reconstructions place parasitoid wasps as early-branching lineages remote from the eusocial hymenopterans. But if both the early-branching parasitoid wasps and the late-branching eusocial ants/bees/wasps have elaborate mushroom bodies, the most parsimonious assumption is that their common ancestor (which was not eusocial) had elaborate mushroom bodies, too. Given that the authors have arrived to the opposite conclusion, could they perhaps explain their argument better?

Reviewer #4 (Remarks to the Author):

The authors have addressed all my concerns, and I support its publication.

Our ref: NATECOLEVOL-211014802A

25th March 2022

Dear Dr. Zhang,

Thank you for your patience as we've prepared the guidelines for final submission of your Nature Ecology & Evolution manuscript, "A single-cell transcriptomic atlas tracking the neural basis of division of labor in an ant superorganism" (NATECOLEVOL-211014802A). Please carefully follow the step-by-step instructions provided in the attached file, and add a response in each row of the table to indicate the changes that you have made. Please also check and comment on any additional marked-up edits we have proposed within the text. Ensuring that each point is addressed will help to ensure that your revised manuscript can be swiftly handed over to our production team.

****We would like to start working on your revised paper, with all of the requested files and forms, as soon as possible (preferably within two weeks). Please get in contact with us immediately if you anticipate it taking more than two weeks to submit these revised files.****

In recognition of the time and expertise our reviewers provide to Nature Ecology & Evolution's editorial process, we would like to formally acknowledge their contribution to the external peer review of your manuscript entitled "A single-cell transcriptomic atlas tracking the neural basis of division of labor in an ant superorganism". For those reviewers who give their assent, we will be publishing their names alongside the published article.

Nature Ecology & Evolution offers a Transparent Peer Review option for new original research manuscripts submitted after December 1st, 2019. As part of this initiative, we encourage our authors to support increased transparency into the peer review process by agreeing to have the reviewer comments, author rebuttal letters, and editorial decision letters published as a Supplementary item. When you submit your final files please clearly state in your cover letter whether or not you would like to participate in this initiative. Please note that failure to state your preference will result in delays in accepting your manuscript for publication.

Cover suggestions

As you prepare your final files we encourage you to consider whether you have any images or

37illustrations that may be appropriate for use on the cover of Nature Ecology & Evolution.

Nature Ecology & Evolution has now transitioned to a unified Rights Collection system which will allow our Author Services team to quickly and easily collect the rights and permissions required to publish your work. Approximately 10 days after your paper is formally accepted, you will receive an email in providing you with a link to complete the grant of rights. If your paper is eligible for Open Access, our Author Services team will also be in touch regarding any additional information that may be required to arrange payment for your article.

Please note that *Nature Ecology & Evolution* is a Transformative Journal (TJ). Authors may publish their research with us through the traditional subscription access route or make their paper immediately open access through payment of an article-processing charge (APC). Authors will not be required to make a final decision about access to their article until it has been accepted. [Find out more about Transformative Journals](https://www.springernature.com/gp/open-research/transformative-journals)

Authors may need to take specific actions to achieve [compliance with funder and institutional open access mandates](https://www.springernature.com/gp/open-research/funding/policy-compliance-faqs). If your research is supported by a funder that requires immediate open access (e.g. according to [Plan S principles](https://www.springernature.com/gp/open-research/plan-s-compliance)) then you should select the gold OA route, and we will direct you to the compliant route where possible. For authors selecting the subscription publication route, the journal's standard licensing terms will need to be accepted, including [self-archiving-and-license-to-publish](https://www.nature.com/nature-portfolio/editorial-policies/self-archiving-and-license-to-publish). Those licensing terms will supersede any other terms that the author or any third party may assert apply to any version of the manuscript.

For information regarding our different publishing models please see our page

href="https://www.springernature.com/gp/open-research/transformative-journals"> Transformative Journals page. If you have any questions about costs, Open Access requirements, or our legal forms, please contact ASJournals@springernature.com.

[REDACTED]

[REDACTED]

Reviewer #1:

Remarks to the Author:

The revised manuscript is improved over the original submission. Unfortunately, the authors have tried to maintain their original focus (testing the Social Brain Hypothesis), even though all reviewers pointed out this focus was flawed and the comparisons conducted in the study do not enable this test:

All reviewers pointed out that the comparison between ants and flies is flawed for a range of reasons, that motivating the study primarily by the Social Brain Hypothesis is flawed, as is attempting to use the comparison between fly and ant brains to understand neural changes associated with the evolution of superorganisms. In their Response letter, the authors acknowledged and agreed with each of these issues. Unfortunately, in the revision, the authors have still motivated their study with the Social Brain Hypothesis, the idea that the evolution of superorganismality might be associated with the evolution of larger brains. As all reviewers pointed out, the current dataset does not allow assessment of this hypothesis.

The authors seem to understand this mismatch between their study design and their stated motivation (focus on Social Brain Hypothesis) too: In the introduction, they state L96-97 "we combined the power of massively parallel single-nucleus RNA sequencing (snRNA-seq) technology with the unique biology of the pharaoh ant *Monomorium pharaonis* to interrogate the neural correlates underlying division of labor and reproductive specialization." And L110-L11 "This allowed us to map important aspects of the multi-brain complementarity of functional coordination within superorganismal pharaoh ant colonies." Here, when the authors are describing what they did and their goals, there is no mention of comparison with flies and a goal of understanding the evolution of superorganismality, but rather with identifying neural correlates of division of labor and reproductive specialization, i.e. the within ant colony comparisons.

In my opinion, to be publishable, the manuscript would need to be rewritten so that stated motivation and discussion is actually in line with what was done.

39Reviewer #2:

Remarks to the Author:

The authors have done a remarkable job addressing my comments and the comments of the other reviewers! The manuscript as it stands now is a very interesting and cohesive story and I am convinced it will be a great fit for the journal.

I only have one remaining comment. Perhaps I am missing something, but I do not fully understand the argument in lines 502-512. The authors state correctly that recent phylogenetic reconstructions place parasitoid wasps as early-branching lineages remote from the eusocial hymenopterans. But if both the early-branching parasitoid wasps and the late-branching eusocial ants/bees/wasps have elaborate mushroom bodies, the most parsimonious assumption is that their common ancestor (which was not eusocial) had elaborate mushroom bodies, too. Given that the authors have arrived to the opposite conclusion, could they perhaps explain their argument better?

Reviewer #4:

Remarks to the Author:

The authors have addressed all my concerns, and I support its publication.

Author Rebuttal, first revision:Reviewer #1 (Remarks to the Author):

The revised manuscript is improved over the original submission. Unfortunately, the authors have tried to maintain their original focus (testing the Social Brain Hypothesis), even though all reviewers pointed out this focus was flawed and the comparisons conducted in the study do not enable this test:

All reviewers pointed out that the comparison between ants and flies is flawed for a range of reasons, that motivating the study primarily by the Social Brain Hypothesis is flawed, as is attempting to use the comparison between fly and ant brains to understand neural changes associated with the evolution of superorganisms. In their Response letter, the authors acknowledged and agreed with each of these issues. Unfortunately, in the revision, the authors have still motivated their study with the Social Brain Hypothesis, the idea that the evolution of superorganismality might be associated with the evolution of larger brains. As all reviewers pointed out, the current dataset does not allow assessment of this hypothesis.

The authors seem to understand this mismatch between their study design and their stated motivation (focus on Social Brain Hypothesis) too: In the introduction, they state L96-97 “we combined the power of massively parallel single-nucleus RNA sequencing (snRNA-seq) technology with the unique biology of the pharaoh ant *Monomorium pharaonis* to interrogate the neural correlates underlying division of labor and reproductive specialization.” And L110-L11 “This allowed us to map important aspects of the multi-brain complementarity of functional coordination within superorganismal pharaoh ant colonies.” Here, when the authors are describing what they did and their goals, there is no mention of comparison with flies and a goal of understanding the evolution of superorganismality, but rather with identifying neural correlates of division of labor and reproductive specialization, i.e. the within ant colony comparisons.

In my opinion, to be publishable, the manuscript would need to be rewritten so that stated motivation and discussion is actually in line with what was done.

Response: We agree that the goal of our research was not to directly test Dunbar’s social brain hypothesis based on fly and ant comparisons, and that we could not directly attribute differences between fly and ant brains to insect social evolution. In the revised manuscript we have now removed all text related to the Dunbar social brain hypothesis, which is indeed not needed or particularly relevant for our study. We also deleted discussion paragraphs under the subtitle ‘Insect brain conservation and diversification: from solitary to superorganismal’ and avoided associated points elsewhere in the text suggesting differences between ant and fly brains relevant to social evolution. However, in the results section, we maintained the comparison between ant and fly brains. The purpose of doing so was to relate our findings to the best-studied non-social insect model system and to highlight the substantial differences between the two brain systems, particularly the more expanded and diverse KC cell types in ants relative to flies. We resolved the mismatch between our study design and the stated motivation by rewriting the first paragraph of introduction in which we now emphasized that the focus of our study was to interrogate exactly how and to what extent the ant colony has

developed a super-organismal brain-collective with specialized modules for castes and sexes at the cellular composition level. This thread of argumentation now forms a consistent narrative throughout our study.

Reviewer #2 (Remarks to the Author):

The authors have done a remarkable job addressing my comments and the comments of the other reviewers! The manuscript as it stands now is a very interesting and cohesive story and I am convinced it will be a great fit for the journal.

I only have one remaining comment. Perhaps I am missing something, but I do not fully understand the argument in lines 502-512. The authors state correctly that recent phylogenetic reconstructions place parasitoid wasps as early-branching lineages remote from the eusocial hymenopterans. But if both the early-branching parasitoid wasps and the late-branching eusocial ants/bees/wasps have elaborate mushroom bodies, the most parsimonious assumption is that their common ancestor (which was not eusocial) had elaborate mushroom bodies, too. Given that the authors have arrived to the opposite conclusion, could they perhaps explain their argument better?

Response: The parasitoid wasps are very distantly related to eusocial hymenopterans, and the parasitoid lifestyle of this solitary lineage is also expected to be cognitive demanding. Thus, the enlarged mushroom body in parasitoid wasps may be explained by convergent evolution. But we agree that this text section was not phrased very well in our previous version. We have now removed the discussion on this point in the revised manuscript due to word count limitation.

Reviewer #4 (Remarks to the Author):

The authors have addressed all my concerns, and I support its publication.

Final Decision Letter:

3rd May 2022

Dear Guojie,

We are pleased to inform you that your Article entitled "A single-cell transcriptomic atlas tracking the neural basis of division of labor in an ant superorganism", has now been accepted for publication in Nature Ecology & Evolution.

Over the next few weeks, your paper will be copyedited to ensure that it conforms to Nature Ecology and Evolution style. Once your paper is typeset, you will receive an email with a link to choose the appropriate publishing options for your paper and our Author Services team will be in touch regarding any additional information that may be required

You will not receive your proofs until the publishing agreement has been received through our system

Due to the importance of these deadlines, we ask you please us know now whether you will be difficult to contact over the next month. If this is the case, we ask you provide us with the contact information (email, phone and fax) of someone who will be able to check the proofs on your behalf, and who will be available to address any last-minute problems . Once your paper has been scheduled for online publication, the Nature press office will be in touch to confirm the details.

Acceptance of your manuscript is conditional on all authors' agreement with our publication policies (see www.nature.com/authors/policies/index.html). In particular your manuscript must not be published elsewhere and there must be no announcement of the work to any media outlet until the publication date (the day on which it is uploaded onto our web site).

Please note that *Nature Ecology & Evolution* is a Transformative Journal (TJ). Authors may publish their research with us through the traditional subscription access route or make their paper immediately open access through payment of an article-processing charge (APC). Authors will not be required to make a final decision about access to their article until it has been accepted. [Find out more about Transformative Journals](https://www.springernature.com/gp/open-research/transformative-journals)

Authors may need to take specific actions to achieve [compliance](https://www.springernature.com/gp/open-research/funding/policy-compliance-faqs) with funder and institutional open access mandates. If your research is supported by a funder that requires immediate open access (e.g. according to [Plan S principles](https://www.springernature.com/gp/open-research/plan-s-compliance))

44then you should select the gold OA route, and we will direct you to the compliant route where possible. For authors selecting the subscription publication route, the journal's standard licensing terms will need to be accepted, including <https://www.nature.com/nature-portfolio/editorial-policies/self-archiving-and-license-to-publish>. Those licensing terms will supersede any other terms that the author or any third party may assert apply to any version of the manuscript.

We welcome the submission of potential cover material (including a short caption of around 40 words) related to your manuscript; suggestions should be sent to Nature Ecology & Evolution as electronic files (the image should be 300 dpi at 210 x 297 mm in either TIFF or JPEG format). Please note that such pictures should be selected more for their aesthetic appeal than for their scientific content, and that colour images work better than black and white or grayscale images. Please do not try to design a cover with the Nature Ecology & Evolution logo etc., and please do not submit composites of images related to your work. I am sure you will understand that we cannot make any promise as to whether any of your suggestions might be selected for the cover of the journal.

You can generate the link yourself when you receive your article DOI by entering it here: <http://authors.springernature.com/share>.

[REDACTED]

P.S. Click on the following link if you would like to recommend Nature Ecology & Evolution to your

45librarian <http://www.nature.com/subscriptions/recommend.html#forms>

** Visit the Springer Nature Editorial and Publishing website at http://editorial-jobs.springernature.com?utm_source=ejp_NEcoE_email&utm_medium=ejp_NEcoE_email&utm_campaign=ejp_NEcoE for more information about our career opportunities. If you have any questions please click [here](mailto:editorial.publishing.jobs@springernature.com).**